# RoboCat: A Self-Improving Generalist Agent for Robotic Manipulation

**Konstantinos Bousmalis**[*], **Giulia Vezzani**[*], **Dushyant Rao**[*], **Coline Devin**[*], **Alex X. Lee**[*], **Maria Bauza**[*], **Todor Davchev**[*], **Yuxiang Zhou**[*], **Agrim Gupta**[*,1], **Akhil Raju**, **Antoine Laurens**, **Claudio Fantacci**, **Valentin Dalibard**, **Martina Zambelli**, **Murilo F. Martins**, **Rugile Pevceviciute**, **Michiel Blokzijl**, **Misha Denil**, **Nathan Batchelor**, **Thomas Lampe**, **Emilio Parisotto**, **Konrad Żołna**, **Scott Reed**, **Sergio Gómez Colmenarejo**, **Jon Scholz**, **Abbas Abdolmaleki**, **Oliver Groth**, **Jean-Baptiste Regli**, **Oleg Sushkov**, **Tom Rothörl**, **José Enrique Chen**, **Yusuf Aytar**, **Dave Barker**, **Joy Ortiz**, **Martin Riedmiller**, **Jost Tobias Springenberg**, **Raia Hadsell**[†], **Francesco Nori**[†], **and Nicolas Heess**[†]

All authors are affiliated with Google DeepMind, [*]Equal contributions, [†]Equal senior contributions, [1]Work done during an internship

**Reviewed on OpenReview:** https://openreview.net/forum?id=vsCpILiWHu

## Abstract

The ability to leverage heterogeneous robotic experience from different robots and tasks to quickly master novel skills and embodiments has the potential to transform robot learning. Inspired by recent advances in foundation models for vision and language, we propose a multi-embodiment, multi-task generalist agent for robotic manipulation. This agent, named *RoboCat*, is a visual goal-conditioned decision transformer capable of consuming action-labelled visual experience. This data spans a large repertoire of motor control skills from simulated and real robotic arms with varying sets of observations and actions. With Robo-Cat, we demonstrate the ability to generalise to new tasks and robots, both zero-shot as well as through adaptation using only 100–1000 examples for the target task. We also show how a trained model itself can be used to generate data for subsequent training iterations, thus providing a basic building block for an autonomous improvement loop. We investigate the agent's capabilities, with large-scale evaluations both in simulation and on three different real robot embodiments. We find that as we grow and diversify its training data, RoboCat not only shows signs of cross-task transfer, but also becomes more efficient at adapting to new tasks.

## 1 Introduction

Much of real-world robot learning research has focused on developing agents for one task at a time. This is because, even though the cost of task design and robot experience generation is very high, leveraging heterogeneous robot data at scale has remained a challenging problem in the field of robotics.

The advent of high-capacity models, such as the transformer model (Vaswani et al., 2017), has enabled recent successes for multi-task learning in language and vision. These developments have led to progress

in modelling multi-modal behaviour and predicting actions with a generalist agent, Gato (Reed et al., 2022), being able to play Atari, caption images, chat, and show some, albeit limited, robotic manipulation capabilities. Specifically in robotics, recent works (Brohan et al., 2022; Driess et al., 2023) have focused on bridging the gap between large pretrained language models and vision-based manipulation by training language-conditioned transformer policies to solve multiple simple, visually-diverse tasks that have the same observation and action spaces.

In this work, we propose RoboCat, a self-improving generalist agent for vision-based robotic manipulation, instantiated as a large transformer sequence model. Inspired by foundation models in other domains (Bommasani et al., 2022), we ultimately aim for a foundation agent for manipulation to be a multi-embodiment agent trained on a large set of robotic episodic experience that enables it to quickly adapt, via fine-tuning, to a broad set of new downstream tasks. As a step towards this goal, we trained RoboCat on a very large dataset of diverse manipulation behaviours: precise and dexterous vision-based tasks, performed with embodiments with different degrees of freedom, various observation and action specifications, and operating at different control frequencies. Our agent handles these variations natively without requiring common action or observation representations, by leveraging the transformer's ability to input and output variable-length sequences based on context. It is able to successfully adapt to multiple new tasks – including new robot embodiments, unseen behaviours, objects and perceptual variants, and sim-to-real – via fine-tuning on a small dataset of new episodic experience of between 100 to 1000 demonstrations. This significantly reduces the cost of acquiring new skills and onboarding new embodiments. We further use the fine-tuned RoboCat models to gather additional data that is later added to train new iterations of our agent. This *self-improvement* process, illustrated in Figure 1, makes for a more capable agent, improving its cross-task transfer and fine-tuning capabilities to even more tasks, and demonstrating better performance on existing tasks. We therefore demonstrate fine-tuning to a large range of unseen tasks at multiple stages of this self-improvement process, in addition to generalist capabilities on training tasks.

RoboCat is based on the Gato architecture with a VQ-GAN encoder (Esser et al., 2021) pretrained on a broad set of images; this choice of encoder enables fast training and iteration. We specify tasks via visual goal-conditioning, which has the desirable property that any image in a trajectory can be labelled as a valid "hindsight goal" (Andrychowicz et al., 2017) for all time steps leading up to it. This means that hindsight goals in existing data can be extracted without additional human supervision and that even suboptimal data collected by the agent can be incorporated back into the training set for self-improvement. Additionally, visual goals provide an intuitive interface to indicate to the robot which task it should perform.

Our main contributions in this work are outlined below: *(1)* we demonstrate, for the first time, that a large transformer sequence model can solve a large set of dexterous tasks on multiple *real* robotic embodiments with differing observation and action specifications; *(2)* we investigate RoboCat's capabilities in adapting to unseen tasks, with just a small dataset of expert demonstrations, lowering the bar of learning a new skill, compared to baselines; *(3)* we show that it is possible to incorporate these skills back to the generalist with a simple but effective self-improvement process; and *(4)* we show that by scaling and broadening the training data, RoboCat performs better on training tasks and is more efficient at fine-tuning.

The rest of the paper is structured as follows. We first describe RoboCat and the self-improvement loop in Section 2. We introduce the embodiments, tasks, and object sets that we have used in this work in Section 3.

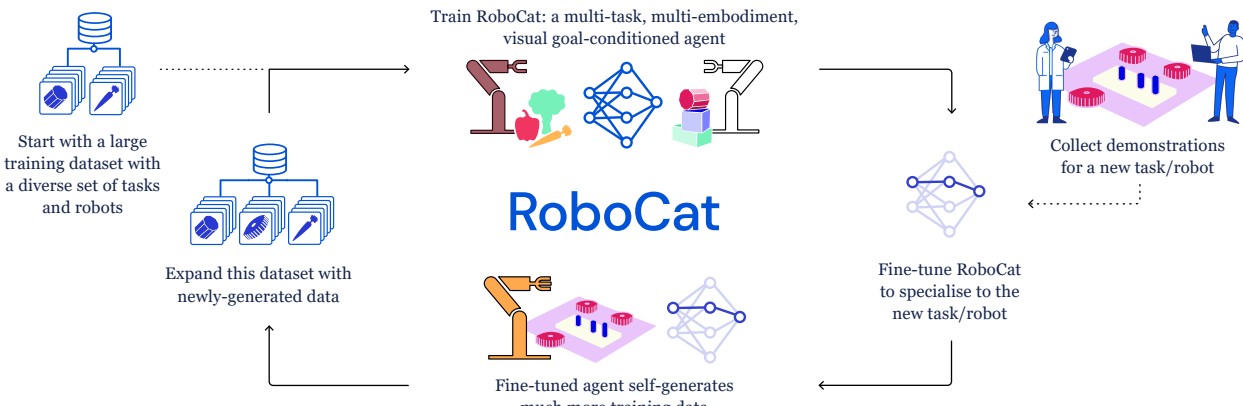

Figure 1: **The self-improvement process.** RoboCat is a multi-task, multi-embodiment visual goal-conditioned agent that can iteratively self-improve. A diverse training set is used to train an initial version of this generalist agent, which can be fine-tuned to new tasks with 100–1000 demonstrations and then deployed on real robots to generate much more data for these tasks. The resulting trajectories are then added to the training dataset for the next iteration of RoboCat, increasing the generalist's repertoire of skills and improving performance across tasks. Our experiments demonstrate one successful iteration of this process.

We describe our experimental setup for both training and evaluation in Section 4, before we present our extensive experiments to support our claims in Section 5. We finally discuss our work in the context of related work in Section 6, and discuss RoboCat's potential avenues for future work in Section 7.

## 2 RoboCat

We introduce RoboCat, a self-improving generalist agent for robotic manipulation that can perform multiple tasks and control multiple embodiments in simulation and the real world. In this section, we describe each phase of the RoboCat training process, summarised in Figure 1. In the *training* phase, the VQ-GAN tokeniser is *pre-trained*, and then the RoboCat generalist agent is trained on a wide dataset covering multiple domains and embodiments, specifying tasks via visual goals. The generalist is then finetuned on a small set of human-teleoperated demonstrations to specialise to a new task, and deployed to collect on-policy data autonomously. This data is finally added to the original data to train the next, *self-improved* RoboCat.

### 2.1 Training and task specification

We consider vision-based tabletop object manipulation tasks. Each task is defined by its (uncountably infinite) set of valid start and end states, and an episode is evaluated for task success by checking if the last state is in the set of valid end states. For example, for the task "Insert the apple into the bowl", the set of valid start states is all states with an apple outside a bowl, and the set of valid end states is all states with the apple inside the bowl. We exclusively consider tasks where success can be determined from only the end state.

We want to train an agent that performs a task when conditioned on an image of a valid end state of that task. Our goal-conditioned agent is represented by a policy $\pi(a_t|o_t, g_t)$, where $a_t$ denotes the action vector, $o_t = (x_t, I_t)$ are the proprioceptive observation (e.g. robot joint positions and velocities) and image observation, respectively, and $g_t$ is an image of the desired task. Note that the goal image is an example of

the task being solved and it does not indicate a specific state that the agent should reach. The goal image effectively indicates the task that the agent should do and the agent is only evaluated for task success.

We model $\pi(a_t|o_t, g_t)$ via an autoregressive transformer model (Vaswani et al., 2017),

$$\pi(a_t|o_t, g_t) = P_\theta(a_t|x_{<t}, I_{<t}, g_{<t}), \tag{1}$$

where the subscript $< t$ denotes observations and goal images prior to time step $t$. Note that the dimensionality of the actions and proprioception observations vary across embodiments. Internally, the autoregressive model operates with tokenised inputs and outputs.

For training, we assume access to a dataset $\mathcal{D} = \{\tau^i\}_{i=1}^{|\mathcal{D}|}$ of trajectories that are transformed into a dataset of tokenised trajectories $\hat{\mathcal{D}} = \{\hat{\tau}^i\}_{i=1}^{|\mathcal{D}|}$. In addition, during tokenisation, the trajectories are augmented with goal images. Concretely, a tokenised trajectory $\hat{\tau} \in \hat{\mathcal{D}}$ is represented as $\hat{\tau} = \left( x_1^{1:L}, I_1^{1:M}, g_1^{1:N}, a_1^{1:Q}, ..., \right.$ $\left. x_T^{1:L}, I_T^{1:M}, g_T^{1:N}, a_T^{1:Q}, x_{T+1}^{1:L}, I_{T+1}^{1:M}, g_{T+1}^{1:N} \right)$, where $L, M, N, Q$ denote the number of tokens required to encode proprioceptive inputs, images, goals, and actions, respectively, and $T$ is the number of transitions in the trajectory. Note that $L$ and $Q$ vary by embodiment. The goal observations $g_t$ are fixed within a trajectory and repeated for each time step.

A natural choice for a goal image is a *hindsight goal*. Since, by definition, a trajectory always "succeeds" at reaching its own last image, we can use the last image of the same episode as the goal image, $g_t^i = I_{T+1}^i$, for any trajectory $\tau^i$. Alternatively, we can also consider goal selection using a *semantically-equivalent goal*. That is, for any successful episode $\tau^i$, we can select the last image of a different episode that succeeded at the same task, $g_t^i = I_{T+1}^j$, where $\tau^j$ is another successful episode from the dataset $\mathcal{D}$, as measured by a success detector or reward function for a given task. We train with both sources of goals for successful episodes, and use only hindsight goals for unsuccessful episodes. Details on how we weight the different tasks and goal sources are available in Appendix E.2.

### 2.1.1 Architecture and pretraining

Our model is based on the transformer architecture described in Gato (Reed et al., 2022). For tokenisation of proprioceptive observations and agent actions, we follow the same procedure as in Reed et al. (2022). For image tokenisation, however, we instead use a pretrained and frozen VQ-GAN (Esser et al., 2021), which allows for faster training of the generalist, as the image can be tokenised once in advance. The VQ-GAN, similarly to a VQ-VAE (van den Oord et al., 2017), consists of an encoder that encodes an input image into a series of latent vectors and a decoder (which we do not use after training). The encoded vectors are discretised via a nearest neighbour lookup in a codebook of quantised embeddings. Each image is tokenised into an $8 \times 8$ grid of tokens.

We pretrain our VQ-GAN encoder on a *diverse* collection of images as we find this improves generalisation. Specifically, the encoder is trained on a dataset that consists of images from ImageNet (Deng et al., 2009), images from the control tasks in Reed et al. (2022) including Atari and MuJoCo (Todorov et al., 2012) locomotion tasks, as well as images from our visual robotic manipulation dataset. These datasets, training details, as well as extensive ablations that informed our design choices can be found in Appendix D.

To train the agent model we use a dataset $\hat{\mathcal{D}}$ containing the joint collection of data from all training tasks (see Section 3) and utilise a standard token prediction loss. While Gato only predicted actions, we find that, when a VQ-GAN is used, performance is improved by additionally training for predicting future image tokens as produced by the VQ-GAN encoder (Appendix D.3). Specifically, we predict image tokens $k = 5$ time steps into the future as images one step apart can look very similar.

Combining the action and observation prediction losses, at the token level, we obtain the following objective to train the model $P_\theta$: $\mathrm{L}(\theta; \mathcal{D}) = \mathbb{E}_{\hat{\tau} \sim \hat{\mathcal{D}}} \left[ \sum_{t=1}^{T} \sum_{q=1}^{Q} \log P_\theta(a_t^q | x_{<t}^{1:L}, I_{<t}^{1:M}, g_{<t}^{1:N}) \right.$
$\left. + \sum_{t=1}^{T+1-k} \sum_{m=1}^{M} \log P_\theta(I_{t+k}^m | x_{\leq t}^{1:L}, I_{\leq t}^{1:m}, g_{<t}^{1:N}) \right]$. Note that, in practice, instead of conditioning on the full history of observations (as indicated by the subscript $< t$), we use a fixed total token length of 1024 for the model (which corresponds to roughly 3 time steps of history).

### 2.1.2 Fine-tuning and self-improvement

A key contribution of our work is our study into how RoboCat agents can be fine-tuned and self-improved given a relatively small number of demonstrations. This capability is especially crucial in a real robotics context—unlike in simulation, data is bottlenecked by real-time operation per robot, and high-quality supervision is scarce.

**Fine-tuning** To perform fine-tuning and self-improvement we first collect 100–1000 demonstrations per task via teleoperation. The generalist RoboCat agent is fine-tuned on these demonstrations, which are tokenised and augmented with goal images in the same way as for the generalist training. Formally, we perform the optimisation $\theta_{\mathrm{ft}}^y = \arg\max_\theta \mathcal{L}(\theta; \mathcal{D}_{\mathrm{demo}}^y)$ where $\mathcal{D}_{\mathrm{demo}}^y$ is the demonstration data for the task $y$ that we want to fine-tune on, and $\theta$ is initialised with the weights from pretraining (Section 2.1.1). At the end of this fine-tuning step, we obtain an agent that is specialised to the new task but that may lose performance on the original training tasks.

**Self-improvement** In order to integrate new tasks into a new generalist, we deploy the fine-tuned policies $P_{\theta_{\mathrm{ft}}^y}$ to autonomously collect a large dataset of additional trajectories for each of the self-improvement tasks $y \in \mathcal{Y}$. After data collection, we perform hindsight goal relabelling as described in Section 2.1. Note that, when using semantically-equivalent goals, we require a reward function to determine the successful trajectories for a given task. For this purpose, we employ learned reward models as described in the next section. The resulting relabelled trajectories form a self-improvement dataset $\mathcal{D}_{\mathrm{imp}}^y$ for the task we want to improve. Finally, using this data, we can construct a new training dataset for training the next iteration of our generalist RoboCat agent. We combine all trajectories with the previous data to form the next dataset,

$$\mathcal{D}_{\mathrm{next}} = \mathcal{D} \cup \bigcup_{y \in \mathcal{Y}} \left( \mathcal{D}_{\mathrm{demo}}^y \cup \mathcal{D}_{\mathrm{imp}}^y \right), \tag{2}$$

which is then used to train a new VQ-GAN model, after which we continue with the next iteration of training a new generalist $\theta_{\mathrm{next}} = \arg\max_\theta \mathcal{L}(\theta; \mathcal{D}_{\mathrm{next}})$.

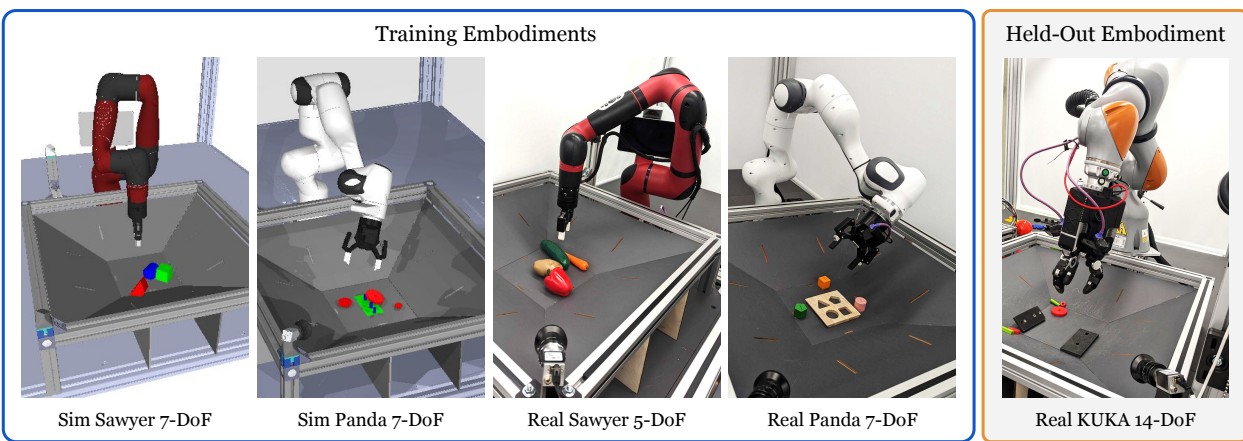

Figure 2: **RoboCat supports multiple robotic embodiments and control modes.** These are all the different embodiments RoboCat is tested on, and the dimensionality of the action it needs to output for each. All robot arms have a Robotiq parallel gripper attached to them, with the exception of the KUKA arm which has a proprietary three-finger hand. Unlike the Panda and Sawyer embodiments, the KUKA embodiment was not seen during training and is only used during fine-tuning.

## 2.2 Real-world deployment

In order to integrate the new task into a new generalist, we deploy the fine-tuned policy on a real robot to collect a large dataset on the new task using images from the demonstrations as goal images. Collecting real-world data autonomously presents two challenges: success classification and task resets.

**Success detection via reward models** While final evaluation numbers are counted manually for accuracy, automated success detection is necessary for the hindsight goal relabelling of semantically-equivalent goals described above during training. In addition, success detection is necessary for determining when a reset is needed. To this end, we train vision-based reward models to detect when a task has succeeded. We first collect human demonstrations and data from policies trained to perform the task (e.g. evaluation episodes of a RoboCat policy). These episodes are annotated via a crowd-sourcing interface, where annotators mark the time step after which the task is solved in each episode (if at all), resulting in binary annotations. These are then used to train a binary classifier that can be used to detect task success from image observations at any given time step.

**Autonomous resets with policy pools** Resetting the environment for a single task requires bringing the state from the end state back into the set of valid start states for that task. However, manually programming such reset routines is a highly non-trivial endeavour (in many cases performing a reset is almost as complicated as solving the task itself) leaving us with a problem for autonomous data collection. We solve this issue by observing that the set of end states for some tasks overlap with the set of start states of other tasks. Thus we can "re-use" tasks trained for a given task as reset mechanisms for tasks whose end states overlap with the valid start states for another task. We implement an autonomous reset mechanism based on this observation that we refer to as a *policy pool*. A policy pool is simply a collection of policies (or policies implicitly defined by a pool of goal images) with overlapping start and end states. In each episode, we then pick a policy from this pool to be run next and record its trajectory and success. By pooling multiple policies in this way, we can get automated resets, increase the robot utilisation (by reducing the need for explicit human resets) and increase the diversity of initial conditions for evaluation and data collection. We utilise two types of policy

pools in our evaluations: *stateless* policy pools, in which the policies are executed in some order regardless of the state of the environment (e.g. for lifting tasks); and a *state-based* policy pool, which samples the next policy to execute based on the state of the environment (e.g. performing a remove task when the initial state corresponds to a successful insertion). In the latter case, the trained reward models are used to evaluate the state of the tabletop and determine which policies are eligible for next execution. More details are provided in Appendix F.2.

## 3 Tasks and Data

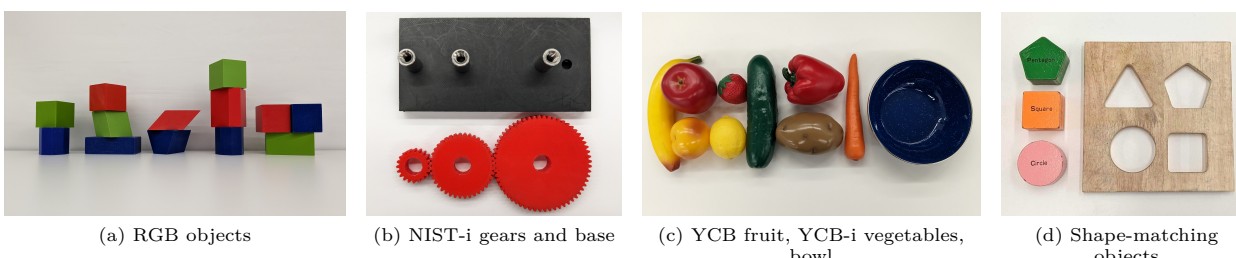

(a) RGB objects      (b) NIST-i gears and base      (c) YCB fruit, YCB-i vegetables, bowl      (d) Shape-matching objects

Figure 3: **The real-world object sets used by RoboCat.** The first two object sets are used to systematically study structure-building and insertion affordances, respectively. The other object sets are store-bought objects that add visual diversity and challenge the agent with various lifting, insertion, and removal tasks.

One of the main contributions of this work is to demonstrate that RoboCat can learn diverse and dexterous behaviours to solve a large set of tasks. The tasks we use require fine motor skills and hand-eye coordination, and the understanding of complex affordances and multi-object interactions. Additional diversity in the data is obtained through the use of multiple simulated and real embodiments and different approaches to data generation: RL-trained expert trajectories, human-teleoperated demonstrations, as well as self-generated data from RoboCat (see Section 2.1.2). In this section, we provide an overview of the embodiments, object sets, tasks, and datasets that we refer to in this paper.

### 3.1 Embodiments

The embodiments used in this work, shown in Figure 2, are all in a standardised cage (see Lee et al. (2021)), which contains a "basket" that defines the robot arm's workspace. RoboCat was trained with data from Rethink Sawyer arms controlled with 7-DoF (simulation) and 5-DoF (real), and Franka Panda robot arms controlled with 7-DoF (simulation and real), all fitted with a Robotiq parallel gripper. These action spaces comprise 6-DoF and 4-DoF Cartesian end-effector control with an additional dimension for the gripper action. The proprioception observations for Panda and Sawyer have different dimensionalities, and even for the common 7-DoF case, the physical and kinematic characteristics between the embodiments means that the action distributions are different. RoboCat is also able to control KUKA 14-DoF arms, which are fitted with a new, custom-made, three-finger robot hand[1]—an embodiment that was only seen during the fine-tuning phase. In total, we used 36 real robots in this work: 15 Panda, 17 Sawyer, and 4 KUKA arms.

The simulated Panda and Sawyer embodiments are analogous to the real ones, though they are only coarsely aligned. We did not perform any careful system identification and the images rendered from the simulation

---

[1] Details of this robot hand will be released in the near future.

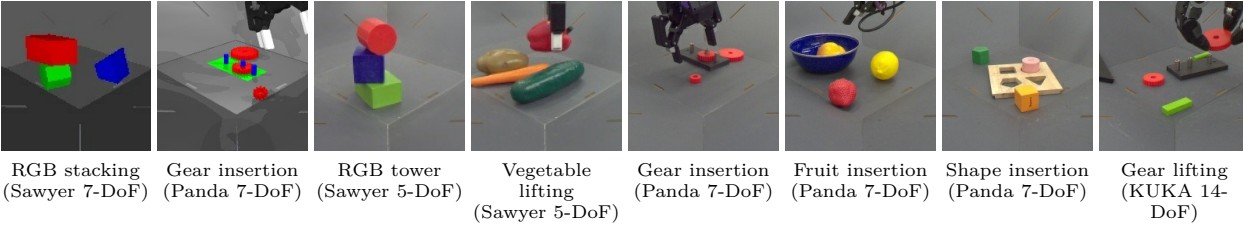

RGB stacking (Sawyer 7-DoF) | Gear insertion (Panda 7-DoF) | RGB tower (Sawyer 5-DoF) | Vegetable lifting (Sawyer 5-DoF) | Gear insertion (Panda 7-DoF) | Fruit insertion (Panda 7-DoF) | Shape insertion (Panda 7-DoF) | Gear lifting (KUKA 14-DoF)

Figure 4: **Example goal images.** These images correspond to a subset of the embodiments, task families, and object sets used by RoboCat. The first two images correspond to simulated embodiments and the remaining images to real-world embodiments. See Figure 12 for more examples.

were not visually realistic. We randomised the physics parameters in simulation but we did not randomise the visual appearance of the scene. More details about embodiments are available in Appendix C.

## 3.2 Object sets

We use different object sets and a total of 134 real objects to enable a variety of complex behaviours and affordances (see Figure 3). The first two sets of objects are 3D-printed and have been designed to systematically study types of robotic manipulation that involve multi-object interaction, specifically, structure-building (RGB objects) and insertion (NIST-i gears). We use a subset of these (123 objects) in simulation. The other real-world sets include store-bought objects.

**RGB objects** These 116 objects with parametrically defined shapes (only a subset shown in Figure 3(a)) were introduced as a benchmark (Lee et al., 2021) to systematically study the physical understanding of multi-object interactions in the context of stacking: To solve the benchmark an agent needs to understand which shapes in which poses can be reliably stacked on top of each other. We use them here to additionally study related structure-building tasks. The basket always contains a triplet of these objects, with respective colours red, green, and blue.

**NIST-i gears and 3-peg base** This set of objects is first introduced in this work to aid a systematic study of the insertion affordance. Inspired by the NIST benchmark for robotic manipulation (Kimble et al., 2020), we designed three gears, different in sizes (small, medium, large), which are to be used in conjunction with a 3-peg base. The pegs are spaced such that successful meshing requires a specific allocation of the gears to the pegs (see Figure 3(b)). In the real world, the shafts are metallic and the base is not fixed to the basket, which significantly increases the difficulty of the task. In simulation, the base is fixed. In both cases, there is a 1 mm tolerance when inserting a gear. See Appendix B.1.4 for more details.

**YCB fruits, YCB-i vegetables, and bowl** In this work, we use a subset of the YCB object set (Calli et al., 2017), namely the fruit (apple, banana, peach, lemon, strawberry), shown in Figure 3(c). The YCB-i vegetables (carrot, cucumber, pepper, potato) and bowl, also shown in Figure 3(c), are inspired by, but not part of, the official YCB benchmark. This collection of textured and geometrically different objects introduces additional visual diversity and allows us to benchmark RoboCat on tasks with everyday objects.

**Shape-matching objects and base** These wooden objects are parts of a real shape-matching cube, used by toddlers to practice fine-motor control skills and 3D shape understanding. We used three shapes (circle, pentagon, square) and the shape-matching cube lid, shown in Figure 3(d). The lid is used as a base with matching holes that can be used for insertion and removal tasks. These objects are used to further study

the insertion affordance. Unlike the NIST-i gears, this toy often requires difficult reorientations to get the objects in the correct orientation for insertion.

### 3.3 Task families

We consider a total of 253 different task variations which we group into task families. We define a *task family* to be a group of tasks that utilise the same skill or sequence of skills. For example, lifting the large NIST-i gear and lifting the YCB apple are two different task variations from the same task family. We provide a complete list of the task families in Table 1.

The task families **stacking**, **tower building**, **pyramid building**, and **inverted pyramid building** consist of building structures with either RGB objects or gears. They differ in difficulty, but in all cases require dexterous and precise movements to ensure that the structure remains stable after completion. The **lifting** task family consists of picking up a specific object in a basket with multiple objects. The objects are either fruits, vegetables, or gears. The motivation behind the lifting tasks is to study goal understanding and generalisation to new embodiments and objects. The **insertion** and **removal** task families come in three flavours, either involving fruits and a bowl, gears and a 3-peg base, or shape-matching objects and a base. We treat them as separate task families since they require different skills. The latter two require precise positioning into low-tolerance pegs or base, and shape-matching requires shape understanding and often reorientation. The bowl and bases can freely move in the real world, which substantially increases the complexity of those tasks. For all insertion and removal tasks, we use no resets other than the learnt respective removal and insertion tasks.

Each task variation refers to the combination of a specific embodiment (e.g. simulated Sawyer vs real-world Panda), task family, object set (e.g. RGB triplet 1 vs NIST-i gears), and perceptual variation (e.g. stacking red on blue vs green on red objects). Example goal images corresponding to specific task variations are shown in Figure 4.

### 3.4 Data sources

RoboCat is trained on both expert and non-expert data. Different subsets of the data are collected in different ways. We use three types of data generation: (i) data produced by specialist RL agents, particularly employed in simulation; (ii) human teleoperated expert data, mostly used for the physical world tasks; and (iii) self-generated data. The primary difference between the two expert types of trajectories is that agent data provides fairly smooth and efficient trajectories due to the way the RL agent acts in the world, while teleoperated data often includes pauses as teleoperators employ behaviours similar to a bang-bang controller. The self-generated data is obtained by running extensive evaluations whenever a new version of RoboCat is available: the data collected this way is saved and then reused for the next RoboCat training. This data is collected from RoboCat agents fine-tuned on teleoperated expert data. Therefore, the self-generated data resemble the teleoperation behaviours. See Appendix B.2 for further details about the nature of the data.

| | Embodiment | Task Family | Object Set | Training Task Variations | Evaluation Task Variations | Average Task Success |
|---|---|---|---|---|---|---|
| Training / Simulation | Sawyer 7-DoF | Stacking | RGB objects & NIST-i gears | 28 & 5 | 28 & 5 | 82% |
| | Panda 7-DoF | Stacking | RGB objects & NIST-i gears | 30 & 6 | 30 & 6 | 80% |
| | | Tower building | RGB objects & NIST-i gears | 8 & 3 | 8 & 3 | 60% |
| | | Pyramid building | RGB objects | 30 | 30 | 65% |
| | | Lifting | NIST-i gears | 3 | 3 | 88% |
| | | Insertion-peg | NIST-i gears | 3 | 3 | 75% |
| Training / Real World | Sawyer 5-DoF | Stacking (red on blue) | RGB objects | 92 | 5 | 80% |
| | | Stacking (blue on green) | RGB objects | 1 | 1 | 45% |
| | | Tower building | RGB objects | 1 | 1 | 23% |
| | | Inverted pyramid building | RGB objects | 1 | 1 | 17% |
| | | Lifting | YCB-i vegetables | 4 | 4 | 54% |
| | Panda 7-DoF | Lifting | YCB fruits | 16 | 4 | 54% |
| | | Lifting | NIST-i gears | 3 | 3 | 94% |
| | | Insertion-peg | NIST-i gears | 3 | 3 | 77% |
| | | Removal-peg | NIST-i gears | 3 | 3 | 97% |
| Fine-tuning / Real World | Panda 7-DoF | Insertion-bowl | YCB fruits and YCB-i bowl | 3 | 3 | 84% / 84% |
| | | Removal-bowl | YCB fruits and YCB-i bowl | 3 | 3 | 64% / 72% |
| | | Insertion-base | Shape-matching objects | 3 | 3 | 6% / 13% |
| | | Removal-base | Shape-matching objects | 3 | 3 | 70% / 82% |
| | KUKA 14-DoF | Lifting | NIST-i gears | 1 | 1 | 56% / 86% |

Table 1: **Final RoboCat performance on evaluation tasks.** This table lists the tasks used for training and fine-tuning of the final RoboCat agent, and highlights the set of tasks used in the self-improvement process. The success rates are averaged across all the respective evaluation task variations. For fine-tuning experiments, we report success rates when fine-tuning on 500 and 1000 demonstrations, respectively. Note that data from the fine-tuning tasks are unseen during generalist training and the fine-tuned agent only uses up to 1000 demonstrations alone for these new tasks.

## 4 Experimental Setup

### 4.1 RoboCat training tasks

The full RoboCat agent is trained on 240 tasks and fine-tuned on a further 13 tasks, for a total of 253 tasks. This includes data from 2 simulated and 3 real-world embodiments, 5 simulated and 11 real task families, and 123 simulated and 134 real objects. Table 1 summarises the tasks, organised separately for **training** and **fine-tuning** tasks. An important contribution is the fact that the RoboCat generalist agent can self-improve, by fine-tuning the previous iteration of the generalist to new tasks, and self-generating additional data for the next round of generalist training. These self-improvement tasks (unseen in the previous RoboCat iteration) are indicated in Table 1, and more detailed experiments are presented in Section 5.3.

### 4.2 Training and fine-tuning

We train our generalists following the procedure outlined in Reed et al. (2022) except for differences in the encoder where applicable. The majority of the experimental results are based on models with a 1.18B-parameter decoder-only transformer (Vaswani et al., 2017) with 24 layers, an embedding size of 2048, and a post-attention feedforward hidden size of 8196. To allow for more extensive experimentation, we use smaller models with 364M parameters for a few ablations (Figure 5(a), Figure 10, Appendix G.2, and Appendix G.4).

We fine-tune our generalists on a set of diverse real tasks using a limited number of human teleoperation demonstrations, between 100 and 1000 demonstrations for each task.

### 4.3 Evaluation

For each of the simulated and real tasks, we evaluate each model by averaging over 100 episodes (or more, if specified), using a different goal image for each episode as well as randomised initial states of the environment. The episode length and control frequency varies from task to task, always matching the length of the expert data used for training. The control frequency of training data is not provided to the agent during training, since it may not be known or readily available. Table 14 in Appendix F report the episode length and control frequency used for each task family in simulation and real.

When fine-tuning a generalist to a specific real-world task, it can be difficult to determine the optimal number of fine-tuning steps, since there is no reliable offline measure of success. To address this in a systematic and reproducible way, we employ the following evaluation protocol for each task: we first evaluate the checkpoint every 5000 steps for 25 episodes each to assess the best performing checkpoint, and then evaluate that checkpoint for 100 episodes to measure the final performance.

### 4.4 Baselines

In order to contextualise the difficulty of the tasks, we compare RoboCat to high capacity, pretrained vision foundation models (VFMs). These present an alternative approach to training robot policies: instead of training a single agent on a diverse set of robotics tasks, we can take a readily-available powerful vision model and fine-tune it on each task separately. This comparison also demonstrates the utility of robotics data in the case of RoboCat, versus vision datasets for the VFM baselines, when adapting to robotics tasks.

We trained and evaluated 59 different VFM baselines (see Appendix G.3 for the complete list) on a subset of tasks in simulation and selected the best two as the main baselines for these experiments: the 438M parameter NFNet-f6 model (Brock et al., 2021) and the 197M parameter Swin-L model (Liu et al., 2021), both pretrained with CLIP (Radford et al., 2021). These models are smaller in size than the main RoboCat models because (i) they only need to deal with a single task (versus hundreds); and (ii) they were obtained by fine-tuning existing VFM architectures, limiting flexibility in size. For each comparison, the VFM models are trained with the same behavioural cloning loss and the same successful episodes that the RoboCat model uses for a given task variant.

We also utilise other baselines for a subset of the tasks. To isolate the impact of a diverse, robotics dataset, we use a Gato baseline (Reed et al., 2022), which employs a similar transformer architecture but with the majority of data from diverse non-robotics domains. We compare RoboCat with Gato on the robotics tasks used in their work, namely the RGB-Stacking Benchmark (Lee et al., 2021), and fine-tuning to blue-on-green stacking (Reed et al., 2022). For the former, we also use the BC-IMP specialist agents from (Lee et al., 2021).

## 5 Experiments

The evaluations and comparisons we present in this section investigate the following questions:

1. Can RoboCat learn from heterogeneous data and solve a large set of tasks, specified with visual goals and requiring dexterity on multiple physical and simulated embodiments? (Section 5.1)

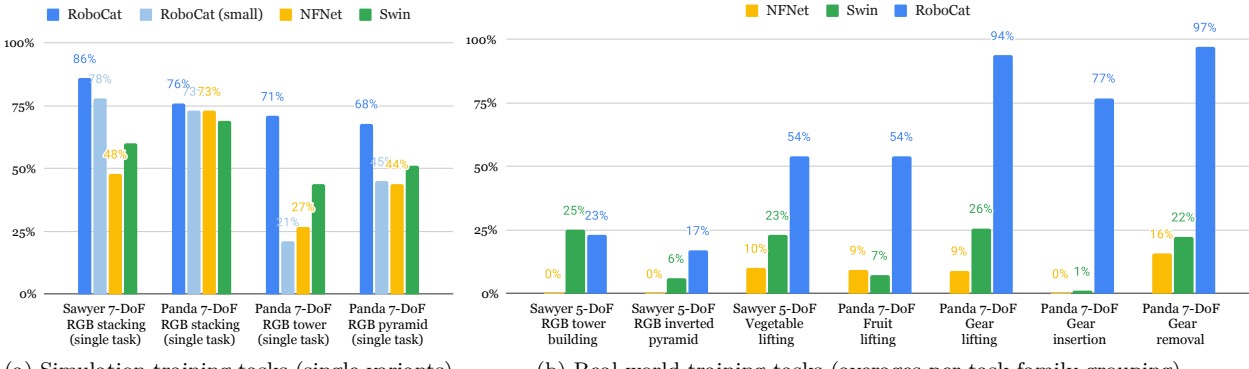

(a) Simulation training tasks (single variants)    (b) Real-world training tasks (averages per task family grouping)

Figure 5: **RoboCat compared to VFM baselines on training tasks.** RoboCat performs better on the vast majority of training tasks, compared to single-task baseline agents trained on the same data for each task. We compare here with the best-performing visual foundation model-based agents, chosen out of 59 strong baselines (see Section 4.4 for details). In the real world, where we have limited data compared to simulation, RoboCat can take advantage of multi-task joint training on robotics data to perform significantly better than the baselines. A much smaller version of the generalist is also evaluated on the sim tasks, and demonstrates similar performance for stacking but much lower success for the more challenging cases.

2. Can RoboCat adapt, with a small number of demonstrations, to challenging new scenarios such as unseen tasks, new objects, and new embodiments with unseen morphology and action spaces? (Sections 5.1, 5.2, and 5.3)

3. Does RoboCat exhibit cross-task transfer and generalisation to held-out tasks? (Section 5.2)

4. Can RoboCat self-improve by autonomously collecting data and integrating that new data into the next RoboCat iteration? (Section 5.3)

## 5.1 Overall RoboCat performance

We evaluated RoboCat over all the training tasks and we report task success rates averaged within each embodiment, task family, and object set, in Table 1 (see Appendix G.1 for per-task success rates). The tasks are broadly categorised into training (which include the tasks from the self-improvement process) and fine-tuning tasks. The RoboCat generalist agent was trained on all of these training tasks and then evaluated on a total of 141 training task variations. We demonstrate that a *single* RoboCat agent is able to perform all of these tasks, which involve multiple embodiments—in simulation and the real world—and multiple task families and objects sets.

For the fine-tuning tasks, the RoboCat generalist agent was fine-tuned to individual task variations and then each fine-tuned agent was evaluated on its respective task. We fine-tuned on either 500 or 1000 demonstrations and report results for both cases also in Table 1. RoboCat is able to fine-tune to tasks that not only include previously unseen task families (e.g. fruit insertion into a bowl), but also new object sets (e.g. shape-matching set) and a previously unseen embodiment (real KUKA 14-DoF robot).

In Figure 5, we compare RoboCat to visual foundation model (VFM) baselines trained on each task independently. In simulation, we only ran these baselines for one task from each task family due to the large number of task variations in simulation, whereas for the real-world tasks, we ran them on all of the task variations. The simulation results in Figure 5(a) show that the VFM agents are competitive on the Panda stacking task, but are outperformed by RoboCat on the other simulated building tasks. As shown in Figure 5(b),

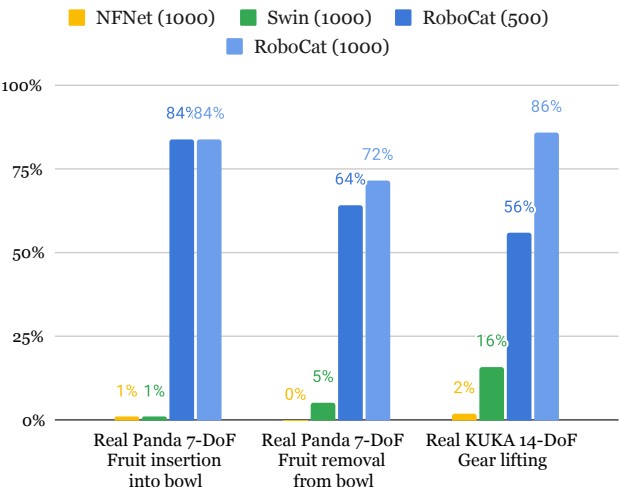

Figure 6: **RoboCat fine-tuning compared to VFM baselines.** RoboCat efficiently adapts to each of these previously unseen tasks which include unseen object sets and a new 14-DoF embodiment, whereas the visual foundation model-based baselines agents perform very poorly. The number of fine-tuning episodes are shown in parentheses for each method.

| Method | T1 | T2 | T3 | T4 | T5 | Average |
|--------|-----|-----|-----|-----|-----|---------|
| RoboCat | **87%** | **70%** | **82%** | 93% | 68% | **80%** |
| BC-IMP | 75% | 61% | 74% | 95% | 88% | 79% |
| Gato | 58% | 59% | 81% | **96%** | **96%** | 78% |

Table 2: **RGB Stacking Mastery Benchmark.** RoboCat performs, on average, similarly to prior works BC-IMP (Lee et al., 2021) and Gato (Reed et al., 2022) on this stacking benchmark, despite also being able to solve many other manipulation tasks. All three methods were evaluated on the same Sawyer robots with identical conditions, evaluation protocol, and successful episodes visually counted.

this is even more apparent in the real-world lifting, insertion, and removal tasks, where the VFM baselines are significantly outperformed by RoboCat. We also evaluate a much smaller (364M) RoboCat generalist agent on the simulation tasks; this model is slightly smaller than the NFNet baseline but still has to jointly learn all training tasks. We see from Figure 5(a) that the performance of this smaller model is comparable to RoboCat on the stacking tasks, but significantly lower for the harder cases: the success rate at least matches single-task baselines for pyramid building, but is poorer for tower building. Thus, the generalist does require sufficient capacity to perform well in the multi-task regime, at least for the harder sim tasks. Indeed, the full 1.18B RoboCat model can outperform the single-task baselines with only 3-6 times the capacity, despite being trained on 250 tasks.

For fine-tuning experiments, where only up to 1000 demonstrations are available per task, we compare fine-tuned RoboCat agents to VFM agents that are trained with only 1000 demonstrations. The results in Figure 6 show that the VFM baselines perform very poorly whereas the fine-tuned RoboCat agents perform well even when only using 500 demonstrations. Since the VFM agents are trained for single tasks, they are unable to leverage the large amounts of existing training data as done by RoboCat.

Lastly, in Table 2, we compare to previously reported performance on the real Sawyer 5-DoF stacking tasks, which are part of the RGB-Stacking Benchmark (Lee et al., 2021). This allows us to compare RoboCat performance on these tasks with vision-based BC-IMP specialists (Lee et al., 2021), as well as the Gato generalist (Reed et al., 2022). The latter allows us to compare the benefit of training on diverse robotic ma-

| Embodiment | Task Family | Object Sets | Training Task Variations | Held-out Task Variations |
|---|---|---|---|---|
| Simulation | Sawyer 7-DoF | Stacking | RGB objects, NIST-i gears | 23 | 10 |
| | Panda 7-DoF | Stacking | RGB objects, NIST-i gears | 30 | 6 |
| | | Tower building | RGB objects, NIST-i gears | 11 | 0 |
| | | Pyramid building | RGB objects | 30 | 0 |
| Real World | Sawyer 5-DoF | Stacking | RGB objects | 4 | 1 |

(a) Training tasks used by RoboCat-lim, with specific objects and task variations explicitly held out

| Generalisation Axis | Embodiment | Task Family | Object Set | Evaluation Task Variations |
|---|---|---|---|---|
| **Perceptual variation** (Stacking blue on green) | Sim Sawyer 7-DoF Sim Panda 7-DoF | Stacking | RGB, NIST-i | 6 6 |
| **Objects** (Sawyer Stacking triplet 5) | Sim Sawyer 7-DoF Real Sawyer 5-DoF | Stacking | RGB triplet 5 | 5 1 |
| **Behaviour source** (Demonstration data) | Real Sawyer 5-DoF | Stacking | RGB triplet 1 | 1 |
| **Sim-to-real** (Tasks seen in simulation) | Real Panda 7-DoF | Stacking Tower building | RGB triplet 5 RGB triplet 5 | 1 1 |
| **Task family** | Real Sawyer 5-DoF | Inverted pyramid building | RGB custom triplet | 1 |
| **Embodiment** | Real KUKA 14-DoF | Lifting | NIST-i large gear | 1 |

(b) Unseen fine-tuning tasks used by RoboCat-lim, grouped by generalisation axis

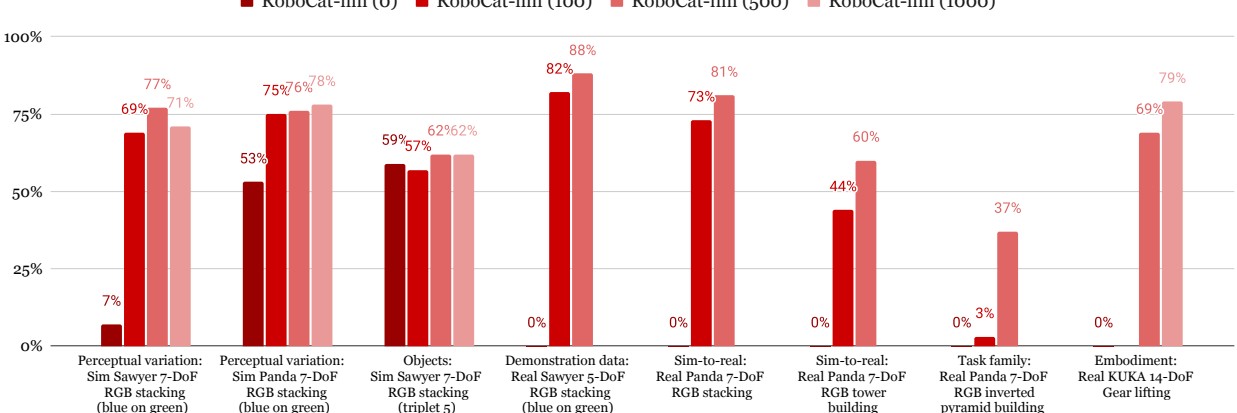

(c) RoboCat-lim 0-shot and k-shot fine-tuning performance by generalisation axis

Figure 7: **Generalisation and adaptation study for RoboCat-lim.** RoboCat-lim can be effectively fine-tuned, given a limited number of demonstrations, to tasks that are novel in terms of objects or task variants, and even to a completely new robot embodiment.

nipulation data rather than training on tasks from vastly different domains such as Atari or VQA. Although the relative success rates vary per object triplet, RoboCat is comparable to prior methods on average on this benchmark, despite being able to solve many other manipulation tasks.

We demonstrate that RoboCat, by using visual goals, is able to learn from heterogeneous data and perform a large set of tasks on multiple embodiments, and can quickly adapt—using a limited number of demonstrations—to unseen tasks, new object sets, and new embodiments.

## 5.2 Generalisation and adaptation

In order to analyse how RoboCat agents generalise and adapt, we trained a separate model of the same size but on only a subset of structure-building tasks (stacking, tower, and pyramid) with specific objects and task variations explicitly held out. The training and held-out tasks are listed in Figure 7(a). We refer to this limited-dataset model as *RoboCat-lim*. This enables us to investigate generalisation and adaptation of

| Data Source | Task Success | |
| --- | --- | --- |
| | 100 episodes | 500 episodes |
| Expert agent data | 63% | 84% |
| Demonstration data | 82% | 88% |

Table 3: **RoboCat-lim fine-tuning using different sources of data.** Despite RoboCat-lim only being trained on agent data originally, the model can be fine-tuned with either agent or human demonstration data. The 0-shot success rate for this task is 0%. This task is the held-out real-world perceptual variant of blue-on-green stacking.

this agent along specific axes (see Figure 7(b)). Furthermore, since the training tasks for RoboCat-lim are a subset of those used by the final RoboCat model, we can evaluate the effect of training on more tasks.

First, we measure how RoboCat-lim generalises to the 23 held-out tasks, both 0-shot and with k-shot finetuning (Figure 7(c)). In simulation, RoboCat-lim generalises 0-shot to a held-out object set on the Sawyer (third plot from the left) and the blue-on-green stacking task variant on the Panda (second plot), but does not generalise to that same task variant on the Sawyer embodiment (first plot). However, the model is effective at fine-tuning to this task variant with as little as 100 demonstrations. On the real-world blue-on-green stacking variant (fourth plot), RoboCat-lim achieves 88% when fine-tuning on 500 demonstrations, compared to the 60% success reported for Gato on the same data[2]. The remaining cases show RoboCat's ability to adapt to real-world variants of previously seen simulation tasks (both stacking and tower building), the challenging inverted pyramid building task family (for which even teleoperator success is only 52%), and to the real-world dexterous KUKA embodiment with nearly 80% success.

Overall, we show that RoboCat-lim adapts with only 100–500 episodes to a broad set of downstream tasks, including unseen variations and objects, different data sources (agent vs demonstrations; see Table 3), and an unseen task on an entirely unseen embodiment with twice as many degrees of freedom than seen in training. In addition, the results demonstrate the importance and potential of multi-embodiment training. The zero-shot performance on the unseen KUKA embodiment is zero (given it has entirely different action and observation spaces), but fine-tuning to a relatively small amount of data from this embodiment yields 69% success. The sim Panda and sim Sawyer also have different proprioception observations, but given that some Sawyer stacking data is used during RoboCat-lim training, the agent can generalise zero-shot to held-out Sawyer tasks that have been trained only for the Panda (object triplet 5).

We also compare RoboCat-lim to the VFM-based agents for few-shot fine-tuning on a couple of individual tasks in simulation. The results in Figure 8 show that our model can learn the tasks with significantly less data than the baselines.

Finally, we measure how much RoboCat benefits from its diverse training set, which includes all the tasks used for RoboCat-lim, the simulated structure building tasks held out from RoboCat-lim, all of the additional real-world data for the self-improvement tasks, and both simulated and real-world NIST-i gears tasks (lifting, insertion, and removal). In Figure 9(a), we compare RoboCat with RoboCat-lim specifically on the tasks that the limited model was trained on. Rather than its performance being negatively impacted due to the additional training tasks, RoboCat exhibits positive transfer across its training tasks and outperforms the more specialised RoboCat-lim. This trend of positive transfer also holds when adapting to new real-world

---

[2] The Gato model was fine-tuned with additional simulation episodes of the task, but was not originally trained with the object set in this task.

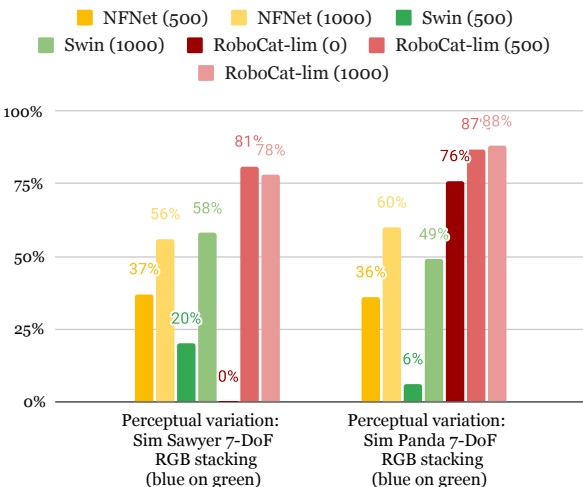

Figure 8: **RoboCat-lim 0-shot and k-shot fine-tuning compared to VFM baselines.** RoboCat-lim performs better than the baselines given the same number of episodes on a new task, even for a task in which RoboCat-lim gets 0-shot 0% success. This shows that the model can quickly adapt by reusing information from the tasks and embodiments seen during training. The number of fine-tuning episodes are shown in parentheses for each method. The results here are for single task variants, unlike the results in Figure 7(c).

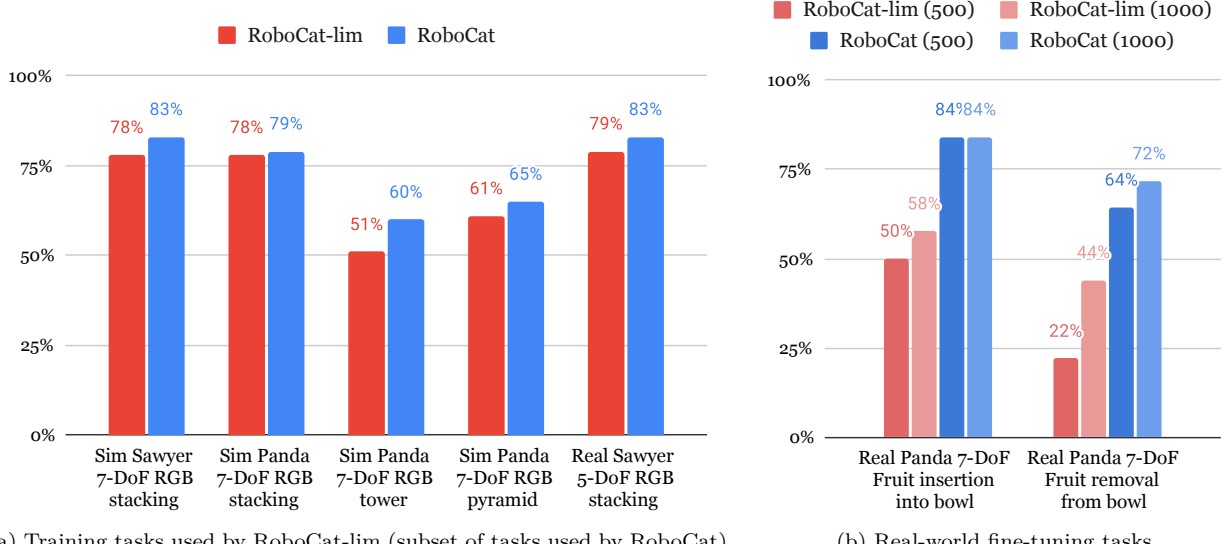

(a) Training tasks used by RoboCat-lim (subset of tasks used by RoboCat)  (b) Real-world fine-tuning tasks

Figure 9: **Positive transfer across tasks: RoboCat-lim vs RoboCat.** Training on more tasks (RoboCat) improves performance on the *limited training tasks* compared to only training on these limited tasks (RoboCat-lim). In addition, RoboCat is better when fine-tuning to the insertion and removal tasks. The reported numbers are averages of task variants within each grouping.

tasks, e.g. as RoboCat was trained on real-world fruit and vegetable lifting data, it adapts better to the insertion and removal tasks with the fruits and bowl (Figure 9(b)).

## 5.3 Self-improvement via RoboCat fine-tuning and self-generation of data

In this section, we demonstrate the key ability of RoboCat to perform self-improvement. That is, to fine-tune to a new task with a limited number of demonstrations, self-generate a larger amount of experience, and

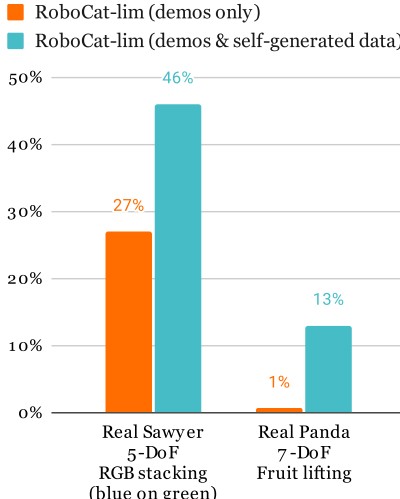

Figure 10: RoboCat-lim trained with additional demonstrations vs with additional demonstrations and self-generated data.

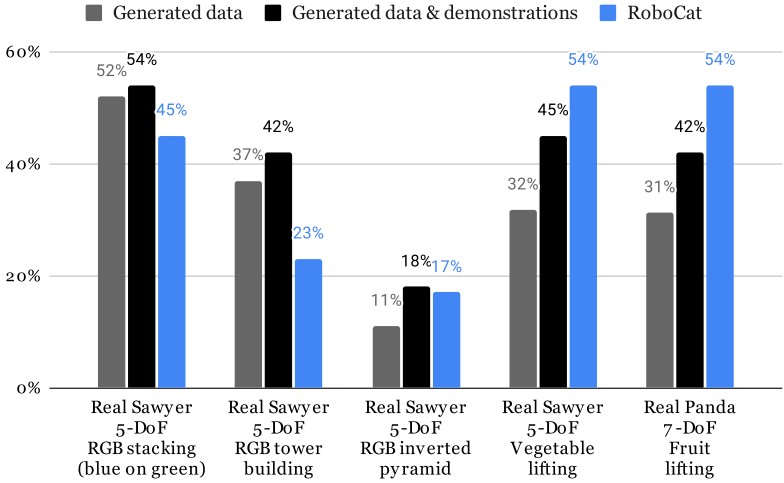

Figure 11: RoboCat compared with the performance of the data-generating agents or the combined performance of these and the demonstrations, the latter of which is used for training RoboCat.

retrain a more capable generalist with this additional data. This represents a first step towards a foundation agent which can iteratively learn new tasks.

To perform self-improvement, we fine-tuned older RoboCat-lim equivalent models to a number of unseen real-world tasks using human-teleoperated data. These included the real blue-on-green stacking and inverted pyramid building task already mentioned in Section 5.2, as well as a tower-building task and 8 vegetable and fruit lifting tasks). We then used these policies to generate large amounts of data autonomously. All of this data was part of the dataset used to train the main generalist shown in Section 5.1.

We first perform a smaller experiment with a subset of the tasks, to provide a proof-of-concept of self-improvement, and carefully isolate and evaluate the contribution of self-generated data alone. We train a smaller 364M model on the structure-building tasks (i.e. those used for RoboCat-lim) and 500 demonstrations from only a few self-improvement tasks: fruit lifting (apple, banana, and peach) and blue-on-green Sawyer stacking. This represents a baseline of directly incorporating the few available demonstrations into the training data for the generalist. We also train a 364M "self-improved" model that additionally sees the self-generated data for these tasks. The results in Figure 10 show that the self-improved agent outperforms the baseline agent in all four of these tasks. In other words, given demonstrations of a task, the self-improvement procedure (fine-tuning and self-generating additional data) is significantly better than using the demonstrations directly in training the generalist.

Next, we demonstrate self-improvement at scale: we incorporate self-generated data from numerous task-specific RoboCat-lim fine-tuned agents to yield a stronger generalist. This is the process by which we obtained the main RoboCat generalist presented in Section 5.1.

Figure 11 shows the performance of these self-data-generating agents, compared with the performance of the full RoboCat generalist. For most cases, the RoboCat generalist performance is similar to or even better than that of the agents generating the data. These results highlight the potential for RoboCat to self-improve

and grow its multi-task capabilities, as we have also seen from other experiments. By self-generating data and incorporating additional data from a diverse set of tasks, the resulting agent has better generalisation and fine-tuning capabilities on a broader set of real-world tasks.

### 5.4 Further ablations

We report a number of additional ablations and evaluations in the appendix. These include ablating the choices for VQ-GAN tokeniser and observation prediction (Appendix D.3), a comparison of different RoboCat model sizes (Appendix G.2), evaluations over many different vision model baselines (Appendix G.3), and ablations of the NIST-i environment (Appendix G.4).

## 6 Related Work

### 6.1 Transformers for decision making

Transformers (Vaswani et al., 2017) have shown impressive results at scale in domains like vision (Dosovitskiy et al., 2020; He et al., 2022), language (Vaswani et al., 2017; Devlin et al., 2018; Brown et al., 2020) and speech (Radford et al., 2022), and can be fine-tuned to many different downstream tasks and modalities (Lu et al., 2021). Inspired by these successes, earlier efforts to leverage transformers for decision making focused on improving their training stability for RL (Parisotto et al., 2020), using self-attention for improving relational reasoning (Zambaldi et al., 2018), one-shot imitation learning (Dasari and Gupta, 2021), fast adaptation to novel tasks (Ritter et al., 2020), on the fly adaptation in 3D environments (Team et al., 2023), 3D reasoning (Shridhar et al., 2023), and multi-embodiment continuous control (Kurin et al., 2020; Gupta et al., 2022). However, these works leverage the transformer architecture within the framework of standard RL and imitation algorithms. Recently, generative pretraining for sequence modeling has been extended to offline RL (Janner et al., 2021; Chen et al., 2021), where a transformer model is trained to autoregressively maximise the likelihood of trajectories in the offline dataset for specialist agents with low-dimensional states. Building on this, Reed et al. (2022); Lee et al. (2022); Jiang et al. (2022) train multi-task generalist agents with high-dimensional image observations. VIMA (Jiang et al., 2022) shows the power of multi-modal prompting for accomplishing many tasks with a single model. Unlike our work, VIMA uses high level observation and action spaces, by assuming accurate object detection and pre-defined action primitives. Our work is closely related to Gato (Reed et al., 2022) but differs in the variety and scale of robotic tasks *mastered* by a single agent. We show positive transfer between tasks and fast adaptations to many real-world robot tasks.

### 6.2 Visual pretraining for control

The use of pretrained visual representations presents a promising approach for efficient robot policy learning, requiring less robot-specific data. Early efforts focused on using supervised pretraining for navigation (Zhou et al., 2019; Chen et al., 2020a; Sax et al., 2018) and manipulation (Zhou et al., 2019; Chen et al., 2020a; Yen-Chen et al., 2020) domains. Building on the progress in self-supervised representation learning, multiple recent works have shown that frozen visual encoders, trained through self-supervision on internet-scale datasets, can enable sample-efficient behaviour cloning (Nair et al., 2022; Parisi et al., 2022; Radosavovic et al., 2023; Majumdar et al., 2023; Sharma et al., 2023), on-policy reinforcement learning (Xiao et al.,

2022; Khandelwal et al., 2022; Majumdar et al., 2023). Robot-agnostic visual dynamics models also show skill transfer between robots when deployed with a visual model-predictive control (MPC) policy (Hu et al., 2022). Our work differs in that we directly learn the action prediction for all embodiments jointly rather than using video prediction for planning. In this work we use a frozen pretrained VQ-GAN (van den Oord et al., 2017; Esser et al., 2021) trained on a diverse collection of images to speed up training time significantly, and combine the VQ-GAN tokens with future frame prediction (Gupta et al., 2023) for sample-efficient transfer learning. Concurrently, Kotar et al. (2023) also find similar generalisation benefits of using the combination of VQ-GAN tokens and future frame prediction during policy learning for the navigation domain.

### 6.3 Goal-conditioned policies

Goal-conditioned agents have long been of interest in policy learning (Kaelbling, 1993; Schaul et al., 2015). Hindsight goal relabelling is a popular method for annotating arbitrary trajectories with goals (Andrychowicz et al., 2017). Learning from visual goals is challenging as images contain a lot of information that may be unrelated to the desired goal-conditioned behaviour, such as lighting or positions of distractors (Pinto et al., 2018). As we are primarily concerned with goal images as task specification in a behaviour cloning setting, this work does not address the question of goal distance, goal generation, or exploration. Unlike Nair et al. (2018), we assume a dataset of goal images is available during evaluation and data collection, as we only deploy our goal-conditioned agent for data collections on tasks for which we had teleoperated episodes to learn from. Davchev et al. (2022) also utilised a dataset of goals, bootstrapped from demonstrations. However, they do not work with images. Similar to RoboCat, Groth et al. (2021) also instruct a behaviour-cloned policy with goal images but rely on explicit inductive biases in the network architecture to infer the task. Ghosh et al. (2019) propose iterated goal-conditioned learning as a form of reinforcement learning, which is similar to our self-improvement step.

### 6.4 Generalist robotic agents

Recent works have looked at the problem of training generalist robot agents. RT-1 takes language instructions to perform a variety of object manipulation tasks (Brohan et al., 2022). While RT-1 trains on data from two different robots, they have the same action specification. PaLM-E demonstrates that large visual-question-answering can serve as planners for robotics tasks. Rather than directly controlling different robots, PaLM-E outputs language instructions (such as "Pick the green rice chip bag from the drawer.") to pretrained lower-level controllers (Driess et al., 2023). Dasari et al. (2020) introduce a large-scale dataset of robotic interactions produced by pre-trained random policies acting on a range of pick-and-place-based manipulation tasks. They show that learned robotic agents with shared observation and action space can operate across a range of environments and hardware, and also demonstrate fine-tuning capabilities.

In this work, we look to solve tasks in both simulation and the real-world, covering a wide set of behaviours and affordances, incorporating precision and dexterity, and embracing high-dimensional low-level control over multiple simulated and real embodiments. To our knowledge, RoboCat is the first work to natively support multiple real-world robotic embodiments with different observation and action specifications. We also demonstrate the ability to self-improve by fine-tuning to new tasks and self-generating data for use in retraining—a unique capability over all of the methods we surveyed. Finally, we focus on visual goal-

conditioning in this work, but could also enable more flexible task specification in the future, such as language conditioning or full demonstrations; this is already facilitated by some of the other methods.

## 7 Summary and Future Work

In this report, we have presented RoboCat, a generalist agent capable of solving a large and diverse set of tasks specified via goal images; across different task families, embodiments, control modes, and objects; in both simulation and the real world, and from different sources of data. RoboCat is additionally able to quickly adapt, via fine-tuning on 100–1000 demonstrations, to a wide set of downstream tasks and across many different axes of generalisation. More importantly, we can use such adapted agents to generate data that can be added to RoboCat's training dataset for future iterations of the agent, a process we call self-improvement. We have thoroughly investigated our agent's capabilities both in simulation and the real world with tens of thousands of real evaluations on 36 real robots of 3 different types. We have shown that the cost of acquisition of new skills is dramatically lower compared to single-task baselines, even when those are based on visual foundation models. Finally, we have observed that by scaling and diversifying its training data we get a RoboCat agent that performs better on training tasks and adapts better to unseen ones. Throughout our experiments, we demonstrate that RoboCat can be adapted to a broad set of unseen downstream tasks (13 with the final agent, 22 more during a thorough generalisation study, and a further 9 tasks in the self-improvement process). These tasks include unseen embodiments (KUKA with a dexterous hand), new task families (eg. lifting, insertion/removal, inverted pyramid building), held-out perceptual variations, real versions of previously-seen sim tasks, and many unseen objects (eg. printed fruits and vegetables, shape-matching objects).

Future work could look into enabling flexible and multi-modal task specification. Incorporating relevant existing, freely-available datasets with language annotations would be a first good step. Task specification via language offers complementary benefits to visual goals, and different tasks may be better specified by either modality. In addition, while this work focused on visual goal-conditioning and VFM baselines, which may be able to reason well over images; language-conditioning and LLM/VLM baselines may offer better temporal reasoning capabilities.

Another research avenue could explore improving both training and fine-tuning capabilities of such a model with reinforcement learning (RL), since RoboCat in its current form only employs behaviour cloning. While visual goal specification already allows the agent to learn from failures and sub-optimal data, incorporating RL would enable both learning with rewards and learning online with real-world interaction. Finally, while RoboCat aims to tackle behavioural diversity in manipulation tasks, the different embodiments are all in a controlled lab setting with visually-similar backgrounds. We hope that next-generation foundation agents will demonstrate robustness to different basket textures and operate in more visually-diverse environments in the wild.

## Broader Impact

This work presents progress on training generalist agents for robotic manipulation. Our work presents a recipe, and first steps, in an emerging area, with experiments in a controlled lab environment demonstrating promising but imperfect performance. Nonetheless, the potential impact on society from generalist robotic agents calls for increased interdisciplinary research into their risks and benefits. Thus, we discuss the broader impact of this line of research, beyond the specific contributions of this paper. To provide an easily accessible reference for RoboCat's intended use-case and potential shortcomings we include a model card in Appendix A. We emphasise that the model is for research use only and not currently deployed in any production scenario to any users, and thus expect no immediate societal impact.

In general, RoboCat inherits many of the safety concerns discussed in Gato (Reed et al., 2022); on which it is based. In addition, since RoboCat takes actions in the physical world—and on multiple embodiments—it may pose new challenges with respect to safety. For example, physical embodiments and imitation from human data can cause humans to anthropomorphise the agent; leading to a potentially misplaced trust and underappreciation for inherent dangers that come from interacting with robots[3]. Additionally, cross-embodiment transfer from one robot to another can lead to undesired movements (such as high gain motor actuation). Considerations with respect to general AGI safety (Bostrom, 2014) may also require updating when considering agents with multiple embodiments.

We consider that value alignment (Russell, 2019) with human preferences (as e.g. expressed via reward labelling in this work) is crucial for a safe evolution of this technology. While our reward labelling process to determine successful and desired behaviours is a starting point for this, future work should consider adapting alignment techniques successfully used for language models to our setting (Ouyang et al., 2022; Kenton et al., 2021; Bai et al., 2022).

Finally, the self-improvement loop we designed for RoboCat allows us to improve the model over time by retraining on data collected from deploying a previous version to our robots. Such a self-improvement loop poses additional challenges with respect to AGI safety since it, partially, implements a reinforcement learning loop; which comes with its own safety concerns (see e.g. Omohundro (2008); Turner et al. (2021)). While further work is needed into AGI safety for reinforcement learning robotic systems, it is important to note that unlike in a reinforcement learning scenario, the self-improvement capabilities of RoboCat are *not* autonomous and *no learning* takes place while interacting with the physical world. That is, data collection is started and stopped by humans and uses frozen versions of RoboCat. Learning an improved version is implemented as a supervised learning problem from a fixed data source and is entirely decoupled from data collection.

## Acknowledgements

We would like to thank Jackie Kay for contributions to the VQ-GAN codebase; Federico Casarini for help with the robotic lab operations; Markus Wulfmeier for initial exploration into alternative fine-tuning methods; Nando de Freitas for general advice; Yixin Lin, Vincent Vanhoucke, Shakir Mohamed, and Michael Neunert for paper feedback; and Jonathan Hutchinson for the graphics of Figure 1.

---

[3] We note that we utilise a force-torque compliant controller with built in safety mechanisms.

## Author Contributions

**RoboCat generalist training**

Konstantinos Bousmalis, Giulia Vezzani, Coline Devin, Dushyant Rao, Alex X. Lee, Maria Bauza, Todor Davchev, Yuxiang Zhou, Agrim Gupta

**RoboCat fine-tuning**

Giulia Vezzani, Dushyant Rao, Alex X. Lee, Coline Devin, Maria Bauza, Todor Davchev, Valentin Dalibard, Martina Zambelli, Agrim Gupta

**Core infrastructure for experiments at scale**

Michiel Blokzijl, Claudio Fantacci, Akhil Raju, Antoine Laurens, Dave Barker

**Data and tasks**

*KUKA:* Murilo F. Martins, Martina Zambelli, Rugile Pevceviciute, Antoine Laurens, José Enrique Chen
*NIST-i:* Todor Davchev, Maria Bauza, Akhil Raju, Jost Tobias Springenberg, Jon Scholz, Misha Denil, Oleg Sushkov, Jean-Baptiste Regli, Tom Rothörl
*RGB:* Giulia Vezzani, Konstantinos Bousmalis, Dushyant Rao, Coline Devin, Alex X. Lee, Thomas Lampe, Abbas Abdolmaleki, Francesco Nori, Antoine Laurens
*Non-NIST-i insertion/removal:* Akhil Raju, Antoine Laurens, Alex X. Lee

**Evaluation infrastructure: success detectors, no-reset evaluation, annotations**

Akhil Raju, Claudio Fantacci, Misha Denil, Michiel Blokzijl, Todor Davchev, Thomas Lampe, Dave Barker, Maria Bauza, Alex X. Lee, Jon Scholz, Tom Rothörl

**VQ-GAN tokenisation**

Agrim Gupta, Coline Devin, Scott Reed

**Gato architecture and infrastructure**

Emilio Parisotto, Konrad Żołna, Scott Reed, Sergio Gómez Colmenarejo, Jost Tobias Springenberg, Oliver Groth

**Single-task VFM baselines**

Yuxiang Zhou, Todor Davchev, Alex X. Lee

**Teleoperated data collection**

Akhil Raju, Antoine Laurens, Michiel Blokzijl, Misha Denil, Nathan Batchelor, Claudio Fantacci, Joy Ortiz

**Paper and blog post content**

Coline Devin, Alex X. Lee, Dushyant Rao, Konstantinos Bousmalis, Giulia Vezzani, Todor Davchev, Maria Bauza, Agrim Gupta, Akhil Raju, Antoine Laurens, Jost Tobias Springenberg, Misha Denil, Nicolas Heess

**Project leadership and coordination**

Konstantinos Bousmalis, Giulia Vezzani, Joy Ortiz

**Advisors**

Nicolas Heess, Francesco Nori, Raia Hadsell, Jost Tobias Springenberg, Martin Riedmiller, Yusuf Aytar

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

# Appendix

| Model details | |
|---|---|
| Organisation | Google DeepMind |
| Model date | June 2023 |
| Model type | Transformer with VQ-GAN encoder for multi-task, multi-embodiment behaviour cloning from human, agent and self-generated data. |
| Model version | Initial release. |
| Feedback on the model | konstantinos@google.com, giuliavezzani@google.com |
| **Intended uses** | |
| Primary intended uses | Research into learning to accomplish a wide variety of tasks from expert demonstrations or multiple real robot embodiments for manipulation. |
| Primary intended users | Google DeepMind Researchers. |
| Out-of-scope uses | Not intended for commercial or production use. Military uses are strictly prohibited. |
| **Factors** | |
| Relevant factors | Salient factors that may alter model performance are: agent embodiment in control data, training data token amount and diversity, performance of experts in training data and goal conditioning. Quality of policy used for self-generated data collection. Quality of the VQ-GAN encoder. Zero-shot evaluation on held-out robots. |
| Evaluation factors | Reported factors are: number of input tokens, importance of different tokenisation schemes, agent performance. |
| **Metrics** | |
| Model performance measures | We chose to report success at the task (measured as having solved the task at the end of an episode) in an episodic evaluation setting. Held-out tasks are used to assess generalisation, ablations show importance of different components. |
| Decision thresholds | N/A |
| Approaches to uncertainty and variability | The reported values do not take into consideration model uncertainty as they are evaluations of a single model and its ablations. It is prohibitive for us to evaluate models from multiple training runs as this would involve constantly training and evaluating robots. We account for noise in the evaluation by averaging success across multiple episodes. |
| **Evaluation** | |
| Tasks | RoboCat is evaluated on in-distribution and out-of-distribution tasks, on both simulated and real-world robot environments. See section 3 for details on our tasks. |
| **Training Data** | |
| Datasets | We use a diverse and large number of datasets for training RoboCat. These include data from agent experience, human demonstrations and self-generated data, on both simulated and real-world robot environments. See subsection 3.4 for details on our datasets. |
| Motivation | Create a multi-modal, multi-task, multi-embodiment generalist policy by collecting as much, diverse, data as possible. Joint training on all the datasets has produced a single network, RoboCat, capable of performing these tasks. |
| Pre-processing | The multi-modal training data is tokenised into a stream of discrete embeddings. See subsection 2.1. |
| **Quantitative Analyses** | |
| Unitary results | We present several evaluations of RoboCat on a variety of manipulation tasks. See section 5 for an analysis of the model capabilities and ablations. |
| **Ethical Considerations** | |
| RoboCat is an early research model that has not yet been evaluated for deployment and safety of use outside a pure research setting. | |
| **Caveats and Recommendation** | |
| Future work | The interaction of diverse training data domains and the different affordances faced in evaluation is poorly understood, and potential ethical and safety risks arise as the generalist's capabilities grow. |

Table 4: **RoboCat model card.**

## A  Model Card

We present a model card for RoboCat in Table 4.

## B  Tasks and Data

In this section, we provide an extensive description of the tasks and data RoboCat has been trained and fine-tuned on.

### B.1  Task families

In total, we consider 253 different task variations, each of which can have an infinite number of state configurations describing it (see Table 5) and can be grouped in different task families. Figure 12 collects examples of the goal images used for each task family.

#### B.1.1  Lifting objects

We include a handful of lifting tasks performed in the physical world and three in simulation. A lifting task is defined as grasping and lifting an object off the basket surface until the natural end of the episode. Our aim with this task family is to primarily study goal understanding and generalisation to new embodiments and tasks. This task family includes four variants for the YCB fruits (see Figure 12(n) for an example of goal image), vegetable objects (see Figure 12(m)) and seven variants for NIST-i gears. For the three NIST-i gears we include lifting in simulation (see Figure 12(h)) and in real with the Panda 7DoF (see Figure 12(o)). We also include lifting the large gear with the KUKA 14DoF (see Figure 12(v)).

#### B.1.2  Building structures

For the RGB-objects and NIST-i objects that are coloured in red, green, and blue[4], we use a set of structure-building tasks. These are difficult due to the non-cubic shapes of the objects: without orienting the objects correctly, they can easily slide off of each other.

- **Stacking:** A top and bottom object are specified and must be stacked stably. This task family has 6 variations for triplet. Although stacking cuboids is relatively easy, RGB objects stacking requires an agent to come up with strategies beyond simple pick-and-place. These strategies differ for each task variation, as well. We consider this task both in simulation (see Figure 12(a) – (d) for examples of goal images) and in the real-world (see Figure 12(j)).

- **Tower Building:** This requires two stacks to build a tower out of 3 objects. For this task family, we consider only a subset of all the variations we could have per triplet. Some variations have been particularly challenging to solve with reinforcement learning agents and therefore we did not collect training data for them. We consider this task in simulation (see Figure 12(e)) and, for one variant, also in the real-world (see Figure 12(k)).

- **Pyramid Building:** Two objects must be placed close to each other and a third on top of both. We consider this task family only in simulation (see Figure 12(g)).

---

[4] NIST-i gears are coloured according to their sizes: red, green, and blue for small, medium, and large, respectively.

| Embodiment | | Task Family | Object Set | Training Task Variations | Source of Data | Number of Training Episodes | Example Goal Image |
|---|---|---|---|---|---|---|---|
| Simulation | Sawyer 7-DoF | Stacking | RGB objects | 28 | RL | 1 022 304 | |
| | | Stacking | NIST-i gears | 5 | RL | 182 650 | |
| | Panda 7-DoF | Stacking | RGB objects | 30 | RL | 1 012 142 | |
| | | Stacking | NIST-i gears | 6 | RL | 129 664 | |
| | | Tower building | RGB objects | 8 | RL | 99 725 | |
| | | Tower building | NIST-i gears | 3 | RL | 42 805 | |
| | | Pyramid building | RGB objects | 30 | RL | 603 133 | |
| | | Lifting | NIST-i gears | 3 | Teleop | 421 638 | |
| | | Insertion-peg | NIST-i gears | 3 | Teleop | 285 690 | |
| Real World | Sawyer 5-DoF | Stacking | RGB objects | 93 | RL, Teleop, RoboCat | 51 553 | |
| | | Tower building | RGB objects | 1 | Teleop, RoboCat | 15 241 | |
| | | Inverted pyramid building | RGB objects | 1 | Teleop, RoboCat | 15 155 | |
| | | Lifting | YCB-i vegetables | 4 | Teleop, RoboCat | 55 325 | |
| | Panda 7-DoF | Lifting | YCB fruits | 16 | Teleop, RoboCat | 49 957 | |
| | | Lifting | NIST-i gears | 3 | Teleop | 25 566 | |
| | | Insertion-peg | NIST-i gears | 3 | Teleop | 21 933 | |
| | | Removal-peg | NIST-i gears | 3 | Teleop | 24 803 | |
| | | Insertion-bowl | YCB fruits and YCB-i bowl | 3 | Teleop | 1000 | |
| | | Removal-bowl | YCB fruits and YCB-i bowl | 3 | Teleop | 1000 | |
| | | Insertion-base | Shape-matching objects | 3 | Teleop | 1000 | |
| | | Removal-base | Shape-matching objects | 3 | Teleop | 1000 | |
| | KUKA 14-DoF | Lifting | NIST-i gears | 1 | Teleop | 1000 | |

Table 5: **Training and fine-tuning tasks used by the final RoboCat agent.** See Figure 12 for more images.

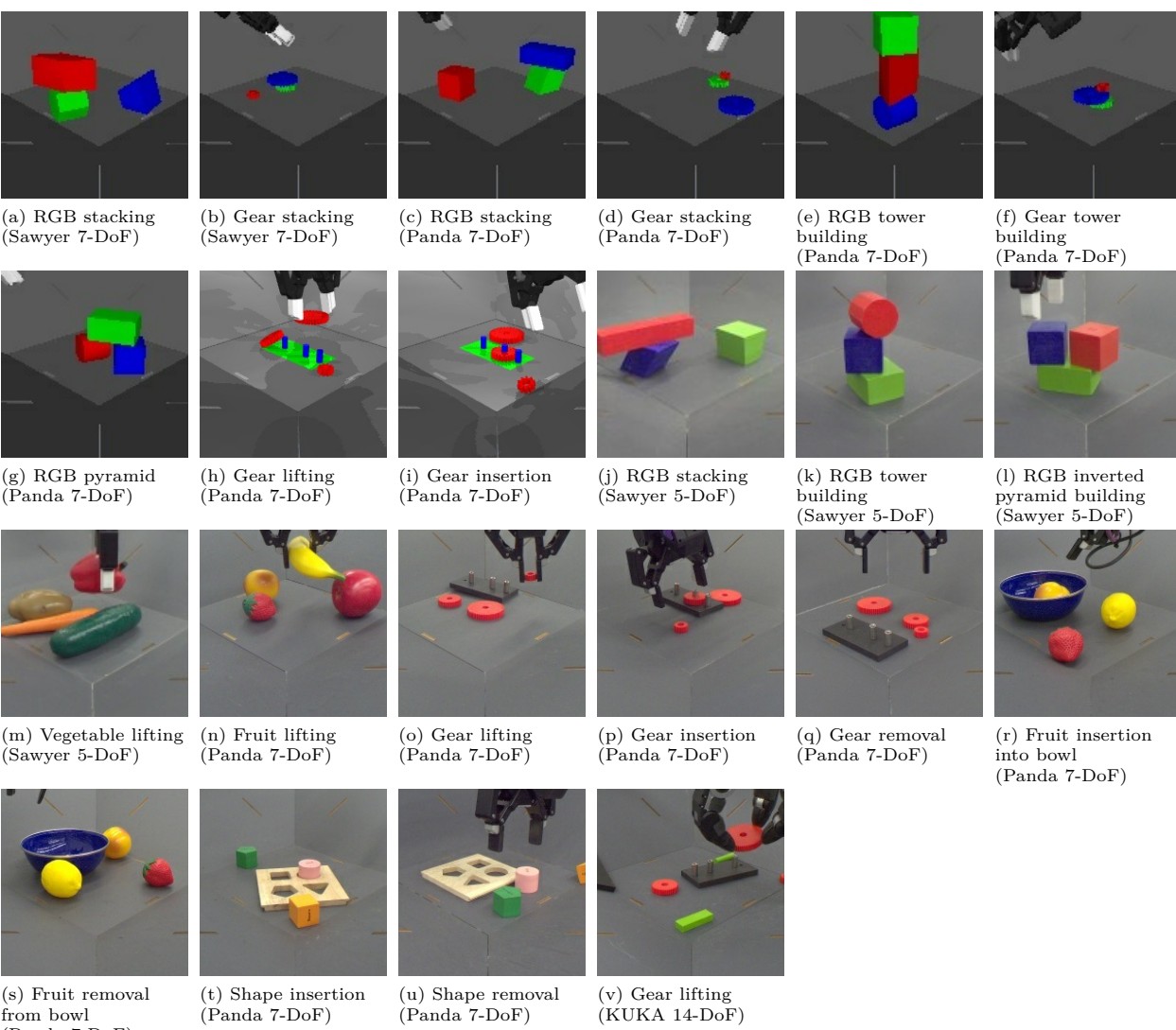

(a) RGB stacking
(Sawyer 7-DoF)

(b) Gear stacking
(Sawyer 7-DoF)

(c) RGB stacking
(Panda 7-DoF)

(d) Gear stacking
(Panda 7-DoF)

(e) RGB tower
building
(Panda 7-DoF)

(f) Gear tower
building
(Panda 7-DoF)

(g) RGB pyramid
(Panda 7-DoF)

(h) Gear lifting
(Panda 7-DoF)

(i) Gear insertion
(Panda 7-DoF)

(j) RGB stacking
(Sawyer 5-DoF)

(k) RGB tower
building
(Sawyer 5-DoF)

(l) RGB inverted
pyramid building
(Sawyer 5-DoF)

(m) Vegetable lifting
(Sawyer 5-DoF)

(n) Fruit lifting
(Panda 7-DoF)

(o) Gear lifting
(Panda 7-DoF)

(p) Gear insertion
(Panda 7-DoF)

(q) Gear removal
(Panda 7-DoF)

(r) Fruit insertion
into bowl
(Panda 7-DoF)

(s) Fruit removal
from bowl
(Panda 7-DoF)

(t) Shape insertion
(Panda 7-DoF)

(u) Shape removal
(Panda 7-DoF)

(v) Gear lifting
(KUKA 14-DoF)

Figure 12: **Example goal images.** These are larger versions of the goal images shown in Table 5 (in order) corresponding to each embodiment, task family, and object set combination. Images (a)–(i) correspond to tasks in simulation and images (j)–(v) correspond to tasks in the real world.

- **Inverted Pyramid Building:** This task family requires to place two objects on top of a third one. We consider only one task variation where the bottom object is wider while the other two objects can barely fit on top of the bottom object. The difficulty of this task comes from the fact that the robot needs to be very precise in placing the top objects such that stacking them becomes successful and stable. We only consider this task in the real world (see Figure 12(l)).

### B.1.3 Inserting and removing objects

We introduce pairs of insertion and removal tasks of quite different flavours using the YCB, NIST-i, and the Shape Matching Objects.

- **Inserting and removing fruit in/from a bowl:** For this task family we use the YCB strawberry, lemon, and peach, and the YCB-i bowl. When inserting the different fruits, the bowl can start either empty or with other fruits in it. This task family has 6 variants: inserting/removing {strawberry, lemon, peach} in/from the bowl (see Figure 12(r) and Figure 12(s) for examples of insertion and removal respectively).

- **Inserting and removing shapes in/from board:** This task requires a given object from the Shape Matching Objects to be inserted into the correct shape in the board. This task poses several challenges: 1) the agent has to understand the right shape for the desired object; 2) it has to reorient it, in order to properly fit into its shape; 3) the board can move, making the task even more challenging (see Figure 12(t) and Figure 12(u) for examples of insertion and removal respectively).

- **Inserting and removing NIST-i gears on/from shafts (NIST-i):** In this task, a specific gear must be precisely inserted in a particular shaft. This task poses several changes as the gears are hard to manipulate, the tolerance for the shafts is low and, in the real-world version of this task, the base of the shafts is not anchored and therefore it can freely move. See Figure 12(p) Figure 12(q) for examples of insertion and removal respectively in real, and Figure 12(i) for an example of insertion in simulation.

### B.1.4 NIST-i tasks

In this paragraph we provide some extra details on the NIST-i tasks.

**Comparison of NIST-i to NIST** The primary differences between our NIST-i task and the official NIST benchmark (Kimble et al., 2020) are three-fold: i) unlike the NIST board, our base is not fixed allowing it to freely move around the robot cell; ii) the NIST-i gears are red as opposed to white with a peg board that is black as opposed to metallic; and iii) the tolerance between the shafts and gear holes is smaller for the original NIST board. The insertion tolerance of the physical NIST-i gears is $1\,\text{mm}$ as opposed to the sub-millimeter tolerance for the actual NIST board which ranges between $0.029\,\text{mm}$ $0.005\,\text{mm}$. In the simulated NIST-i environment, we used a green board with blue pegs and a $1\,\text{mm}$ insertion tolerance. In the KUKA 14DoF environment the NIST-i object set also includes two green pegs and an additional base. These extra objects serve as distractions for the task of lifting the large gear.

**Comparison of simulated and physical tasks** In this work the simulated and physical version of the NIST-i tasks have different colours (see Figure 27). For instance, when stacking with NIST-i gears in simulation we found that teleoperation using only red gears was quite challenging for the human demonstrators mostly due to the reduced depth perception. The red-green-blue setup for stacking tasks is nicer for contrast and therefore for people to collect data. In addition to this, for insertion/removal tasks we chose to fix the base in simulation, primarily for simplicity. Early on this project, we used the simulated environment as a testing ground, so we wanted to simplify the problem. Specifically, by fixing the gear base we could guarantee two things: i) this makes the task easier for raters and for agents (see Appendix G.4.1); and ii) it also makes the simulation run much faster while having a freely moving base makes the sim slower and overall more complex.

## B.2 Data sources

This section provides further details on how RoboCat training data has been generated.

**RL-agent data** For all our structure building tasks in simulation we trained expert policies, per task variation, via off-policy RL with a shaped reward for each. We used the MPO algorithm (Abdolmaleki et al., 2018) for this purpose, but any RL algorithm could have been used. We found training directly from state to be significantly faster than training from vision and thus generated all our simulation structure building datasets from state-based agents. It is important to note that when training RoboCat on this data, we discard any privileged information, such as the object poses, that would not be readily available in our real-world tasks. The data for the real Sawyer 5-DoF stacking tasks was also generated by RL agents using simulation-to-reality transfer and offline RL, as discussed in Lee et al. (2021). Table 6 shows the number of successful and failed episodes obtained from RL agents for these tasks.

**Human teleoperation** The rest of the data were either collected by human teleoperation or by a RoboCat agent fine-tuned on such data.

Human teleoperators collected demonstrations by controlling the robot arm and gripper with a 3DConnexion SpaceMouse compact⁴ and a web-based UI. The web UI is similar to the one used by Abramson et al. (2021) and enabled us to work with participants from around the globe. We worked with over 100 participants from 4 different countries to collect over 4000 hours of human demonstrations across many of our tasks. Remote teleoperators could use the basket and wrist cameras to operate the arm while on-site teleoperators would sit next to the robots while controlling them. We noticed that on-site teleoperators achieve higher success rates than their remote counterparts, but the human baselines reported in this paper are aggregated across all locales.

During each episode, we showed a different language instruction to the teleoperator in the web UI, like "lift the apple". Some episodes were used for training or calculating human baselines, while other episodes helped ensure we had diverse starting configurations for the following episodes. Empirically, we found that these "shuffle" episodes made our agents more robust.

We also worked with the same pool of participants and used the same setup to collect data in simulation for human baselines.

---

⁴ https://3dconnexion.com/uk/spacemouse/

Table 7 reports the numbers of successful and failed human teleoperations collected for the NIST-i tasks, both in simulation and with the real robots. Table 8 shows the number of successful and failed human teleoperations collected on the real robot for the tasks used for self-improving RoboCat. Table 9 shows instead the number of successful episodes collected and used for fine-tuning our final RoboCat agent. Finally, Table 10 reports the teleoperators success rates for each real robot task to contextualize the quality of the data and the difficulty of teleoperating each task.

**Self-generated data** Once human teleoperation had generated a minimum number of successful episodes for a given task variation, we fine-tuned a RoboCat agent on 500 or 1000 successful episodes, as explained in subsubsection 2.1.2. We then used this specialised agent to gather, autonomously, more data for the same task variation. This self-generated RoboCat data, combined with all demonstrations that ended up being collected for that task variation, were used to train our final RoboCat agent. The total number of episodes collected via the self-generation process can be found in Table 8.

## C   Embodiments and Environments

In this section we describe the physical embodiments used, shown in Figure 2 in the paper, and the environments they are in, both in simulation and in the real world. Each embodiment consists of a arm and an end-effector, described in the following paragraphs. Table 11 shows the subset of proprioception observations used for each embodiment both during RoboCat training and evaluation.

We use 3 different arms for our experiments: the Sawyer from Rethink Robotics, the Panda Franka Emika, and the KUKA LBR IIWA14 arm. Across these setups, we use different controllers and end-effectors. And though all arms have the same degrees of freedom, their dynamics and state distributions are significantly different given differences between arm link lengths and their controllers. In addition, we sometimes constrain the Sawyer's degrees of freedom (see subsection 3.1).

For each arm, we did not align the simulation and real environments. For instance, we have different observations and the controllers are often different between simulation and reality. We also did not perform system identification for any embodiment to try to align the control dynamics of the simulated system with the real world. In simulation, we use physics randomisation but no visual randomisation.

Although the combination of different arms, end-effectors and discrepancies between simulation and real-world provides a total of 5 different embodiments, in the next sections we organise information by grouping it per robot arm.

All the environments for the arms are written with the open-source MoMa library[5], unless otherwise specified below. MoMa uses MuJoCo (Todorov et al., 2012) underneath as a physics engine for the simulation environments and a forward kinematics library for the real ones.

---

[4] The Sawyer is now developed and retailed by the Hahn Group.
[5] https://github.com/deepmind/dm_robotics/tree/main/py/moma

| | | Colour permutation | Triplet 1 Successes | Triplet 1 Failures | Triplet 2 Successes | Triplet 2 Failures | Triplet 3 Successes | Triplet 3 Failures | Triplet 4 Successes | Triplet 4 Failures | Triplet 5 Successes | Triplet 5 Failures | NIST-i gears Successes | NIST-i gears Failures |
|---|---|---|---|---|---|---|---|---|---|---|---|---|---|---|
| Simulation — Panda 7-DoF — Stacking | | Red-on-blue | 22416 | 10723 | 28285 | 3876 | 31096 | 3045 | 33131 | 2868 | 30336 | 1800 | 22587 | 2556 |
| | | Red-on-green | 31923 | 3454 | 32426 | 1297 | 24052 | 6998 | 33368 | 1639 | 27048 | 2780 | 19363 | 2248 |
| | | Blue-on-green | 32379 | 1122 | 27616 | 4407 | 31506 | 2575 | 32618 | 1708 | 30018 | 2529 | 18056 | 2582 |
| | | Blue-on-red | 28897 | 7008 | 29709 | 2122 | 30123 | 2304 | 32499 | 1715 | 29761 | 4124 | 24036 | 659 |
| | | Green-on-red | 31324 | 2828 | 31857 | 2458 | 26375 | 1548 | 32550 | 1511 | 28563 | 5104 | 17308 | 895 |
| | | Green-on-blue | 32559 | 1483 | 39255 | 4432 | 32437 | 1526 | 32473 | 1392 | 27739 | 7427 | 17993 | 1381 |
| | Pyramid | Red-on-blue-and-green | 17253 | 3212 | 14987 | 5022 | 13694 | 6220 | 14215 | 5533 | 15420 | 4455 | — | — |
| | | Red-on-green-and-blue | 17834 | 2567 | 14987 | 5022 | 14558 | 5513 | 16050 | 4282 | 14723 | 5807 | — | — |
| | | Blue-on-green-and-red | 17400 | 3001 | 15124 | 5269 | 15107 | 5232 | 18099 | 2180 | 15738 | 4388 | — | — |
| | | Blue-on-red-and-green | 15800 | 4491 | 13971 | 6152 | 12184 | 7907 | 18237 | 1960 | 15097 | 5108 | — | — |
| | | Green-on-red-and-blue | 11826 | 8300 | 11960 | 7923 | 11470 | 7088 | 15092 | 4663 | 17661 | 2648 | — | — |
| | | Green-on-blue-and-red | 13787 | 6188 | 12893 | 7370 | 15797 | 4463 | 14469 | 5450 | 17100 | 3466 | — | — |
| | Tower | Red-on-blue-on-green | — | — | — | — | — | — | — | — | 9851 | 2974 | 16692 | 5150 |
| | | Red-on-green-on-blue | — | — | — | — | — | — | 9427 | 2612 | 11643 | 5617 | — | — |
| | | Blue-on-green-and-red | — | — | 7775 | 4272 | — | — | 10775 | 2181 | — | — | 11016 | 992 |
| | | Blue-on-red-on-green | — | — | — | — | — | — | 9649 | 2116 | — | — | 6823 | 2132 |
| | | Green-on-red-on-blue | — | — | — | — | — | — | 10058 | 2856 | — | — | — | — |
| | | Green-on-blue-on-red | — | — | — | — | — | — | 6112 | 1807 | — | — | — | — |
| Sawyer 7-DoF — Stacking | | Red-on-blue | 28586 | 14238 | 33992 | 3971 | 32709 | 3156 | 33536 | 1534 | 36993 | 2642 | 17771 | 1901 |
| | | Red-on-green | 32217 | 2257 | 36067 | 2349 | 30834 | 3483 | 35457 | 1015 | 35161 | 3364 | — | — |
| | | Blue-on-green | 36802 | 1574 | 33009 | 2657 | 36296 | 3224 | 32567 | 2499 | 32836 | 1058 | 41595 | 4760 |
| | | Blue-on-red | 31011 | 4600 | 33877 | 2553 | 37078 | 3334 | 35258 | 2191 | 15658 | 1560 | 42044 | 2996 |
| | | Green-on-red | 36000 | 3446 | 36383 | 2802 | 33084 | 3575 | 34077 | 717 | — | — | 43968 | 3422 |
| | | Green-on-blue | 3114 | 15384 | 36284 | 3188 | 35120 | 1630 | 35072 | 2222 | 33885 | 1616 | 20234 | 3959 |
| Real — Sawyer 5-DoF — Stacking | | Red-on-blue | 12590 | 3886 | 9515 | 8515 | 11909 | 7800 | 15922 | 2493 | 9106 | 3548 | — | — |

Table 6: **Quantities of RL agent data for training simulated and real tasks.** Note that no real data has been collected with RL agents for the NIST-i tasks, lifting, insertion and removal.

| Embodiment | Task Family | Object Set | Variant | Human teleop demos | |
|---|---|---|---|---|---|
| | | | | Successes | Failures |
| **Panda 7-DoF** (Simulation) | Lifting | NIST-i gears | Small | 120701 | 18385 |
| | | | Medium | 122409 | 16517 |
| | | | Large | 126481 | 17145 |
| | Insertion-peg | | Small | 54813 | 30513 |
| | | | Medium | 70638 | 30895 |
| | | | Large | 84859 | 13972 |
| **Panda 7-DoF** (Real World) | Lifting | | Small | 8282 | 210 |
| | | | Medium | 7850 | 193 |
| | | | Large | 8791 | 240 |
| | Insertion-peg | | Small | 6618 | 889 |
| | | | Medium | 6715 | 601 |
| | | | Large | 6558 | 552 |
| | Removal-peg | | Small | 6699 | 1993 |
| | | | Medium | 6634 | 1642 |
| | | | Large | 6375 | 1460 |

Table 7: **Quantities of human demonstrations collected for the NIST-i tasks in simulation and with real robots**.

| Embodiment | Task Family | Object Set | Variant | Human teleop demos | | Agent trajectories | |
|---|---|---|---|---|---|---|---|
| | | | | Successes | Failures | Successes | Failures |
| **Sawyer 5-DoF** (Real World) | Lifting | YCB-i vegetables | Carrot | 2819 | 1389 | 3195 | 6438 |
| | | | Cucumber | 3162 | 1007 | 4052 | 5676 |
| | | | Pepper | 1938 | 2317 | 2217 | 7122 |
| | | | Potato | 2553 | 1570 | 2768 | 7102 |
| | Stacking | RGB objects | BG, triplet 1 | 424 | 0 | 4159 | 3853 |
| | Tower building | RGB objects | RBG, triplet 5 | 926 | 1821 | 4718 | 7776 |
| | Inverted pyramid | RGB objects | RBG, blocks | 1409 | 1510 | 1081 | 11155 |
| **Panda 7-DoF** (Real World) | Lifting | YCB fruits | Apple | 1956 | 121 | 8117 | 6058 |
| | | | Banana | 2320 | 90 | 4556 | 6352 |
| | | | Peach | 2200 | 98 | 2409 | 11125 |
| | | | Strawberry | 2052 | 137 | 195 | 2171 |

Table 8: **Quantities of human demonstrations and self-generated data.**

| Embodiment | Task Family | Object Set | Variant | Human teleop demos | |
|---|---|---|---|---|---|
| | | | | Successes | Failures |
| **Panda 7-DoF** (Real World) | Insertion-bowl | YCB fruits and YCB-i bowl | Lemon | 4039 | 557 |
| | | | Peach | 2807 | 355 |
| | | | Strawberry | 1958 | 394 |
| | Removal-bowl | YCB fruits and YCB-i bowl | Lemon | 2630 | 419 |
| | | | Peach | 2661 | 417 |
| | | | Strawberry | 1889 | 385 |
| | Insertion-base | Shape-matching objects | Pentagon | 2531 | 469 |
| | | | Circle | 2526 | 376 |
| | | | Square | 1492 | 383 |
| | Removal-base | Shape-matching objects | Pentagon | 2684 | 313 |
| | | | Circle | 2534 | 336 |
| | | | Square | 1688 | 284 |
| **KUKA 14-DoF** (Real World) | Lifting | NIST-i gears | Large | 1956 | 121 |

Table 9: **Quantities of human demonstrations collected for final fine-tuning tasks.** Note that for fine-tuning we only used a subset, either 500 or 1000, of successful episodes and no failed episodes.

| Embodiment | Control Frequency | Task Family | Episode Length | Number of Steps | Success Rate |
|---|---|---|---|---|---|
| **Panda 7-DoF** (Real World) | 10 Hz | Stacking | 60 s | 600 | 96.5% |
| | | Tower Building | | | 81.6% |
| | | Fruit Lifting | 30 s | 300 | 95.4% |
| | | Fruit Inserting | | | 90.0% |
| | | Fruit Removing | | | 89.1% |
| | | Gear Lifting | 120 s | 1200 | 93.8% |
| | | Gear Inserting | | | 76.7% |
| | | Gear Removing | | | 84.3% |
| | | Shape Inserting | 30 s | 300 | 84% |
| | | Shape Removing | 30 s | 300 | 88% |
| **Sawyer 5-DoF** (Real World) | 10 Hz | Stacking | 40 s | 400 | 74.3% |
| | 20 Hz | Vegetable lifting | 20 s | 1200 | 63.2% |
| | | Tower Building | 60 s | 1200 | 50.0% |
| | | Inverted Pyramid | 120 s | 2400 | 52.6% |
| **KUKA 14-DoF** (Real World) | 10 Hz | Gear lifting | 60 s | 600 | 88.4% |

Table 10: **Human teleoperator success rates on real robot tasks**. This table also reports the control frequency, episode length and number of steps used during human teleoperation for each task.

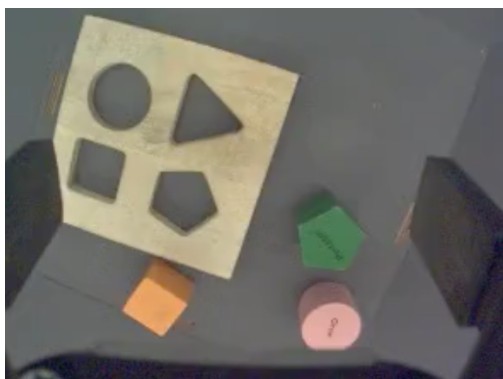

Figure 13: **Example view of the wrist camera.** This image comes from one of the Basler dart cameras attached to the wrist of the Franka Panda arms, with resolution $296 \times 222$.

### C.1 Robot arms

### C.1.1 Sawyer

The Rethink Sawyer arm is fitted with a Robotiq 2F-85 gripper. In simulation, the environment exposes a 7-DoF action space: 6-DoF end-effector Cartesian velocity control and a 1-DoF velocity command to control the aperture of the parallel gripper. The 6-DoF Cartesian control builds on top of 7-DoF joint integration actuators whose dynamics are not aligned with their real-world counterparts.

For the real robots, we follow the RGB Stacking benchmark (Lee et al., 2021). The environment exposes a simplified 5DoF action space consisting of a 4-DoF end-effector Cartesian velocity control and the same 1-DoF velocity control for the gripper. The action is reduced by restricting the gripper to be oriented vertically (3D translation and 1D rotation along the vertical axis). This Cartesian end-effector controller builds on top of the Sawyer's pure joint velocity controller. The real Sawyer environment is the only one which does not use the MoMa library.

In both simulation and the real world, besides the basket cameras (described in Appendix C.2), the full set of available observations comes from the robot's proprioception and a Robotiq FT 300 force-torque sensor affixed to the Sawyer's wrist, again matching the RGB Stacking benchmark. See (Lee et al., 2021) for more details on the setup.

### C.1.2 Panda

Like for the Sawyer, the Panda Franka Emika arm is fitted with a Robotiq 2F-85 gripper. In both simulation and the real world, the environment exposes a 7-DoF action space: 6-DoF Cartesian velocity control that is integrated into a reference pose that the end-effector tracks, and a 1-DoF velocity command to control the aperture of the parallel gripper. In simulation the 6-DoF integrated velocity Cartesian controller builds on top of a 7-DoF integrated joint velocity controller. Whereas in the real world we use a joint torque controller that tracks the target velocity. In addition, 2 Basler dart cameras are mounted on the gripper providing the view shown in Figure 13. The cameras mounted on the gripper have *not* been used to train or evaluate RoboCat.

In both simulation and the real world, the full set of proprioception observations includes:

- Joint positions, velocities and torques of the arm.
- Pose and velocity of the tool center point located between the 2 fingers of the gripper.
- Pose of the integrated reference pose that the TCP tracks.
- Position and velocity of the fingers of the gripper.

See Table 11 for the subset of proprioception observations used for training RoboCat.

### C.1.3   KUKA

We use the KUKA LBR IIWA14 in combination with a 3 fingered robotic hand. We only use this environment in the real world, not in simulation. The environment exposes a 14-DoF action space: 6-DoF Cartesian velocity control that is integrated into a reference pose that the end-effector tracks, and a 8-DoF abstraction command to control the fingers of the hand. The 6-DoF integrated velocity Cartesian controller builds on top of a 7-DoF joint torque controller that tracks a target joint velocity. Similarly to the panda, 2 cameras are mounted on the gripper and, also in this case, they are not used for training or evaluating RoboCat.

The full set of proprioception observations include:

- Joint positions, velocities and torques of the arm.
- Estimated force and torque at the wrist of the robot used computing the joint torque sensors of the robot.
- Pose of the integrated reference pose that the TCP tracks.
- Proprioception of the 3 fingered hand.

Note that we only use a subset of proprioception observations for training RoboCat which we describe in Table 11.

### C.2   Basket and setup

All embodiments, in both simulation and reality, are mounted in a standardised cage in front of a standardised basket which defines the robot arm's workspace (Lee et al., 2021). The basket's base is 25 cm by 25 cm, and it has 2 Basler ace cameras fixed at its front corners. In the Sawyer environment, these cameras provide $128 \times 128$ image observations. In the Panda and KUKA environments, they provide $296 \times 222$ images.

### C.3   Environment resets

In our real-world environments, for most object sets, we do not reset the objects between episodes. Rather, we rely on the next episode's task to achieve the "reset." In between episodes, we only verify that the physical system still functions and reset the arm into a new random pose above the basket. See Appendix F.2 for more details.

However, for RGB objects, we reuse the scripted resets from the RGB Stacking benchmark (Lee et al., 2021). The reset uses a blob-based 3D position tracker for the RGB objects to locate objects and then uses a simple scripted pick-and-place controller to move the objects to new locations in between episodes.

In simulation, the environment simply places the objects in new randomised poses at the start of each episode automatically.

# D  VQ-GAN Training Details

## D.1  Datasets

**RoboCat-lim VQ-GAN**  We train the RoboCat-lim VQ-GAN on several sources of data: ImageNet (Deng et al., 2009), images from the DeepMind Control Suite (Tassa et al., 2018), images from Meta-World (Yu et al., 2020), domain randomised sim sawyer red-on-blue-stacking data from Lee et al. (2021), and images from the tasks used to train the RoboCat-lim agent: the sim panda data and the real sawyer red-on-blue stacking data. Images and reconstructions from these datasets are shown in Figure 14. Reconstructions of tasks not included in the VQ-GAN training are shown in Figure 15.

**RoboCat VQ-GAN**  The RoboCat VQ-GAN was additionally trained on data from a simulated version of the YCB fruit lifting task, the NIST-i gear task images (real and sim), and real sawyer data of a red-on-blue agent being run with random objects in the bin, including YCB fruits and the vegetables. We didn't include the YCB fruit lifting/insert/remove and vegetable lifting data from human teleoperators in the training. Images and reconstructions from these datasets are shown in Figure 16. Reconstructions of tasks not included in the VQ-GAN training are shown in Figure 17.

## D.2  Model architecture and loss

The model architecture is derived from Esser et al. (2021), which combines convolutional layers with Transformer attention layers to encode the image. The encoder is a ResNet with 2 groups of 2 blocks, followed by 3 attention layers. The decoder has 3 attention layers followed by a ResNet with 2 groups of 3 blocks. The vector quantiser (van den Oord et al., 2017) uses 64 embeddings of dimension 128. The loss is weighted as $(0.25 * \text{discretisation\_loss} + \text{l2\_reconstruction\_loss} + 0.1 * \text{log\_laplace\_loss})$. We train with a batch size of 64 for 1 million training steps.

## D.3  Ablations for our VQ-GAN design choices

In this subsection we discuss the ablation experiments that informed certain design choices that we made regarding the VQ-GAN encoder and the observation prediction loss that we used. For these comparisons, we trained smaller models of 62-million parameter on subsets of the 30 **Sim Panda Stacking** tasks with **RGB Objects**. For these experiments, we only trained the models with hindsight goals. Also, during evaluation we always reset our simulation to the first state of the trajectory that corresponds to the goal image, which is a much easier task compared to the tasks described in the rest of this paper. We considered 3 generalisation problems within this context: `red_on_blue` where 5 red-on-blue variations are held out from each triplet; `red_on_green` where now the 5 red-on-green variations are the ones held out; and `triplet_1` where the 6 stacking task variations of the RGB Triplet 1 are held out across all variations. In each case, we trained and ablated separate generalists on the remainder tasks.

As shown in Figure 19, in the context of these generalisation problems, a RoboCat model with the VQ-GAN tokeniser performs much better than the patch ResNet tokeniser, especially on the held-out test tasks, but requires both training on a diverse dataset that includes ImageNet, and the observation token prediction auxiliary loss.

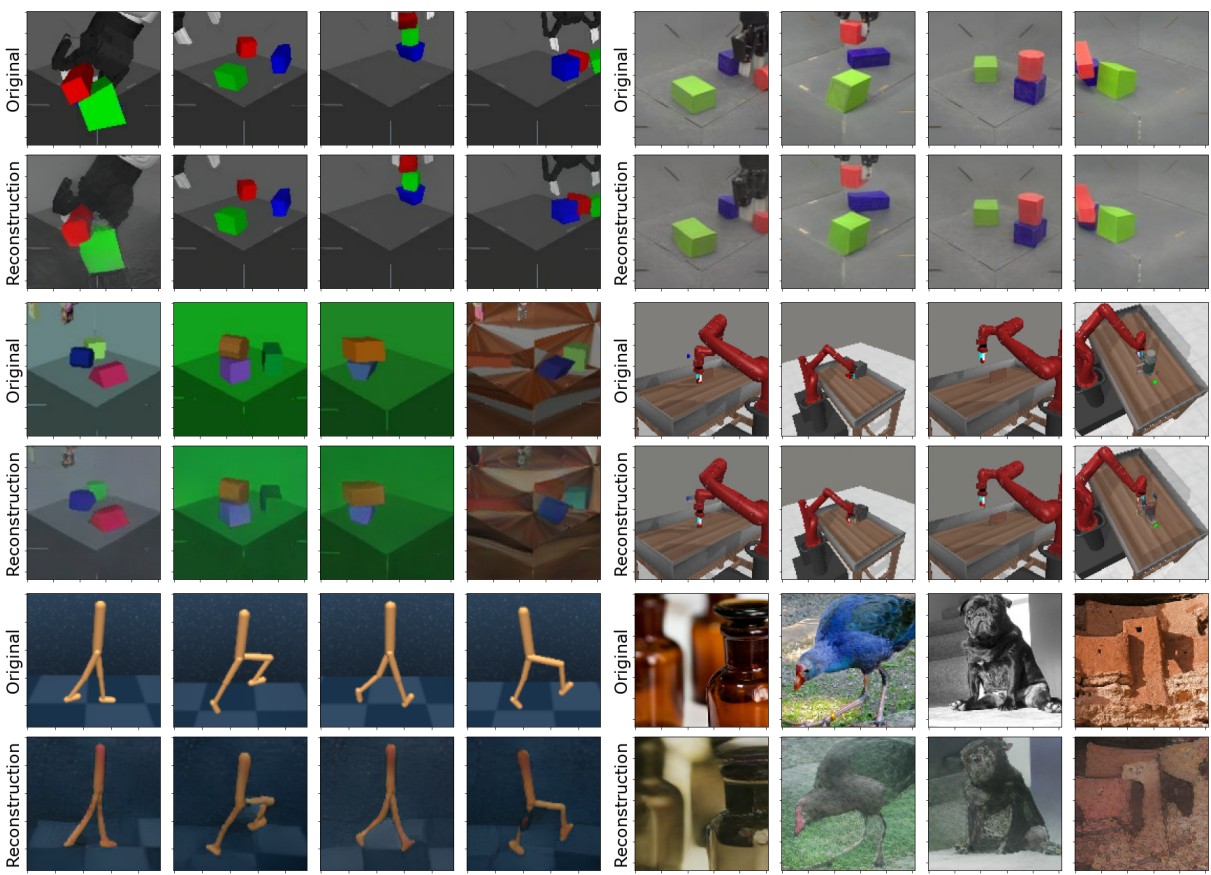

Figure 14: **Reconstructions from our RoboCat-lim VQ-GAN on the training datasets.** From right to left, panda sim, sawyer real red-on-blue, sawyer sim (with visual domain randomisation), MetaWorld, DM control, and ImageNet.

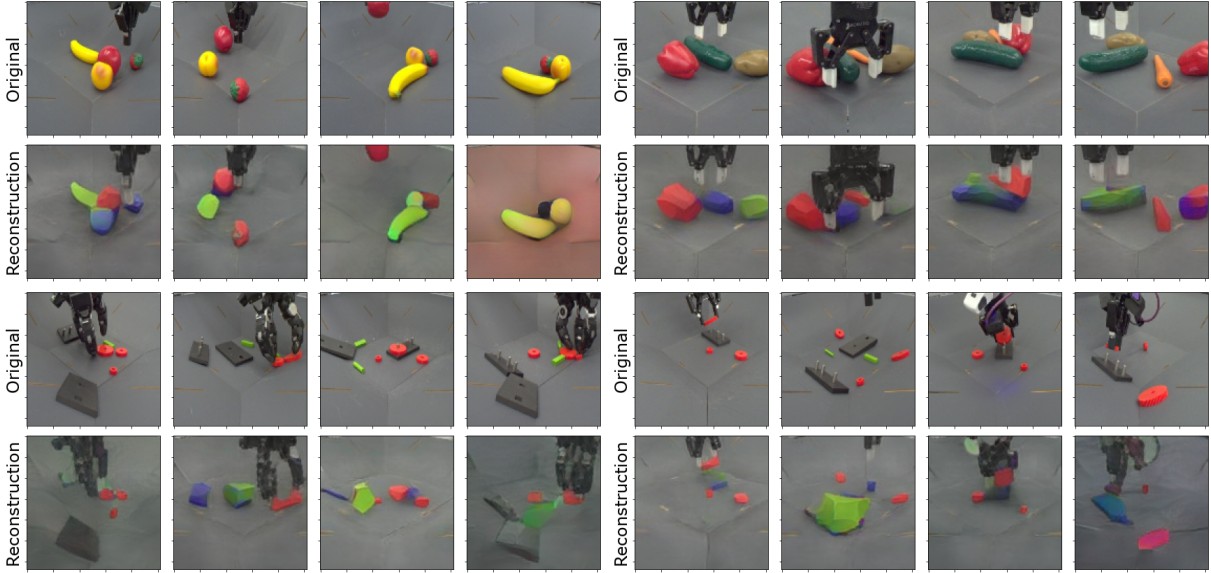

Figure 15: **Reconstructions of tasks not included in the RoboCat-lim VQ-GAN training:** YCB fruit lifting and vegetable lifting. Although the reconstructions are inaccurate, they contain enough information for the agent to learn the task.

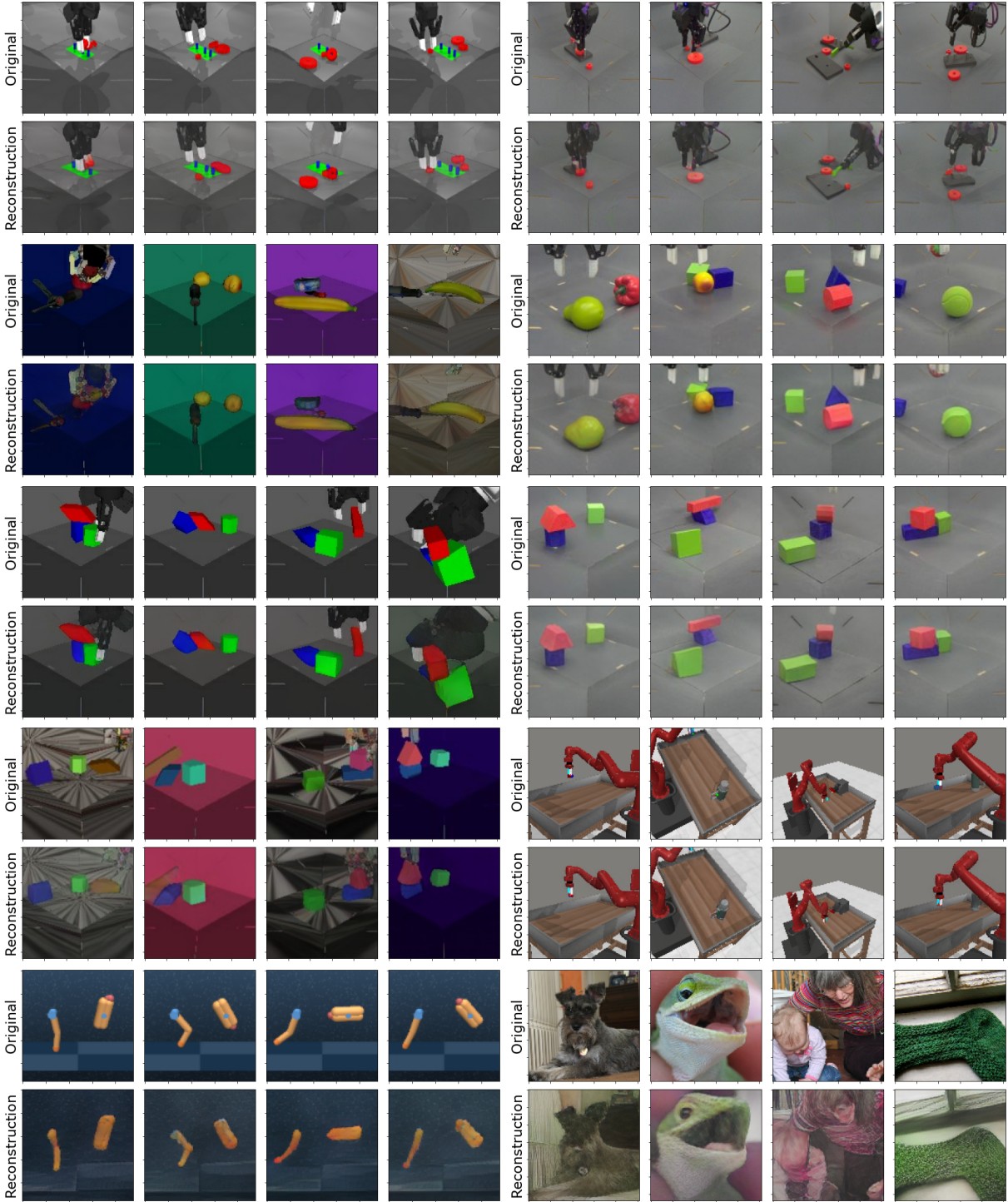

Figure 16: **Reconstructions from our RoboCat VQ-GAN on the training datasets.** From right to left, panda sim, sawyer real red-on-blue, sawyer sim (with visual domain randomisation), Meta-World, DM control, and ImageNet.

| Embodiment | | Object set | Observation | Observation dimensions |
|---|---|---|---|---|
| Simulation | Sawyer 7-DoF | RGB objects | Joint angles | 7 |
| | | | TCP position | 3 |
| | | | Gripper joint angle | 1 |
| | | | Gripper grasp status | 1 |
| | Panda 7-DoF | RGB objects | Joint angles | 7 |
| | | | TCP position | 3 |
| | | | Pose of the integrated reference pose the TCP tracks | 7 |
| | | | Gripper position | 1 |
| | | | Gripper grasp status | 1 |
| | | NIST-i gears and base | Joint angles | 7 |
| | | | TCP pose | 7 |
| | | | Gripper position | 1 |
| | | | Gripper grasp status | 1 |
| Real World | Sawyer 5-DoF | RGB objects, YCB-i objects | Joint angles | 5 |
| | | | Gripper joint angle | 1 |
| | | | Pinch pose | 7 |
| | | | TCP pose | 7 |
| | Panda 7-DoF | YCB, YCB-i objects | Joint angles | 7 |
| | | | TCP position | 3 |
| | | | Pose of the integrated reference pose the TCP tracks | 7 |
| | | | Gripper joint angle | 1 |
| | | | Gripper grasp status | 1 |
| | | NIST-i gears and base | Joint angles | 7 |
| | | | TCP pose | 7 |
| | | | Gripper joint angle | 1 |
| | | | Gripper grasp status | 1 |
| | KUKA 14-DoF | NIST-i gears and base | Joint angles | 7 |
| | | | TCP position | 3 |
| | | | TCP orientation (rotation matrix) | 9 |
| | | | Pose of the integrated reference pose the TCP tracks | 7 |
| | | | Finger joint angles | 12 (4 per finger) |
| | | | Fingertip positions | 9 (3per finger) |
| | | | Fingertip orientations (rotation matrix) | 27 (9 per finger) |
| | | | Fingers clutch slip | 15 (5 per finger) |

Table 11: **Proprioception observations** used for different embodiments and object sets while training RoboCat.

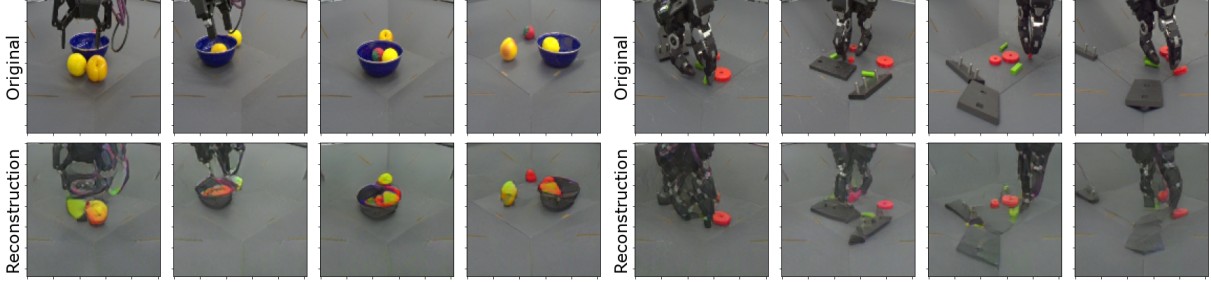

Figure 17: **Reconstructions of tasks not included in the v1.0 VQ-GAN training:** YCB fruit insert/remove, and the 3-fingered gripper. While the vegetables and YCB fruit were seen in the agent play data, the bowl was not seen at all in the training data.

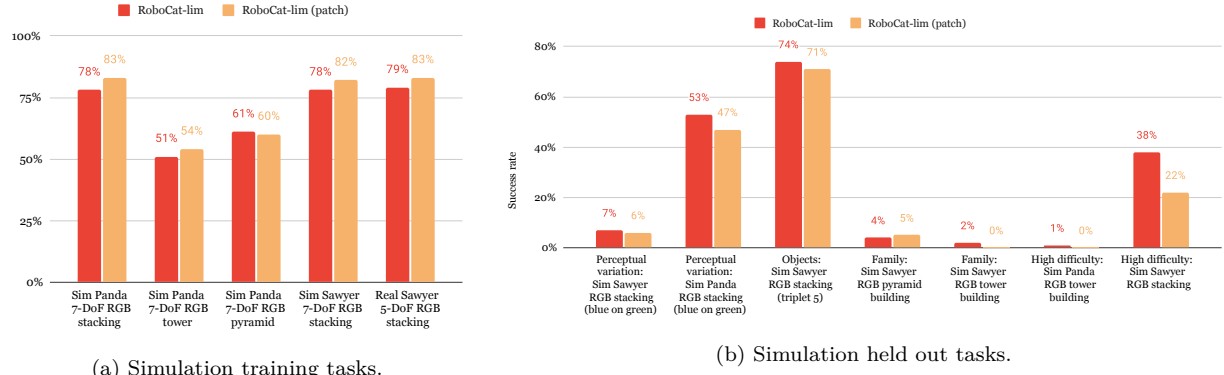

(a) Simulation training tasks.

(b) Simulation held out tasks.

Figure 18: **Ablating the VQ-GAN tokeniser vs the patch ResNet used in Gato.** The patch ResNet tokeniser performs better on the training tasks, but worse on the held out tasks.

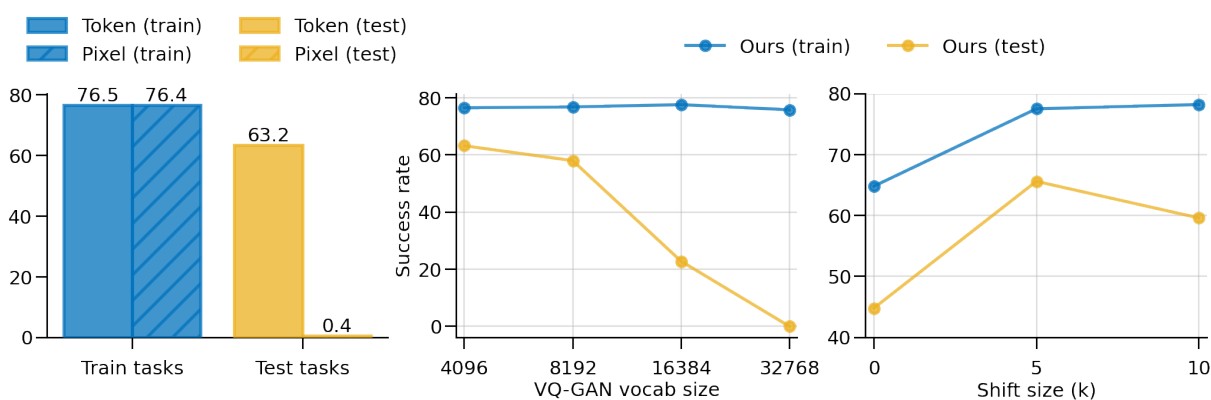

Figure 19: **Tokenisation and obs prediction ablations** on the `red_on_green` generalisation problem. Unless otherwise specified, we use a vocabulary size of 4096 and image prediction with a time step shift of $k = 5$. We compare the success rate (%) for train and test tasks to ablate important design decisions. *Left*: Predicting future *tokens* is significantly better than predicting future *pixels*. *Middle*: Increasing VQ-GAN vocabulary size deteriorates test performance. *Right*: Predicting the next token in the current image ($k = 0$) is less beneficial than predicting tokens farther in the future. We find predicting $k = 5$ time steps in the future is optimal.

At the larger 1.18B parameter scale, we compare RoboCat-lim to the patch ResNet tokeniser used in Gato. Both models are trained on the same data. We evaluate training task performance, as well as zero-shot performance for various groups of held-out tasks. These groups include held-out perceptual variations (blue-on-green stacking), objects (triplet 5 Sawyer stacking), task family (Sawyer pyramid and tower building), and high difficulty tasks (tasks within the training families that did not have a sufficiently good expert for inclusion in RoboCat training). As shown in Figure 18, the patch ResNet tokeniser performs better on most training task families, but generally worse for the different types of held out tasks. This mirrors our findings with smaller models, which showed VQ-GAN to be better for generalisation and adaptation.

Predicting future image pixels does not give us a similar advantage. We compared different reconstruction targets in Figure 19. Like the decoder in MAE (He et al., 2022), we added an extra linear layer whose number of output channels equals the number of pixel values in a patch. Output tokens corresponding to observation tokens were processed via this linear layer to produce a vector of pixel values in a patch. We used mean squared error between the predicted ($k = 5$) and original frame in pixel space. The choice of prediction

target does not seem to affect the performance on training tasks. However, there is a huge difference in the generalisation capabilities of the two methods. Qualitatively we found that the policy learnt by predicting pixels fails (zero success rate) as it seems to ignore the goal and performs a different task.

Next, we investigated predicting future VQ-GAN tokens for the `red_on_green` generalisation task, with different $k$ values, where $k$ represents the number of time steps ahead at which we are predicting the next observation token. We sweep from $k = 0$, i.e. predicting the next token, to $k = 10$ i.e. predicting the next token of the observation 10 time steps ahead. As $k$ increases, the performance on the training tasks increases, but we find diminishing returns as we increase past $k = 5$. However, we see a 50% relative improvement in generalisation performance as we increase $k$ from 0 to 5. Further increase in prediction horizon results in performance degradation. These results highlight the importance of designing a challenging self-supervision task. There is a huge amount of spatial and temporal redundancy between consecutive image observations especially when the agent is operating at 20 Hz. Despite using VQ-GAN tokens, there can be a significant overlap of tokens among future consecutive frames. Predicting $k$ time steps in the future alleviates this issue and results in strong generalisation capabilities in this setting.

Finally, we study the effect of the VQ-GAN vocabulary size in Figure 19. Increasing the vocabulary size ($V$) has no effect on the performance of the training tasks. However, the generalisation ability monotonically declines as we increase $V$.

## E  RoboCat Training Details

### E.1  Data processing

The training data for each task contains

- a subset of embodiment proprioception observations (see Table 11);

- the embodiment actions (see Appendix C.1);

- and two camera images, in particular from a front left and front right view of the basket (see Appendix C.2).

Each goal image used in training comes from the front left camera and correspond to the image obtained at the end of a training trajectory. All images, including the goal images, are cropped and resized to have dimensions $128 \times 128 \times 3$. We apply image augmentation by randomising brightness, saturation, contrast, and translation to the images, as described in Lee et al. (2021). All actions are scaled to be in $[-1, 1]$. We trim trajectories discarding the final steps if no significant difference is measured with respect to the final images of the scene. This is based on the intuition that expert actions that do not generate any changes in the images should not be used at training as they likely do not have an effect on the task at hand (e.g. the robot arm is not moving or is just hovering around on the objects). The metric we use to evaluate similarity between images is the the LPIPS (Learned Perceptual Image Patch Similarity) (Zhang et al., 2018). The trajectories involving just lifting of objects are also trimmed in order to have the gripper (and thus the lifted object) visible in the final image used as goal image.

### E.2 Data weighting

Based on previous experiments we use the following formula to sample the data during training. For training the RoboCat-lim generalist, by default every task variant is sampled equally, regardless of how many episodes are available. And within each task variant, failed episodes and successful episodes are also sampled equally. For the successful episodes, the hindsight goal image is used half the time and a semantically equivalent goal image (from a different successful episode of the same task variant) is used the other half of the time. When fine-tuning to a limited number of demonstrations, we find best performance when only training on successful episodes. For the final RoboCat generalist we included NIST-i data only from successful episodes and with a weighting three times bigger to mitigate a potential imbalance towards stacking and lifting other object sets. Similarly, after 1 000 000 training steps we decreased by half the weight of all simulated data to focus the remaining training time towards the real world tasks. Note that these decisions were preventive and thus keeping all weights equal may have also been effective.

### E.3 Training and fine-tuning parameters

For training all RoboCat models we use the AdamW optimiser (Loshchilov and Hutter, 2017) with a linear warm-up and cosine schedule decay. The linear warm-up lasts for 15 000 steps, starting from and ending at a different minimum and maximum learning rates depending on the model (see Table 13). This learning rate is then cosine decayed by a factor 10 over 2 000 000 steps. The AdamW optimiser has parameters $\beta_1 = 0.9$, $\beta_2 = 0.95$ and $\epsilon = 1e^{-8}$. We use a batch size of 256 and a sequence length of 1024 tokens for all models. We train with an AdamW weight decay parameter of 0.1. Additionally, we use stochastic depth (Huang et al., 2016) during pretraining, where each of the transformer sub-layers (i.e. each Multi-Head Attention and Dense Feedforward layer) is skipped with a probability of 0.1.

| Hyper parameter | RoboCat | RoboCat-lim | 364M |
|---|---|---|---|
| Max learning rate | $1e^{-4}$ | $5e^{-6}$ | $1e^{-4}$ |
| Min learning rate | $1e^{-5}$ | $5e^{-7}$ | $1e^{-5}$ |

Table 13: **Learning rates used when training our models**. Note that both RoboCat-lim and RoboCat-lim (patch) used the same learning rates.

For fine-tuning we use the Adam optimiser (Kingma and Ba, 2015) with a constant learning rate of $1e^{-5}$. The Adam optimiser has parameters $\beta_1 = 0.9$, $\beta_2 = 0.95$ and $\epsilon = 1e^{-8}$. We use a batch size of 32 and a sequence length of 1024 tokens for all models. We train for up to 50 000 gradient steps. As regularisation, we use dropout (Srivastava et al., 2014) with a rate of 0.1.

### E.4 RoboCat-lim training data

In subsection 5.2, we evaluate the properties of the RoboCat-lim agent and systematically measure its ability to generalise and fine-tune across a diverse set of axes: objects, task variants, embodiments, behaviour sources, task family, and from sim to real. In this section, we provide more details on our choice of held-out tasks.

The training tasks for RoboCat-lim are primarily in sim, comprising the simulated structure building tasks: stacking, pyramid building, and tower building for the Panda 7DoF; and stacking for the Sawyer 7DoF. The only real-world data used in training is the Sawyer 5DoF RGB-stacking benchmark.

From these training tasks, we also hold out all blue-on-green stacking tasks and Sawyer stacking with RGB triplet 5. This provides the first two axes of generalisation: to the **unseen perceptual variation** (blue-on-green task, which has not been observed before) and the **held-out objects** (triple 5, which has never been seen with the Sawyer embodiment).

To measure the ability to fine-tune to different **behaviour sources** we use two versions of data for blue-on-green stacking in the real-world: from an expert agent, and from human teleoperation. These have different state and action distributions, as (for example) human teleoperators tend to favour quick but constant movements with pauses between 'stages' of a task.

We measure **sim-to-real capability** with two tasks on the real panda 7DoF: one from the stacking task family and one from tower building.

We also evaluate the ability to fine-tune to an **unseen task family**: inverted pyramid building. This task is particularly challenging as it requires careful and precise motion to balance two objects on top of a third.

Finally, we evaluate **embodiment generalisation** and fine-tuning using the KUKA 14DoF. While the KUKA 14DoF is a dexterous embodiment with a 3-fingered hand, it also presents a considerable challenge for RoboCat training. This is due to the significantly different observation and action spaces that it poses compared to all other tasks in the RoboCat training. For instance, all training tasks rely on a 1DoF parallel gripper and have a total of either 5DoF or 7DoF.

| | Embodiment | Task Family | Episode Length | Control Frequency | Number of Steps |
|---|---|---|---|---|---|
| Simulation | Panda 7-DoF | Stacking Tower Building Pyramid Building | 20 s | 20 Hz | 200 |
| | | Lifting | 120 s | 10 Hz | 1200 |
| | | Inserting | 120 s | | 1200 |
| | Sawyer 7-DoF | Stacking | 20 s | 20 Hz | 400 |
| Real World | Panda 7-DoF | Stacking Tower Building | 60 s | 10 Hz | 600 |
| | | Fruit Lifting Fruit Inserting Fruit Removing | 30 s | | 300 |
| | | Gear Lifting Gear Inserting Gear Removing | 120 s | | 1200 |
| | | Shape matching | 30 s | | 300 |
| | Sawyer 5-DoF | Stacking | 40 s | 10 Hz | 400 |
| | | Vegetable lifting | 20 s | | 400 |
| | | Tower Building | 60 s | 20 Hz | 1200 |
| | | Inverted Pyramid | 120 s | | 2400 |
| | KUKA 14-DoF | Gear lifting Peg lifting | 60 s | 10 Hz | 600 |

Table 14: **Evaluation setup.** Episode length, control frequency and episode number of steps per task on simulated and real robots.

# F   Evaluation and Real-World Deployment

In this section, we provide more details about the evaluation procedure used in the paper.

We evaluated RoboCat and the baselines with 100 (or more, if specified) episodes for each evaluation task, both in simulation and on the real robot. For the fine-tuning experiments, we ran shorter evaluations of 25 episodes for different fine-tuned checkpoints. We then selected the best checkpoint and ran the 100 episode evaluation with it.

In simulation, the initial state is randomised by moving the arm to a randomised initial pose and by dropping the objects in the basket. In the real world, the arm position is randomised similarly. The objects initial positions are either randomised via a scripted reset or with alternative approaches. We detailed these processes in Appendix F.2.

For all our evaluations, we use environments with the control frequencies and episode lengths that match the ones used for collecting the data for each task. Table 14 reports this information for our simulated and real robot tasks.

In the next two sections, we detail the evaluation procedure used the real-world tasks for the RGB tasks and all other tasks.

### F.1 Evaluation for real-world RGB tasks

To evaluate our agent on tasks with the RGB objects we used the infrastructure implemented in Lee et al. (2021). We rely on the scripted resets mentioned in Appendix C.3 and follow the same evaluation procedure, apart from the reward function. We noticed that the reward function was missing some clear successes. This is because the reward function defined in Lee et al. (2021) requires the arm height to end withing some thresholds. While such a criteria for success can be fair for RL agents trained to optimise such a reward, we thought it didn't represent fairly the ability of RoboCat to stack the correct objects and move the arm away to *any* position. For this reason, we decided to visually count the successes during evaluation. For fairness we also visually recounted the evaluations of Lee et al. (2021) and Reed et al. (2022).

### F.2 Evaluation for real-world tasks with no rewards or scripted resets

For all the other real tasks in this work we did not implement a scripted reward model nor a reset policy. This greatly simplified the implementation of the tasks, but added significant complexity to evaluation. Our evaluation system thus consists of two components. In place of scripted rewards, we train success detectors from human annotations to detect when each task has been completed. For resets we have a general solution that groups several policies for different tasks together into a *policy pool*, with each policy in the pool serving as a component of the reset policy for its peers. For instance a policy pool containing the insertion and removal policies for the same object will allow the removal policy to serve as a reset for the insertion policy and vice versa.

#### F.2.1 Success detection

We treat success detection for each task as a per-time step binary classification problem. For each task we annotate several episodes with per-time step success labels using a annotation-efficient annotation procedure. Our annotation procedure takes advantage of the following simple observation: If a task is solved (or not) at time $t$ it is overwhelmingly likely that it is also solved (or not) at time $t + 1$. Put more simply, there are relatively few time steps where a task transitions from unsolved to solved (or vice versa). Rather than collecting labels for every frame directly, we ask annotators to mark transition points between solved and unsolved states and extend these to per-time step labels by painting the transition label forward in time until the next annotation. This strategy dramatically reduces the number of annotations compared to

annotating every frame. Using this strategy labelling a 1200 time step episode typically requires fewer than five annotations (and frequently requires only one).

Using data annotated in this way we train success detectors for each task. Each success detector is a ResNet-18 backbone with a binary classification head. To handle occlusions we use the three basket cameras in the reward model in order to observe the scene from multiple perspectives. Each camera image is scaled to $132 \times 132$ and further cropped to $128 \times 128$ before being processed separately by the ResNet backbone. The resulting embeddings from each camera are averaged before being fed to the binary classification head. At train time we use random cropping and image augmentation, while at deployment time we disable the augmentations and take a central crop of each image.

Once trained the success detectors are used together with policy pools for resetting our environments as we describe in Appendix F.2.2. We did not use success detectors for reporting RoboCat performance, as they have an accuracy around 90%. To ensure we counted all success from RoboCat we visually assessed when an experiment has succeeded or failed. In future work, we aim at improving accuracy of the reward models in order to automatise also this step of the evaluation.

### F.2.2  Policy pools as generalised reset policies

Policy pools are our general solution to resets in the real world. Their role in our set up is three-fold: i) policy pool provides resets for non-self-resetting tasks, ii) generates diverse initial conditions for evaluation, and iii) makes efficient use of robot time by interleaving the evaluation of multiple policies at a time.

A policy pool groups together a set of diverse tasks and allows us to schedule episodes of different tasks to be run in sequence. In this simplest case, we can replicate the standard evaluation setting by building a pool comprised of a forward task and its reset, allowing the schedule to alternate between them. However, the policy pool machinery allows us to extend beyond this simple setting in several ways.

Each prop set typically affords several tasks, for example the NIST-i gears set affords nine distinct tasks: lift, insert, and remove for each of the three gears. A policy pool allows us to group all nine of these tasks together and interleave their execution in arbitrary orders. This is desirable for two reasons which we discuss next.

From a global perspective this procedure makes efficient use of robot time, because every episode is an evaluation of some policy on some task. There is no time wasted by running a specialised behaviour that resets the environment; instead the evaluation episodes for each task simultaneously serves as the reset behaviour for other tasks in the pool.

From a local perspective this procedure generates a wider variety of initial conditions than a single fixed reset policy. From the perspective of each task its reset is provided by a composition of all the other tasks in the pool. This has the nice effect that larger pools (with more diverse tasks) tend to generate more diverse initial conditions within each task.

Another extension of the basic setting is that we can control the ordering of the schedule of tasks within a policy pool. In practice, this is how we handle non-self-resetting tasks. In the general case not all tasks are eligible to be run from all reachable states of the environment. Returning to the gears example, an insert task is eligible to be run only if the corresponding gear is not already inserted on the target shaft (similarly

a remove task is only eligible when the corresponding gear is inserted). At the beginning of each episode our scheduling system for the policy pool uses the success detectors (Appendix F.2.1) to determine which tasks are eligible to be run given the state of the environment, and then randomly chooses one of these tasks to actually run.

The mapping between tasks and policies within a policy pool is arbitrary, and need not be 1-1. The same machinery can be used to implement single task evaluation (by providing a different policy for each task in the pool) or multi-task evaluation (by defining a pool where every task is executed by the same policy). Given RoboCat is generalist, when using policy pool we only use the policy from RoboCat to evaluate any task.

## G  Additional Experiments

### G.1  Generalist capabilities and expert performance

In this section, we report more details on the performance of RoboCat on training and fine-tuning tasks, and we compare it to the success rate of the training data, defined as percentage of the successful episodes among all the training episodes. Table 15 shows the overall RoboCat performance per each task variant.

#### G.1.1  Training tasks

We use **RL-trained experts** for all of the structure-building tasks in sim (Panda 7-DoF stacking, pyramid, and tower building; Sawyer 7-DoF stacking, Figure 20) and in real (Sawyer 5-DoF stacking, Figure 21). These experts are state-based for the sim tasks, and vision-based for the real-world tasks. In the sim case, the vision-based RoboCat generalist has a performance reduction of 10% or less compared to the state-based single-task experts (Figure 20), for all families except tower building. It is important to note that these experts were trained via RL with privileged state information which included the poses of the target objects. In contrast, our vision-based generalist agent discards any privileged state information in order to perform effectively in the real-world. As such, we would not expect RoboCat to improve upon or even match the success rate of the data for all tasks. On the other hand, when RoboCat has access to same information as the experts that generated the data, as for the real tasks Figure 21, it is able to overcome the performance of the agents generating data.

**Human-teleoperated demonstrations** were used for the NIST-i tasks in both sim and real. Figure 22 and Figure 23 compare the success rate of the human teleoperators and RoboCat separately for each task variant. In both cases, we can see that RoboCat on average matches the human teleoperator performance.

**Self-generated data** were used together with a limited number of human-teleoperated demonstrations to train RoboCat on additional real robot tasks. Given a limited number of demonstrations for a new task, we first fine-tuned previous versions of RoboCat to solve this new task. We then deployed the fine-tuned RoboCat in the real world to collect more self-generated data for the specific task. The newly generated data together with the successful human teleoperations were used in the final generalist. Figure 24 and Figure 25 aim to contextualise the performance of RoboCat on these tasks. The 'generated + demos' numbers refer to the percentage of successful episodes among the total number of episodes used to train RoboCat on these tasks. Successful episodes include successful self-generated episodes and successful teleoperations. We use

| | Embodiment | | | Task Family | Object Set | Variations | | | | | |
|---|---|---|---|---|---|---|---|---|---|---|---|
| **Training** | **Simulation** | | Sawyer 7-DoF | Stacking | | **RB** | **RG** | **GR** | **GB** | **BR** | **BG** |
| | | | | | RGB objects, triplet 1 | 62% | 86% | 85% | – | 73% | 92% |
| | | | | | RGB objects, triplet 2 | 79% | 87% | 85% | 82% | 85% | 83% |
| | | | | | RGB objects, triplet 3 | 90% | 86% | 80% | 90% | 89% | 91% |
| | | | | | RGB objects, triplet 4 | 92% | 95% | 91% | 82% | 81% | 85% |
| | | | | | RGB objects, triplet 5 | 69% | 80% | – | 87% | 63% | 89% |
| | | | | | NIST-i gears | 80% | – | 80% | 62% | 79% | 82% |
| | | | Panda 7-DoF | Stacking | | **RB** | **RG** | **GR** | **GB** | **BR** | **BG** |
| | | | | | RGB objects, triplet 1 | 69% | 82% | 77% | 80% | 74% | 85% |
| | | | | | RGB objects, triplet 2 | 77% | 84% | 82% | 76% | 77% | 77% |
| | | | | | RGB objects, triplet 3 | 77% | 72% | 89% | 91% | 63% | 85% |
| | | | | | RGB objects, triplet 4 | 88% | 90% | 93% | 85% | 83% | 90% |
| | | | | | RGB objects, triplet 5 | 92% | 77% | 83% | 80% | 80% | 80% |
| | | | | | NIST-i gears | 67% | 69% | 66% | 72% | 82% | 69% |
| | | | | Tower building | | **RBG** | **RGB** | **GRB** | **GBR** | **BRG** | **BGR** |
| | | | | | RGB objects, triplet 2 | – | – | – | – | – | 57% |
| | | | | | RGB objects, triplet 4 | – | 58% | 71% | 66% | 68% | 72% |
| | | | | | RGB objects, triplet 5 | 34% | 58% | – | – | – | – |
| | | | | | RGB objects, triplet 6 | 59% | – | – | – | 44% | 73% |
| | | | | Pyramid building | | **RBG** | **RGB** | **GRB** | **GBR** | **BRB** | **BGR** |
| | | | | | RGB objects, triplet 1 | 71% | 69% | 52% | 64% | 79% | 75% |
| | | | | | RGB objects, triplet 2 | 63% | 61% | 51% | 44% | 73% | 55% |
| | | | | | RGB objects, triplet 3 | 66% | 59% | 62% | 68% | 48% | 58% |
| | | | | | RGB objects, triplet 4 | 61% | 55% | 67% | 61% | 80% | 78% |
| | | | | | RGB objects, triplet 5 | 71% | 48% | 77% | 68% | 75% | 77% |
| | | | | Lifting | NIST-i gears | **Small** | | **Medium** | | **Large** | |
| | | | | | | 86% | | 81% | | 72% | |
| | | | | Insertion-peg | NIST-i gears | 56% | | 79% | | 79% | |
| | **Real World** | | Sawyer 5-DoF | Stacking | | **RB** | **RG** | **GR** | **GB** | **BR** | **BG** |
| | | | | | RGB objects, triplet 1 | 87% | – | – | – | – | 45% |
| | | | | | RGB objects, triplet 2 | 70% | – | – | – | – | – |
| | | | | | RGB objects, triplet 3 | 82% | – | – | – | – | – |
| | | | | | RGB objects, triplet 4 | 93% | – | – | – | – | – |
| | | | | | RGB objects, triplet 5 | 68% | – | – | – | – | – |
| | | | | Lifting | YCB-i vegetables | **Carrot** | **Cucumber** | **Pepper** | **Potato** | – | – |
| | | | | | | 50% | 50% | 49% | 67% | – | – |
| | | | | Tower building | RGB objects, triplet 5 | 23% | | | | | |
| | | | | Inverted pyramid building | RGB objects, custom triplet | 17% | | | | | |
| | | | Panda 7-DoF | Lifting | YCB fruits | **Apple** | **Banana** | **Lemon** | **Peach** | – | – |
| | | | | | | 40% | 59% | 60% | 57% | – | – |
| | | | | Lifting | NIST-i gears | **Small** | | **Medium** | | **Large** | |
| | | | | | | 92% | | 94% | | 95% | |
| | | | | Insertion-peg | NIST-i gears | 65% | | 78% | | 88% | |
| | | | | Removal-peg | NIST-i gears | 96% | | 97% | | 98% | |
| **Fine-tuning** | **Real World** | | Panda 7-DoF | Insertion-bowl | YCB fruits and YCB-i bowl | **Lemon** | | **Peach** | | **Strawberry** | |
| | | | | | | 84% / 82% | | 76% / 86% | | 92% / 84% | |
| | | | | Removal-bowl | YCB fruits and YCB-i bowl | 60% / 62% | | 69% / 76% | | 64% / 77% | |
| | | | | Insertion-base | Shape-matching objects | **Circle** | | **Pentagon** | | **Square** | |
| | | | | | | 7% / 19% | | 6% / 10% | | 4% / 11% | |
| | | | | Removal-base | Shape-matching objects | 47% / 72% | | 87% / 94% | | 75% / 79% | |
| | | | KUKA 14-DoF | Lifting | NIST-i gears | 56% / 86% | | | | | |

Table 15: **Per-task RoboCat performance on training and fine-tuning tasks**. This table expands Table 1 by providing the success rate for each task variant instead of the average per task families. Fine-tuning results are reported for 500 and 1000 demonstrations respectively, separated by a slash.

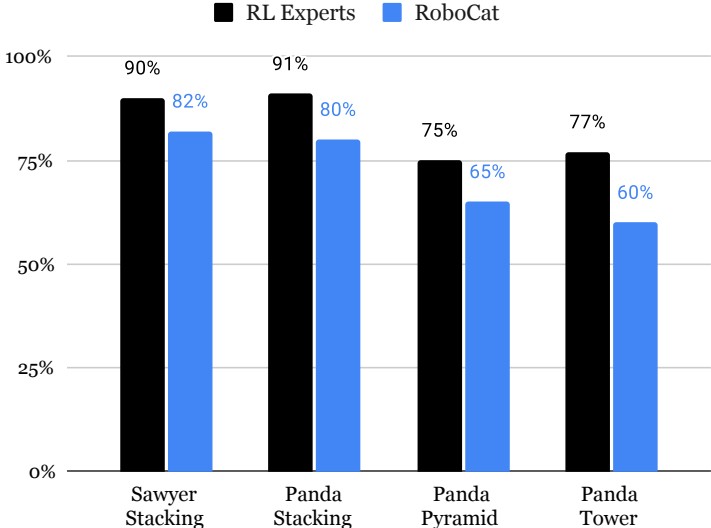

Figure 20: **RoboCat vs experts performance: Panda 7-DoF structure-building training tasks (sim).** RoboCat performance compared to the success rate of the training data for each task family.

failed self-generated episodes, but not failed demonstrations. On average RoboCat performance matches the success rate of the training data, overcoming it in some cases (see lifting banana, peach, strawberry Figure 24 and potato Figure 25).

### G.1.2 Fine-tuning performance

Figure 28 compares the success rate of the human teleoperators to contextualise the fine-tuning results. In general, we see how fine-tuning on more demonstrations in general improves performance, sometimes reaching similar performance as the human teleoperators. This does not apply to the insert tasks, which result very challenging for our agents due to the required reorientation and shape understanding.

### G.2 Impact of model size

To evaluate the effect of model size, we compare the 1.18B-parameter RoboCat-lim model from Section 5.2 with a smaller 364M model on the simulated structure-building tasks. The results in Figure 26 demonstrate that the larger model yields an improvement of 6-21% in the success rate for the different task families. In particular, while the smaller model only slightly degrades performance for the easier stacking tasks, it is much poorer for the more challenging pyramid and tower-building tasks.

### G.3 Visual foundation baselines

We consider four different network architectures as baseline methods for manipulation tasks: ResNet (He et al., 2016), normaliser-free networks (NFNets) (Brock et al., 2021), vision transformers (ViTs) (Dosovitskiy et al., 2020), and the ViT variant Swin transformers (Liu et al., 2021). Within each category, we used the following models: ResNet-50, ResNet-101, ResNet-200, NFNet-f0, NFNet-f3, NFNet-f6, ViT-b, ViT-l, Swin-b, Swin-l, and Swin-s. Each model was pretrained on a selection of datasets: ImageNet 1k (Deng et al., 2009), ImageNet 21k (Ridnik et al., 2021), Microsoft COCO (Lin et al., 2014), and the JFT Dataset (Riquelme et al., 2021). We also evaluated the use of CLIP (Radford et al., 2021), masked auto-encoder (MAE) (He et al.,

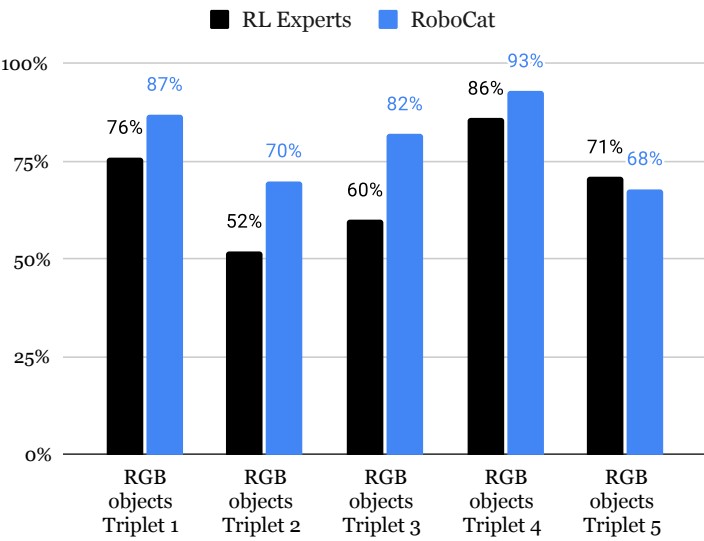

Figure 21: **RoboCat vs experts performance: Sawyer 5-DoF RGB stacking tasks (real).** This plot compares the performance of RoboCat on the real stacking tasks with respect to the overall success rate of the training data available for each task variant.

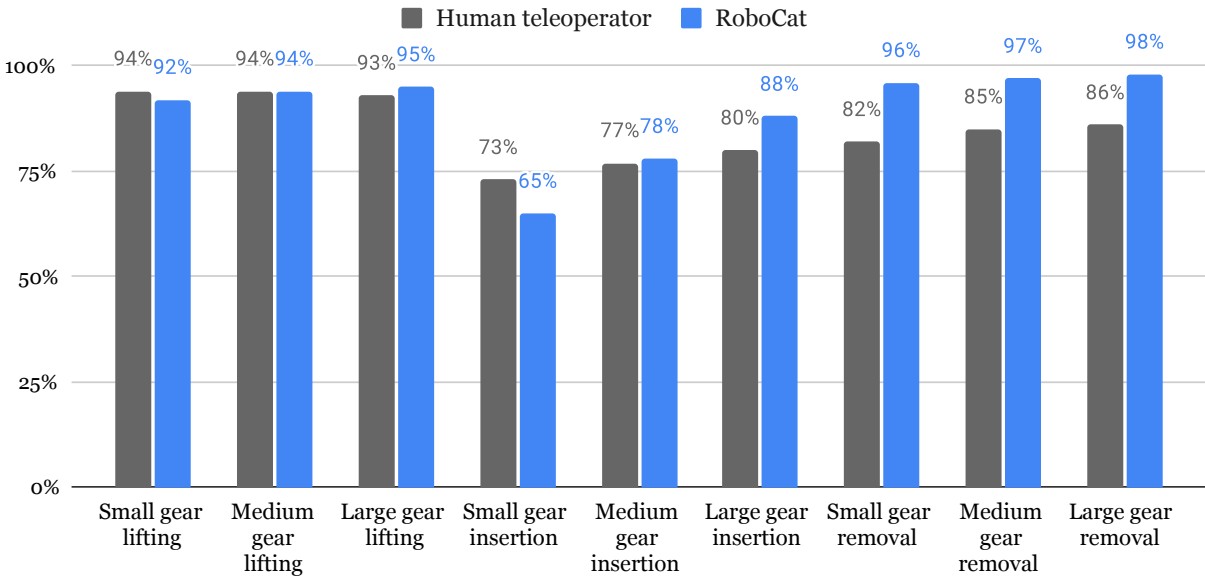

Figure 22: **RoboCat vs teleoperation performance: Panda 7-DoF NIST-i tasks (sim).** RoboCat compared with the success rate of the data collected by human teleoperators, for each task variant.

2022), SimCLR (Chen et al., 2020b), BYOL (Grill et al., 2020), DetCon (Hénaff et al., 2021), Odin (Hénaff et al., 2022), and DINO (Caron et al., 2021).

One common way to use pretrained visual models for control tasks is to add a learned policy head (Nair et al., 2022; Parisi et al., 2022). The image input is first embedded by the foundation model. The output embedding is then concatenated with any proprioception information and fed into the policy head. The policy head consists of two multi-layer perceptrons (MLPs), each with 256 parameters. The output of the top MLP is then used by a linear policy head to generate the robot action. While this means that the

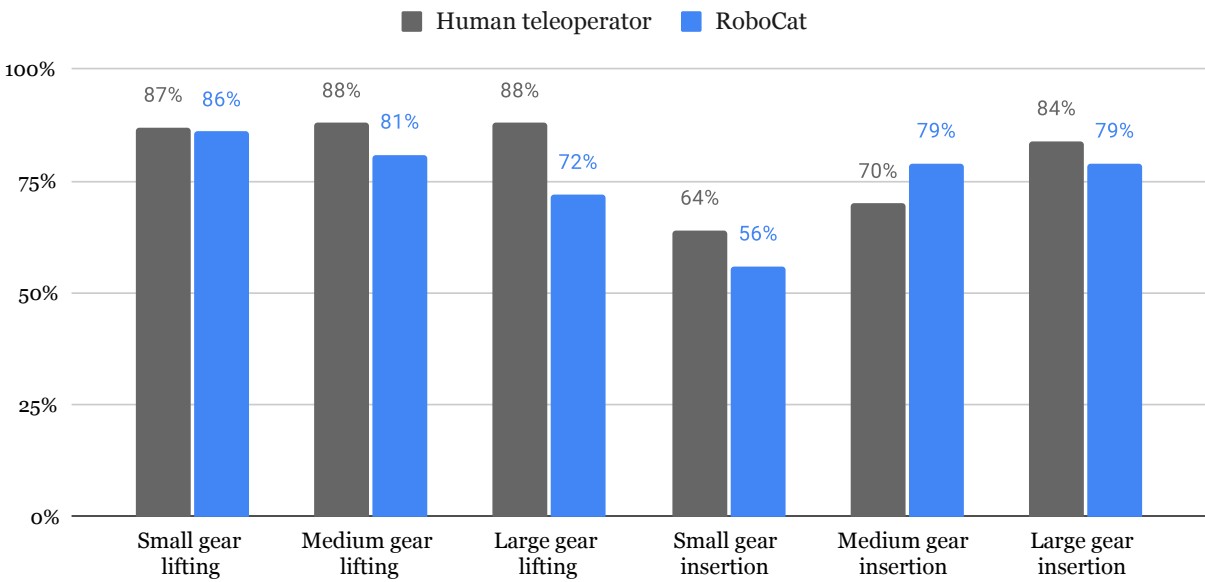

Figure 23: **RoboCat vs teleoperation performance: Panda 7-DoF NIST-i tasks (real).** RoboCat compared with the success rate of the data collected by human teleoperators, for each task variant.

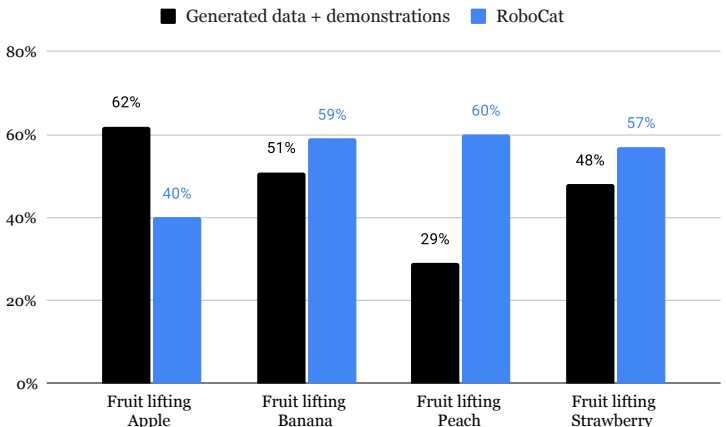

Figure 24: **RoboCat vs training data: Panda 7-DoF YCB lifting tasks (self-improvement, real).** This plot compares the performance of RoboCat on the self-improvement tasks with respect to the overall success rate of the training data available for these tasks (self-generated data and successful human teleoperations).

VFM baselines only use the current timestep (versus approximately 3 timesteps for the transformer-based RoboCat agent), prior experiments with stacking tasks found that incorporating context, via either LSTMs or observation stacking of 4 previous timesteps, did not improve performance. Finally, we use behaviour cloning with mean squared loss as the optimisation objective.

We fine-tuned a total of 59 combinations of the above methods and evaluated on a subset of RoboCat tasks. Then, we selected the top representatives of each of the network architecture categories and evaluated them on more tasks. For the sake of clarity, we only report the top two methods in the main paper. We use the average final policy success rate as the performance metric in our experiments. Each experiment ran for 100 episodes, with each episode lasting 400 time steps.

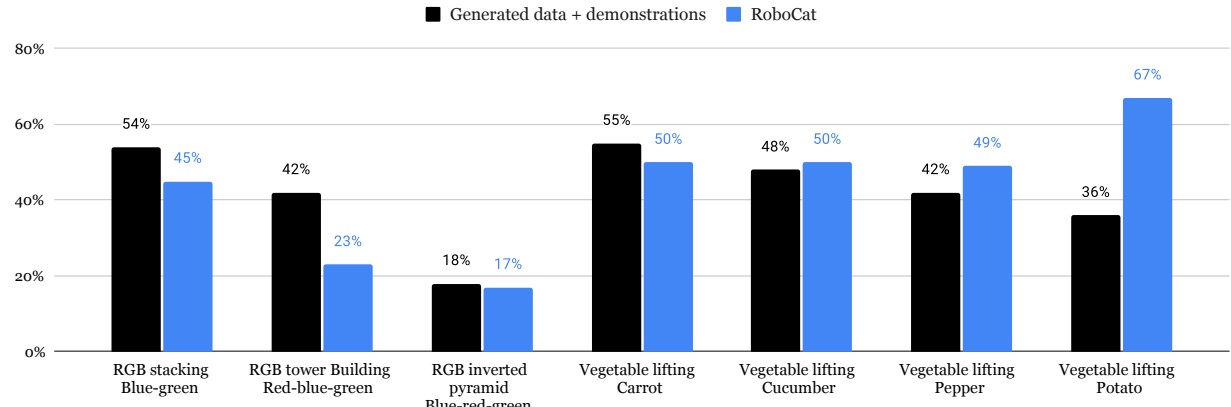

Figure 25: **RoboCat vs training data: Sawyer 5-DoF RGB and YCB tasks (self-improvement, real).** This plot compares the performance of RoboCat on the self-improvement tasks with respect to the overall success rate of the training data available for these tasks (self-generated data and successful human teleoperations).

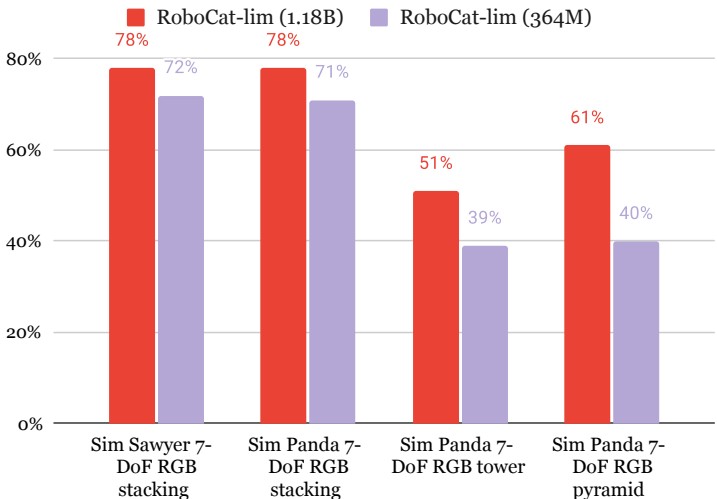

Figure 26: **Impact of model size.** This plot compares the duccess rate of a full-sized 1.18B-parameter model with that of a smaller 364M model, on the RoboCat-lim simulated structure-building tasks.

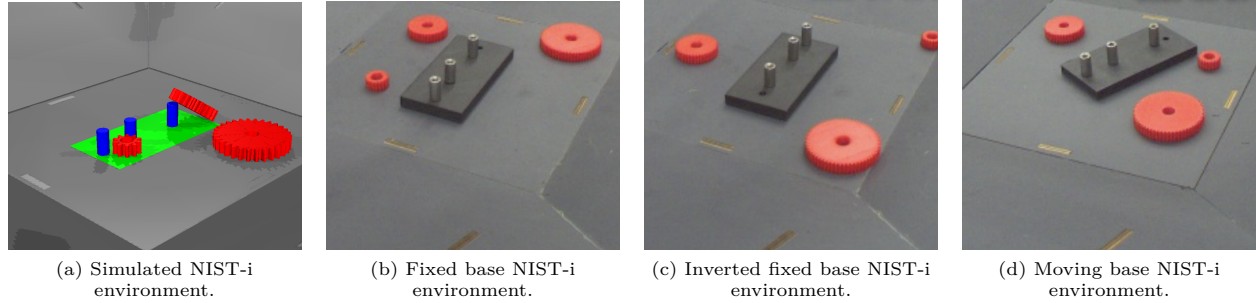

(a) Simulated NIST-i environment.

(b) Fixed base NIST-i environment.

(c) Inverted fixed base NIST-i environment.

(d) Moving base NIST-i environment.

Figure 27: **The different types of NIST-i based environments we ablate performance against.** Note, in the main paper we report performance against environments from (a) and (d) only.

We fine-tuned the models with different data limitations: 500 demonstrations, 1000 demonstrations, and more than 10 000 demonstrations. Training took 200 000 training steps for all models. We validated the models in simulation during training and selected the best model snapshot based on the validation success rate for

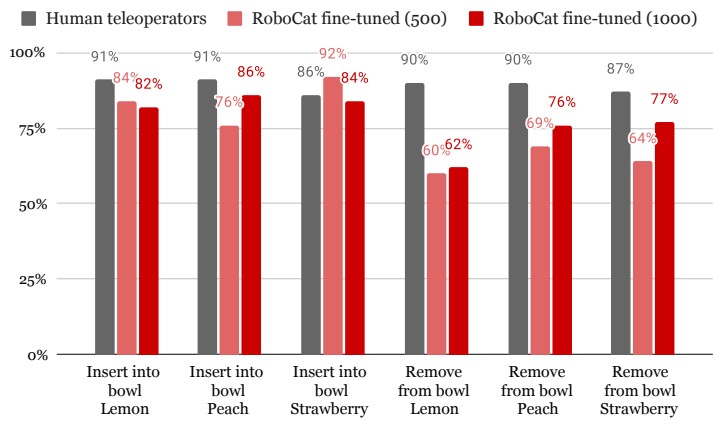

(a) **Panda 7-DoF insert and remove tasks with YCB and YCB-i objects (real).**

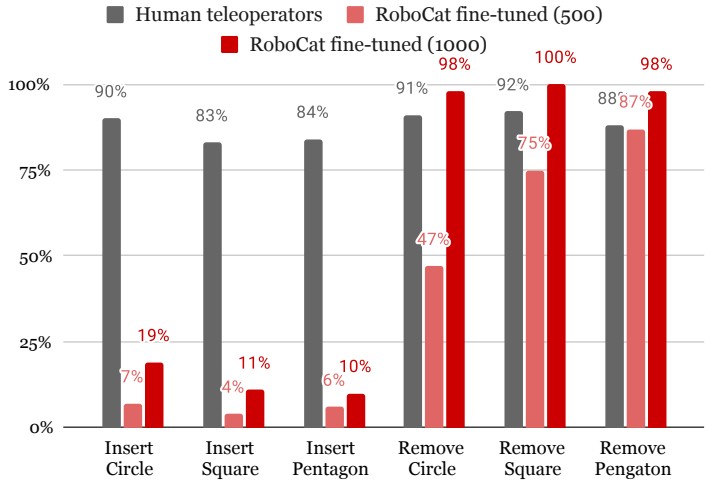

(b) **Panda 7-DoF insert and remove tasks with shape matching objects (real).**

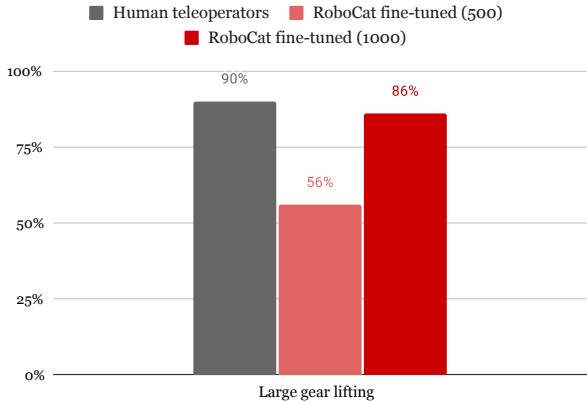

(c) **KUKA 14-DoF gear lifting task (real).**

Figure 28: **Final performance on each fine-tuning task**.

| model | obj | dataset | pretrain |
|---|---|---|---|
| nfnet-f0 | Supervised | ImageNet-1k | imagenet1k-supervised |
| nfnet-f0 | Supervised | JFT-4B | jft4b-8epochs-supervised |
| nfnet-f0 | CLIP | ImageNet-1k | f0-clip-4dataset-lowres |
| nfnet-f0 | CLIP | ImageNet-1k | f0-clip-align |
| nfnet-f0 | CLIP | ImageNet-1k | f0-clip-4dataset |
| nfnet-f0 | CLIP | ImageNet-1k | f0-clip-align-stock |
| nfnet-f1 | Supervised | ImageNet-1k | imagenet1k-supervised |
| nfnet-f1 | Supervised | JFT-4B | jft4b-8epochs-supervised |
| nfnet-f2 | Supervised | ImageNet-1k | imagenet1k-supervised |
| nfnet-f3 | Supervised | JFT-4B | jft4b-8epochs-supervised |
| nfnet-f3 | Supervised | ImageNet-1k | imagenet1k-supervised |
| nfnet-f3plus | Supervised | JFT-4B | jft4b-8epochs-supervised |
| nfnet-f4 | Supervised | ImageNet-1k | imagenet1k-supervised |
| nfnet-f5 | Supervised | ImageNet-1k | imagenet1k-supervised |
| nfnet-f5 | CLIP | ImageNet-1k | f5-clip-align |
| nfnet-f6 | CLIP | ImageNet-1k | f6-clip-align-stock-sum__grads |
| nfnet-f6 | CLIP | ImageNet-1k | f6-clip-4dataset |
| nfnet-f6 | CLIP | ImageNet-1k | f6-clip-align |
| nfnet-f6 | Supervised | ImageNet-1k | imagenet1k-supervised |
| nfnet-f7 | Supervised | JFT-4B | jft4b-4epochs-supervised |
| nfnet-f7plus | Supervised | JFT-4B | jft4b-4epochs-supervised |
| ResNet-101 | Supervised | ImageNet-1k | imagenet1k-supervised |
| ResNet-101 | DetCon | ImageNet-1k | imagenet1k-detcon |
| ResNet-101 | BYOL | ImageNet-1k | imagenet1k-byol |
| ResNet-200 | DetCon | ImageNet-1k | imagenet1k-detcon |
| ResNet-200 | BYOL | ImageNet-1k | imagenet1k-byol |
| ResNet-50 | ODIN | ImageNet-1k | imagenet1k-odin |
| ResNet-50 | Supervised | ImageNet-1k | imagenet1k-supervised |
| ResNet-50 | DetCon | ImageNet-1k | imagenet1k-detcon-coco-finetune |
| ResNet-50 | BYOL | ImageNet-1k | imagenet1k-byol-lowres |
| ResNet-50 | DetCon | ImageNet-1k | imagenet1k-detcon |
| ResNet-50 | BYOL | ImageNet-1k | imagenet1k-byol |
| swin-b | Supervised | ImageNet-1k | imagenet1k-supervised |
| swin-e | CLIP | JFT-4B | swin-e-clip-align-stock-jft |
| swin-h | CLIP | ImageNet-1k | swin-h-clip-align-stock-highres |
| swin-h | CLIP | ImageNet-1k | swin-h-clip-align-stock-lowres |
| swin-l | CLIP | ImageNet-1k | swin-l-clip-align-stock |
| swin-s | Supervised | ImageNet-1k | imagenet1k-supervised |
| swin-s | ODIN | ImageNet-1k | imagenet1k-odin |
| swin-t | ODIN | ImageNet-1k | imagenet1k-odin |
| swin-t | Supervised | ImageNet-1k | imagenet1k-supervised |
| vit-b | Supervised | JFT-4B | b16-224-jft-pretrain |
| vit-b | Supervised | ImageNet-21k | b16-224-i21k-pretrain-augreg |
| vit-b | Supervised | ImageNet-21k | b16-224-i21k-pretrain |
| vit-b | Supervised | ImageNet-1k | b16-224-i21k-pretrain-i1k-finetune |
| vit-b | Supervised | ImageNet-1k | b16-384-i21k-pretrain-i1k-finetune |
| vit-b | MAE | ImageNet-1k | b16-224-i1k-mae-pretrain |
| vit-b | Supervised | ImageNet-1k | b16-224-i1k-pretrain |
| vit-deit-b | MAE | ImageNet-1k | b16-224-i1k-mae-pretrain |
| vit-deit-l | MAE | ImageNet-1k | l16-224-i1k-mae-pretrain |
| vit-dino-b | MAE | ImageNet-1k | b16-224-i1k-mae-pretrain |
| vit-l | Supervised | ImageNet-1k | l16-224-i1k-pretrain |
| vit-l | Supervised | ImageNet-1k | l16-384-i21k-pretrain-i1k-finetune |
| vit-l | Supervised | ImageNet-21k | l16-224-i21k-pretrain |
| vit-l | Supervised | ImageNet-1k | l16-384-jft-pretrain-i1k-finetune |
| vit-l | MAE | ImageNet-1k | l16-224-i1k-mae-pretrain |
| vit-l | Supervised | JFT-4B | l16-224-jft-pretrain |
| vit-l | Supervised | ImageNet-1k | l16-224-i21k-pretrain-i1k-finetune |
| vit-mae-b | MAE | ImageNet-1k | b16-224-i1k-mae-pretrain |

Table 16: **Details of the baseline methods:** including the model name, training objectives and pretaining dataset.

further comprehensive evaluation. We tried both freezing and non-freezing scenarios for the parameters of the pretrained models. Since the non-frozen pretrained model performed much better, we only report the results for non-frozen models.

| | | final_success | | | | | | | |
|---|---|---|---|---|---|---|---|---|---|
| | **Model** | nfnet-f0 | nfnet-f3 | nfnet-f6 | swin-b | swin-l | swin-s | vit-dino-b | vit-l |
| | **Training Objective** | CLIP | Supervised | CLIP | Supervised | CLIP | Supervised | MAE | MAE |
| | **Dataset** | ImageNet-1k | JFT-4B | ImageNet-1k | ImageNet-1k | ImageNet-1k | ImageNet-1k | ImageNet-1k | ImageNet-1k |
| **# Demo** | **Task** | | | | | | | | |
| more than 10 000 | panda__7-DoF__stacking__BG_set__1 | 0.92 | 0.86 | 0.87 | 0.91 | 0.93 | 0.91 | 0.81 | 0.87 |
| | panda__7-DoF__stacking_RG_set__1 | 0.83 | 0.51 | 0.78 | 0.81 | 0.92 | 0.81 | 0.75 | 0.62 |
| | sawyer__5-DoF__stacking__set_5__RB | 0.79 | 0.79 | 0.78 | 0.79 | 0.87 | 0.76 | 0.71 | 0.70 |
| | sawyer__7-DoF__stacking__BG_set__1 | 0.83 | 0.80 | 0.88 | 0.91 | 0.90 | 0.84 | 0.77 | 0.82 |
| 1000 | panda__7-DoF__inv_pyra__GRB_set__4 | 0.00 | 0.00 | 0.00 | 0.00 | 0.00 | 0.00 | 0.00 | 0.00 |
| | panda__7-DoF__stacking__BG_set__1 | 0.25 | 0.40 | 0.60 | 0.24 | 0.49 | 0.20 | 0.02 | 0.07 |
| | panda__7-DoF__stacking__RG__set__1 | 0.20 | 0.31 | 0.33 | 0.25 | 0.28 | 0.25 | 0.03 | 0.12 |
| | panda__7-DoF__tower__GRB_set__4 | 0.00 | 0.00 | 0.00 | 0.00 | 0.00 | 0.00 | 0.00 | 0.00 |
| | panda__7-DoF__tower__RBG_set__5 | 0.04 | 0.08 | 0.14 | 0.08 | 0.16 | 0.06 | 0.01 | 0.03 |
| | sawyer__5-DoF__stacking__set_5__RB | 0.66 | 0.76 | 0.74 | 0.62 | 0.70 | 0.43 | 0.24 | 0.30 |
| | sawyer__7-DoF__stacking__BG_set__1 | 0.39 | 0.50 | 0.56 | 0.36 | 0.58 | 0.36 | 0.06 | 0.11 |
| 500 | panda__7-DoF__stacking__BG_set__1 | 0.09 | 0.19 | 0.36 | 0.08 | 0.06 | 0.09 | 0.01 | 0.03 |
| | sawyer__5-DoF__stacking__set_5__RB | 0.33 | 0.55 | 0.64 | 0.18 | 0.16 | 0.08 | 0.03 | 0.05 |
| | sawyer__7-DoF__stacking__BG_set__1 | 0.10 | 0.24 | 0.37 | 0.13 | 0.20 | 0.14 | 0.01 | 0.02 |
| training set | panda__7-DoF__pyramid__GRB_set__3 | 0.28 | 0.28 | 0.35 | 0.27 | 0.33 | 0.21 | 0.15 | 0.19 |
| | panda__7-DoF__stacking__BG_set__1 | 0.63 | 0.69 | 0.70 | 0.62 | 0.73 | 0.58 | 0.41 | 0.46 |
| | panda__7-DoF__stacking__GB_set__2 | 0.48 | 0.61 | 0.64 | 0.52 | 0.56 | 0.50 | 0.38 | 0.41 |
| | panda__7-DoF__tower__GRB_set__5 | 0.37 | 0.40 | 0.55 | 0.37 | 0.42 | 0.40 | 0.22 | 0.31 |

Table 17: **Evaluation of baselines on training and held-out tasks with limited number of demonstrations.**

Table 17 shows the success rate of all baseline methods on a held-out task. Then we took the top representative methods of each network architecture and further evaluated them on other tasks with limited data: 500, 1000, and more than 10 000. We also evaluated these methods on a subset of the training tasks.

### G.4 NIST-i extended study

We evaluate the complexity and some design decisions made for the NIST-i tasks using RoboCat. We find that fixing the base in the physical world can have interesting implications, more cameras can lead to improved performance and using data from largely unrelated tasks can lead to increased performance on NIST-i, suggesting indications of positive skill transfer.

#### G.4.1 Task complexity

So far, we considered a fixed base in simulation and a freely moving base in the physical world (see Figure B.1 for details). We argue that a moving base leads to both a harder task in the physical world and provides a more diverse set of tasks for training. We challenge this argument in this section. We do this by fixing the base to the basket in the same orientation as what we do for simulation and measure if this new setting is easier to solve. Figure 27 illustrates the different environments. Note that at train time we only use demonstrations from Figure 27(a) and Figure 27(d).

| Base Location | NIST-i task | |
| --- | --- | --- |
| | Insert | Remove |
| Fixed base | **64%** | **100%** |
| Inverted fixed base | 40% | 99% |
| Moving base | 38% | 97% |

Table 18: **Real-world NIST-i task ablation.** Average success over all three NIST-i gear sizes. A 364M agent performs much better on the fixed base environment.

Table 18 reports the performance of a smaller, 364M agent, trained on the same data as RoboCat-lim. We observe that despite not having fixed base data from the physical world, the agent performs a lot better on the fixed-based task as opposed to the other two tasks. This suggests that the fixed base environment is easier possibly due to its similarity with the simulated environment, suggesting positive skill transfer from simulation data.

#### G.4.2 Data bias

Appendix G.4.1 showed that there may be some positive transfer across similar tasks on a smaller 364M parameter agent. However, the performance reported in that section could also be due to data bias. That is, it could be that the collected demonstrations from raters might be primarily centred around the middle of the basket with the peg base orientated similarly to the one in the simulated data. This itself would interfere with the hypothesis for positive skill transfer in favour of having data bias. To test this, we utilise specialist behaviour cloning agents (Specialist BC) that we separately train on each NIST-i task using only the subset of NIST-i training data belongs to that specific task. Behaviour cloning agents are known to overfit to the training data, leading to poor generalisation (Osa et al., 2018). If the reported performance in Table 18 is due to data bias, then we expect to see the same type of improved performance of a specialist BC agent when evaluated on the fixed based tasks as opposed to the moving base tasks.

| Base Location | Specialist BC |
| --- | --- |
| | NIST-i Insert Task |
| Fixed base | 8% |
| Moving base | **13%** |

Table 19: **Real-world NIST-i data bias.** Average success over all three NIST-i gear sizes. The agents have similar perform on both fixed and moving base environments. No evidence for data bias.

| NIST-i Insert Task | 364M error | RoboCat error | Reduction factor |
|---|---|---|---|
| Fixed base | 0.36 | **0.13** | 2.77 |
| Moving base | 0.62 | **0.23** | 2.70 |

Table 20: **Skill transfer analysis.** Average accumulated error over all three NIST-i gear sizes. Moving from the 364M model to the full RoboCat agent eliminates just under two thirds of the gear insertion failures in both the fixed and the moving base settings, indicating a smaller performance gap for the RoboCat.

Table 19 shows that having a fixed or moving base results in a negligible difference in the performance for the specialist BC agents. In fact, the relationship between the performance on fixed and moving base is inverted (all training data is collected while having a moving base). This indicates that for both the fixed and moving base tasks there is no particular data bias towards the peg base being situated in the centre and oriented in the same way as in simulation.

### G.4.3 Skill transfer

So far, we showed that there are some indications of skill transfer for the smaller 364M generalist. In this section, we study to what extent this holds for our final RoboCat agent. Notably, we use the same NIST-i insertion data for training both the 364M and RoboCat models. However, our final version of RoboCat is trained on a larger number of tasks and is of bigger capacity (1.2B). In this sense, a successful skill transfer would close the gap between the achieved performance on the fixed and moving base environments for the final generalist. This is what we observe in Table 20. Specifically, moving from the 364M model to the full RoboCat agent eliminates just under two thirds of the gear insertion failures in both the fixed base and the moving base setting. This indicates that indeed training a bigger model on more data that involves different types of bring to pose tasks can be beneficial for solving the moving base insertion tasks. Moreover, the additive performance gap for RoboCat depending on the base state is much smaller than the additive performance gap for the 364M agent suggesting again indications of positive skill transfer.

### G.4.4 Additional observations

Finally, in the process of finding the best performing specialist BC agent we noticed that including the wrist cameras to the observation of the agent significantly improved the performance of our specialists. That is, instead of using just two camera observations, as we do for RoboCat, using a total of four camera observations - two from the basket and two from the wrist of the robot, was very beneficial for a specialist BC agent (see Table 21).

| Performance of | NIST-i task | |
|---|---|---|
| | Insert | Remove |
| Specialist BC (2 cameras) | 13% | 24% |
| Specialist BC (4 cameras) | **36%** | **54%** |

Table 21: **Dependency on the number of cameras.** Average performance over all three NIST-i gear sizes. More cameras have a significant effect on the specialist's performance.

An exciting direction for future research is to understand the effect of different camera observations for contact-rich manipulation tasks in the context of RoboCat.

