# OpenReview forum: "RoboCat: A Self-Improving Generalist Agent for Robotic Manipulation"
_TMLR — Accepted by TMLR_

### Review · Reviewer_68NV · 2023-09-18

**Summary Of Contributions:**

The paper proposes a system for training several robots with different embodiments and action spaces that autonomously improves through interaction with the environment and benefits from shared experience across robots. Training on different embodiments is handled by tokenizing them to embed them into the same observation and action space. Autonomous improvement is done via goal-conditioned supervised learning (Ghosh'19). There are experiments on three different robots and 2 different grippers, all in the same lab setup. The experiments demonstrate training on data from different robots together is better than training on a single robot only.

**Audience:**

Yes

**Broader Impact Concerns:**

\-

**Claims And Evidence:**

Yes

**Requested Changes:**

- It is unclear whether the paper proposes a foundation agent or a first step towards a foundation agent. The title claims the former but the intro claims the latter:
> Inspired by foundation models in other domains (Bommasani et al., 2022), we ultimately aim for a foundation agent for manipulation to be a multi-embodiment agent trained on a large set of robotic episodic experience that enables it to quickly adapt, via fine-tuning, to
a broad set of new downstream tasks. As a step towards this, we trained RoboCat...

This needs to be clarified.
- Fig 1 proposes a perpetually self-improving agent. As far as I can tell, only one iteration of that cycle was ever executed, which makes the figure misleading. This needs to be clarified.

**Strengths And Weaknesses:**

Strengths
- The proposed method is sound and straightforward.
- The experiments demonstrate positive transfer across robots, which has been challenging historically.

Weaknesses
- The paper claims to demonstrate a foundation agent. This is an overstatement that goes contrary to the definition of a foundation agent used in this paper. A large number of demonstrations (500 - 1000) is required to adapt to a new embodiment and there is no zero-shot transfer to new embodiments. Furthermore, all embodiments demonstrated in the paper are very similar. The proposed system will likely not generalize to new backgrounds or bin textures. Such generalization is important for a foundation agent since it is impractical to perfectly replicate the setup in this paper in another lab.
- The writing is confusing. Figure 1 demonstrates the fine-tuning process but it's unclear what is the original agent that is being fine-tuned. Section 2.1.1 on "pretraining" explains how the encoder is pretrained but not the agent itself. The section 2.1.2 on fine-tuning does not talk about pretraining. A training and fine-tuning split is later mentioned in the experimental section, from which we could guess what pretraining looks like. Most experiments do not contain the cycle in Fig 1, and only in one experiment the cycle is executed once (but not more). Overall, Fig 1 is not helpful for understanding what was done in the paper.
- The legend in Fig 5 is rather confusing. It looks like pretraining with CLIP objective significantly hurts performance. It would seem to me that what actually is being compared is multi-task vs single task training, which is a much less surprising result.
- There are relevant papers missing: Dasari'19, Hu'22
- The broader impact statement is confusing. There is significant discussion of AGI safety risk which is very confusing given that current system has a 14% failure rate on gear lifting in a lab environment. Perhaps it would be useful to put this discussion in context.

Dasari'19, RoboNet: Large-Scale Multi-Robot Learning

Hu'22, Know Thyself: Transferable Visual Control Policies Through Robot-Awareness

---

> ### Author Response · Authors · 2023-10-04
> **Response to Reviewer 68NV (1/2)**
>
> **_The proposed method is sound and straightforward...The experiments demonstrate positive transfer across robots, which has been challenging historically._**
>
> Thank you for recognising the contributions of our work and for the valuable feedback! We have added a new revision of the paper and highlighted the changes that are relevant to this review, in blue. We hope that this will contribute to an easier and clearer discussion. We address your comments below.
>
> **_Training on different embodiments is handled by tokenizing them to embed them into the same observation and action space._**
>
> A minor clarification: we do not use a common space for actions and observations, and indeed, our different embodiments all have different dimensions for action (5DoF, 7DoF, and 14DoF) and observations. While the image sizes are fixed, the proprioception state is different for each embodiment. The transformer-based RoboCat model is able to input/output variable-length sequences based on context. We’ve further clarified this in the introduction.
>
> **_The paper claims to demonstrate a foundation agent. This is an overstatement that goes contrary to the definition of a foundation agent used in this paper [due to the number of demonstrations used for a new embodiment, the similarity between embodiments, and the lack of variation of backgrounds/bin textures/etc]._**
>
> Thank you for this feedback, and we’ve now modified the text in multiple places to clarify this. As in your review, our proposed definition in the introduction mentions that "we ultimately aim for a foundation agent for manipulation to be a multi-embodiment agent trained on a large set of robotic episodic experience that enables it to quickly adapt, via fine-tuning, to a broad set of new downstream tasks." This is a more specific and stricter version of the original definition in (Bommasani et al, 2022) stating that a “foundation model is any model that is trained on broad data (generally using self-supervision at scale) that can be adapted (e.g., fine-tuned) to a wide range of downstream tasks”. To our knowledge, RoboCat is the first robotic generalist to handle different embodiments natively (as clarified in the previous point), and demonstrate quick finetuning capabilities to a diverse set of manipulation tasks. We hope that next-generation foundation agents will also enable robustness of the kind mentioned in your review, such as different bin textures and in-the-wild settings. Instead of focusing on this form of visual diversity, RoboCat deals with significant behavioural diversity. We’ve now clarified this in the introduction, and added to the future work to openly discuss these limitations of the RoboCat agent.
>
> **_The writing is confusing, and Fig 1 is not helpful for understanding what was done in the paper, [especially in terms of the relationship between pre-training, finetuning, and the improvement cycle]_**
>
> Thank you for flagging this confusion! We have now added an explanation at the start of Section 2 to clarify this nomenclature before the different stages are introduced. The pre-training phase refers to training the VQ-GAN tokenizer, training refers to the main training of our RoboCat generalist, and this generalist is then finetuned with data from a given task for the finetuning phase.
>
> We have also modified the caption for Figure 1 to explicitly mention the scope of the experiments. To summarise, the experiments in this paper are designed to demonstrate the benefits of self-improvement; via one full-cycle of finetuning, data collection, and retraining with self-generated data. Figure 1 is intended as a motivating factor for this process and an illustration of what a self-improving system looks like.
>
>
> **_The legend in Fig 5 is rather confusing. It looks like pretraining with CLIP objective significantly hurts performance. It would seem to me that what actually is being compared is multi-task vs single task training, which is a much less surprising result._**
>
> The goal of Figure 5 is in part to compare the RoboCat multi-task generalist agent with single-task specialists. As detailed in Section 4.4, these were obtained by trialling 59 different Visual Foundation Model baselines (pre-trained on non-robotics data and finetuned to a given task), and choosing the two that performed best on average over the entire set of tasks.
> However, the experiment also demonstrates the utility of a generalist agent trained on robotics data: it shows that RoboCat is significantly better at real tasks where data is limited and at least as good as the specialist agents in simulation where data is abundant.
>
> We have further clarified this by changing the caption for Figure 5 and explicitly pointing to Section 4.4 where we provide a detailed discussion.

---

> > ### Author Response · Authors · 2023-10-04
> > **Response to Reviewer 68NV (2/2)**
> >
> > **_There are relevant papers missing: Dasari'19, Hu'22_**
> >
> > Thank you for pointing us to them, we have gone through them carefully and added them to the related work section.
> > Dasari et al. (2019) introduce a large-scale robotics dataset of pick-and-place-based manipulation tasks and demonstrate robotic agents that can be effective across different embodiments and environments by using shared observation and action spaces.
> > Hu et al. (2022) aim to address the cross-embodiment transfer problem by factorizing video prediction into an analytical robot-module and a learned non-robot module. This allows the learned video prediction to be reused with a new robot to do visual mpc. Our work differs in that we do not use video prediction for planning and instead directly learn the action prediction for all embodiments jointly.
> >
> >
> > **_The broader impact statement is confusing. There is significant discussion of AGI safety risk which is very confusing given that current system has a 14% failure rate on gear lifting in a lab environment. Perhaps it would be useful to put this discussion in context._**
> >
> > Thank you for pointing this out. We believe that we have understated the intention of our broader impact statement and have therefore improved the prose to improve clarity and address your concerns.
> >
> > Specifically, we have updated the text to more explicitly point out that the experiments with the current system are still in a controlled lab environment with imperfect performance. We have also made it clearer that our intention with the broader impact statement is to discuss the longer-term consequences of this line of research into large-scale, multi-task, robotic generalist agents operating and improving in the real world. This should be beyond the immediate impact and specific contributions of this paper.

---

> > ### Comment · Reviewer_68NV · 2023-10-12
> > **The claims are misleading**
> >
> > I do not believe the changes address all of my comments.
> >
> > Both definitions of "foundation models" mentioned by authors require generalization to a "broad set of downstream tasks". This paper instead evaluates on only two new downstream tasks (insertion, removal) and one new downstream robot (note that the new robot is not evaluated on new tasks and the new tasks are not evaluated on the new robot). I disagree that this constitutes a new foundation agent and I believe this claim should be removed from the paper.
> >
> > Fig 1. Is still not helpful in understanding what the paper does. Where does the loop start?
> >
> > On page 4, it is stated "To train the agent model we use a dataset Dˆ containing the joint collection of data from all tasks". Which tasks? ImageNet or the ones from Reed'22?
> >
> > The legend in Fig 5 is still confusing.

---

> > > ### Author Response · Authors · 2023-10-17
> > > **2nd Followup response to Reviewer 68NV**
> > >
> > > *__I do not believe the changes address all of my comments.__*
> > >
> > > Thank you for your reply. We are eager to make sure we address all your comments and concerns, as we see them as crucial to strengthening this submission. We appreciate your communication and persistence, which has resulted in significant changes to the paper since the last revision. We address your other comments in detail, below.
> > >
> > > *__Both definitions of "foundation models" mentioned by authors require generalization to a "broad set of downstream tasks". This paper instead evaluates on only two new downstream tasks (insertion, removal) and one new downstream robot (note that the new robot is not evaluated on new tasks and the new tasks are not evaluated on the new robot). I disagree that this constitutes a new foundation agent and I believe this claim should be removed from the paper.__*
> > >
> > > Thanks for re-iterating this point in more detail, as it allows us to clarify this claim and add details that we now see were not clear in the text. In summary, although the final version of our model is evaluated on a total of 13 finetuning tasks, we also used an earlier version, RoboCat-lim, to extensively study generalisation and adaptation on a total of 32 finetuning tasks along various axes. We have now clarified this when discussing the tasks used for fine-tuning (in the introduction, method, and experimental setup). We have also modified the conclusion to better justify this claim later in the text. We hope this addresses your concerns.
> > >
> > > In more detail:
> > >
> > > The benefit of the self-improving agent is that one iteration of the generalist can be fine-tuned to unseen tasks, and those tasks can be absorbed into the main training data for the next iteration. This also means that the final model (trained on all of the tasks which have sufficient data) will necessarily have fewer held-out tasks to evaluate, and that the numerous finetuning and adaptation tasks from the limited-task version of the model form part of the basis for our claim.
> > >
> > > The final RoboCat model is evaluated on a total of 13 finetuning tasks. These consist of 5 task families with some variations: bowl insertions & removals, shape-matching insertions & removals, and gear lifting with an unseen embodiment (see Table 1 for details and Fig 4 to visualise the difference between the different insertions). We also want to highlight that even though the behaviour we demonstrate with the new embodiment is a seen one, the different appearance and action space needed - 14 dimensions vs 5 and 7 in our seen embodiments - essentially make this a new skill: we’d need to remap perception to this new 14-dim action space, and it is particularly difficult as also demonstrated with our baselines.
> > >
> > > However, we have also extensively studied generalisation and adaptation with an earlier version of the model, which we call RoboCat-lim (same architecture and size, but with a subset of the tasks). RoboCat-lim was evaluated on a total of 32 finetuning tasks: 23 of those were part of our adaptation study, discussed in Sec 5.2 (see Fig. 7), and 9 were part of the self-improvement process discussed in Sec 5.3 (Fig. 11). As part of the adaptation study we investigated adaptation along different axes (eg unseen perceptual variations and objects, new task families, different data sources, sim-to-real), which also included the case of a new embodiment with a new task: lifting gears with the KUKA embodiment, neither of which are in the RoboCat-lim training set (see Fig 7a and b).
> > >
> > > *__Fig 1. Is still not helpful in understanding what the paper does. Where does the loop start?__*
> > >
> > > The loop starts with an initial dataset used to train the first iteration of RoboCat. We have clarified this in the caption and updated the figure to show the starting point.
> > >
> > > *__On page 4, it is stated "To train the agent model we use a dataset Dˆ containing the joint collection of data from all tasks". Which tasks? ImageNet or the ones from Reed'22?__*
> > >
> > > This refers to the entire set of tasks which is introduced in Section 3; we have updated this sentence to clarify this.
> > >
> > > *__The legend in Fig 5 is still confusing.__*
> > >
> > > Thanks for raising this. Our understanding from your previous comment was that it was unclear whether this figure was about single-task versus multi-task training or something else, and as such we attempted to address this point (ie. it is about multi-task versus single-task, as well as the utility of a generalist agent trained on robotics data).
> > >
> > > Perhaps the confusion is instead from the mention of CLIP at all in the legend, which is descriptive of the baselines, but not the important discriminating factor here? To address this, we have now removed this and refer to the baselines as Swin and NFNet in all Figures. We hope this addresses your concerns; if not then please do clarify.

---

> ### Comment · Reviewer_68NV · 2023-10-23
> **Still disagree with naming**
>
> Thank you for the additional clarifications. I think the changes make the paper easier to understand.
>
> However, I disagree with the naming "foundation agent" despite the clarifications. Bommasani'21 describes emergence and homogenization as two key attributes of a foundation model.
> > Emergence means that the behavior of a system is implicitly induced rather than explicitly constructed; ... Homogenization indicates the consolidation of methodologies for building machine
> learning systems across a wide range of applications.
>
>  Bommasani et al. provide an example of homogenization, in which a single model (BERT) was used by essentially the majority of the work in a field (NLP).
>
> > Before 2019, self-supervised learning with language models was essentially a _subarea_ in NLP, which progressed in parallel to other developments in NLP. After 2019, self-supervised learning with language models became more of a _substrate_ of NLP, as using BERT has become the norm.
>
> The submission discusses neither emergence nor homogenization. I do not believe that the proposed paper can become a substrate for robotics - indeed since the current system is evaluated in only one lab, there is no reason to think that another lab could beneficially use it. Such transfer is however key to the definition of a foundation model in Bommasani'21.

---

> > ### Author Response · Authors · 2023-10-25
> > **We have changed the naming and wording**
> >
> > Thanks for your detailed response, and for your continued engagement with our paper - it has helped to improve its clarity and impact.
> > With this additional context, we feel that we better understand your concerns. We have changed the title to “generalist agent”, and changed the abstract, intro, and conclusions to mention the capabilities of foundation agents as a source of inspiration and end-goal rather than an explicit claim of this paper. We hope this addresses your points.

---

> > > ### Comment · Reviewer_68NV · 2023-10-26
> > > **Acknowledged**
> > >
> > > I believe the new naming is more precise and makes the connections to existing work clearer.

---

### Review · Reviewer_qZkA · 2023-10-03

**Summary Of Contributions:**

The paper proposes a large transformer model to be used as a foundation for robotic control for a variety of different manipulation tasks on multiple different arm embodiments with different control spaces. Such a foundation model is shown to require only a small number of demonstrations, to learn new tasks through finetuning. Additional synthetic data generated from the model labeled via hindsight relabeling can improve finetuning performance, which the paper calls self-improvement. Results are validated both in simulation and in substantial real-world experiments.

**Audience:**

Yes

**Broader Impact Concerns:**

This is well addressed in the section on Broader Impact already and I don't have additional concerns. I would like to add that this paper would be of particular significance if the model were made available openly, though I don't see any promise of that in the text.

**Claims And Evidence:**

No

**Requested Changes:**

- Please answer the first few questions in the weaknesses section, and depending on the answers add a bar for smaller-sized RoboCat model sizes for comparisons in Figures 5, 6 and 8. I don't have an issue with the larger model being important to the results, but I just don't understand if it is and that is critical.
- Add a same data, different architecture baseline and a different data, same architecture baseline, this seems critical to understanding the different components of the method. That or the phrasing of the contribution here will need to be more careful. I see the paper as one of a handful of demonstrations of a foundation model in a control context, and one that can be used with different embodiments. If the paper is largely a statement about the fact that large, diverse data + large sequence models also works in control that is totally fine, yet I am uneasy about the presentation of clear wins over baselines without the delineation of what makes the method work in the first place.
- Consider adding VIMA as a baseline or discuss why it is not comparable. This seems critical as it is much closer to the application domain than the chosen visual transformers, and it is open-source (https://github.com/vimalabs/VIMA).

**Strengths And Weaknesses:**

Strengths:
- Simple and clear approach to the problem
- Large number of complex, visual control tasks tested
- First large-scale multi-embodiment agent that I'm aware of, really cool and of definite interest to the community given the trend toward large models that others can build on
- Low expert data requirements for new tasks is ideal, though synthetic collection on real robots is still far from trivial
- Clear prose

Weaknesses:
- Details about baselines unclear/missing comparisons:
  - In section 4.2 the full model size is given at 1.18B and for some ablations 346M, while in 4.4 the baselines are listed at 438M and 197M parameters. Is the comparison in Figures 5, 6 and 8 with the smaller or larger RoboCat model? Could you guide me to text describing exactly which experiments use which version of RoboCat? It seems Figures 10 and 11 are the smaller and Table 20 compares between the two sizes, so the rest would be the larger, but I am not positive.
  - The method uses ~3 timesteps of conditioning (Section 2.1.1), while in Appendix G.2 it's stated a "common way to use... robot action," but this claim does not make it clear if only a single image is used for baseline methods or also a history of a few images (and whether this makes a difference).
  - Baseline methods are built on existing models pretrained from a different data sources that do not include images close to the deployment task, unlike the proposed method. That would lead me to conclude that Figures 5, 6 and 8 are really a statement about data, which leads me to wonder about a setting with exactly the same architecture but different data source (language is a prime candidate, discussed below), as well as a different architecture with the same data source (NFNet with Section 3.4 data).
  - There seem to be good already existing, and cited, baselines close to the tasks considered like VIMA (https://arxiv.org/pdf/2210.03094.pdf) and GATO, I'm a bit confused as to why these were not chosen. VIMA especially as it is open-source. There is only a single citation with no discussion as to why it wouldn't make sense to compare in Section 6.1.
  - If my reading is correct that there isn't a directly comparable baseline method with the same data and different architecture, is there some particular justification for the VQGAN + autoregressive transformer? I can see some possible interest in in-context learning, but contexts seem too short to leverage that (unlike a method like VIMA that condenses observations to single tokens).
- From above, I think this paper missed an opportunity to make a comparison to finetuned language models (using the same VQGAN tokens as the proposed method as input). All baselines are pretrained visual foundation models which may be adept at reasoning over the content in the image, but less adept at reasoning over the history. It seems pretty straightforward, and supported in the literature to use language models as a base for control (https://arxiv.org/abs/2103.05247) given that they see an enormous variety of sequences in their data.
- The use of" 0-shot" and "k-shot" when referring to finetuning in Section 5.2 is a bit confusing. I would take this terminology to refer to in-context learning, not the size of the dataset for further training. I can understand why in-context learning wasn't explored here, given that it takes 1024 tokens to model only 3 steps of time, but on a first read I did not parse this.

---

> ### Author Response · Authors · 2023-10-04
> **Response to Reviewer qZkA (1/2)**
>
> **_First large-scale multi-embodiment agent that I'm aware of, really cool and of definite interest to the community given the trend toward large models that others can build on_**
>
> Thank you for recognising the contributions of our work, and we appreciate the constructive feedback! We have added a new revision of the paper and highlighted the changes that are relevant to this review, in red. We hope that this will contribute to an easier and clearer discussion, and address your comments below.
>
>
> **_In section 4.2 the full model size is given at 1.18B and for some ablations 346M, while in 4.4 the baselines are listed at 438M and 197M parameters. Is the comparison in Figures 5, 6 and 8 with the smaller or larger RoboCat model? Could you guide me to text describing exactly which experiments use which version of RoboCat? It seems Figures 10 and 11 are the smaller and Table 20 compares between the two sizes, so the rest would be the larger, but I am not positive._**
>
> Thanks for raising this and helping to clarify the paper. All of the RoboCat numbers are with the larger model size, with the exception of Figure 10, which was an ablation performed with the smaller models, and experiments in Appendix G.3, which perform various ablations with different-sized models on NIST-i tasks. We have now clarified this upfront in the text in section 4.2.
>
> Figures 5, 6, and 8 use the full RoboCat model (1.18B parameters) and the baselines are indeed 438M and 197M parameters. The difference in size is because (1) the RoboCat model needs to deal with the multi-task setting with hundreds of tasks, while the baselines are single-task; and (2) the baselines were obtained via finetuning existing VFM architectures, limiting the flexibility in size. We reiterate that we performed an extensive search to find the best-performing baselines (with 59 candidate architectures), and we have now added this explanation for the model size in Section 4.4.
>
>
> **_The method uses ~3 timesteps of conditioning (Section 2.1.1), while in Appendix G.2 it's stated a "common way to use... robot action," but this claim does not make it clear if only a single image is used for baseline methods or also a history of a few images (and whether this makes a difference)._**
>
> The Gato baseline has the same context length of 3 timesteps, but the VFM baselines and BC-IMP (for the RGB-Stacking benchmark) only use a single image. In past experiments with stacking tasks, we employed observation stacking of up to 4 previous timesteps to allow for context, and experimented with LSTM architectures, but found that both made little difference to the performance. One of the benefits of our transformer-based approach is indeed to provide a principled method of considering history (i.e. the context). We have now clarified this in Section G.2 at the point you referenced in your review.
>
>
> **_Baseline methods are built on existing models pretrained from a different data sources that do not include images close to the deployment task, unlike the proposed method. That would lead me to conclude that Figures 5, 6 and 8 are really a statement about data, which leads me to wonder about a setting with exactly the same architecture but different data source (language is a prime candidate, discussed below), as well as a different architecture with the same data source (NFNet with Section 3.4 data)._**
>
> This is a good point, and the reason we also compare with the Gato baseline (Reed et al, 2022) for all of the robotics tasks which were used in their paper (RGB stacking mastery benchmark; and blue-on-green stacking in the finetuning setting). These comparisons appear in Section 5.1 and 5.2.
> We have now clarified upfront in Section 4.4 where this baseline is used. We have also clarified the text around which experiments are about different data or architecture: Figures 5, 6, and 8 are indeed more about the use of robotics data and a generalist versus a single-task specialist, while the Gato comparison is to demonstrate the benefit of diverse robotics data versus an similar architecture trained on data from other domains.
>
>
> **_If my reading is correct that there isn't a directly comparable baseline method with the same data and different architecture, is there some particular justification for the VQGAN + autoregressive transformer? I can see some possible interest in in-context learning, but contexts seem too short to leverage that._**
>
> The architecture we use is very similar to that of the Gato paper (Reed et al, 2022), and our choice of the VQGAN tokeniser is largely because it was easier and faster to train than the patch-based tokeniser used in that work. We show experiments in the appendix with the patch tokeniser baseline (see Section 5.4, pointing to Appendix D.3) but have tried to be careful not to claim this choice as a key contribution. We have now added clarification of this into the introduction.

---

> > ### Author Response · Authors · 2023-10-04
> > **Response to Reviewer qZkA (2/2)**
> >
> > **_From above, I think this paper missed an opportunity to make a comparison to finetuned language models (using the same VQGAN tokens as the proposed method as input). All baselines are pretrained visual foundation models which may be adept at reasoning over the content in the image, but less adept at reasoning over the history. It seems pretty straightforward, and supported in the literature to use language models as a base for control (https://arxiv.org/abs/2103.05247) given that they see an enormous variety of sequences in their data._**
> >
> > This is great feedback. Note that the paper you mentioned uses language models for various non-language tasks but not control. However, there are other papers we cite that do indeed leverage language models or language datasets for control (eg. Driess et al. (2023)).
> >
> > We agree that exploiting LLMs and language structure may have strong temporal reasoning properties, but in this work we wanted to carefully explore the benefits of visual goal-conditioning, and hence also used vision foundation models as baselines. We have now added this to the discussion and future work section.
> >
> > **_The use of" 0-shot" and "k-shot" when referring to finetuning in Section 5.2 is a bit confusing. I would take this terminology to refer to in-context learning, not the size of the dataset for further training. I can understand why in-context learning wasn't explored here, given that it takes 1024 tokens to model only 3 steps of time, but on a first read I did not parse this._**
> >
> > From our experience, the use of zero-shot and k-shot is also common for training/finetuning, eg. in the few-shot learning or meta-learning literature, rather than just for few-shot prompting. We feel this is the clearest way to emphasise how many examples are used, but to ensure there is no confusion, we have added a sentence on this upfront, and ensured that any mention of “few-shot” also mentioned finetuning.
> >
> >
> > **_Requested changes_**
> >
> > As requested, we have answered the first few questions in the weaknesses section (and made changes in the text). We have also clarified the role of the different baselines, and changed the phrasing of contributions to delineate data, architecture, and baseline comparisons. We hope this addresses all of your concerns.

---

> > > ### Author Response · Authors · 2023-10-07
> > > **Re: VIMA comparison**
> > >
> > > Thanks for your suggestion, this is an important point we should be clearer about. Although related to RoboCat, we see VIMA's contribution as orthogonal to this work. We provide more details next and will also add this discussion in the paper.
> > > VIMA is similar to our work in that it aims to train a general manipulation agent. Unlike our work, it also looks at language grounding and few-shot prompting. However, these capabilities are possible due to the high-level input and output spaces used. Unlike robocat, which learns to directly map from images to end-effector displacement and rotation, it is our understanding that  VIMA takes as input bounding boxes of the relevant objects and outputs primitives such as “pick and place” or “wipe”, effectively sidestepping the control problem. Training VIMA on a multi-embodiment setting would require manually writing controllers for each embodiment to “pick and place” or “wipe” and training object detectors for each object of interest.

---

> > > > ### Comment · Reviewer_qZkA · 2023-10-16
> > > > **Response to "Re: VIMA comparison"**
> > > >
> > > > Thanks for taking this up in more detail. I have since had a look through VIMA again, and I agree that it is orthogonal as there is language-grounding and a much higher-level action space. I appreciate the clarification.

---

> ### Comment · Reviewer_qZkA · 2023-10-05
> **Response to response**
>
> Thanks for the prompt and specific responses, I'll go into detail below.
>
> ***In section 4.2 the full model size is given at 1.18B...***
>
> Thank you for the clarification. Now that this is established, it still seems that there is a missing number for the smaller RobotCat model in order to match the scales of baselines. Again it is fine that this number is not SOTA, but it is important to see in order to understand the contribution. If the argument is that it is too unfair to compare a similarly sized multitask model to single-task models, then wouldn't it be possible to use the RoboCat-lim setting in Section 5.2 for each respective task (or perhaps on the subset of tasks that RoboCat-lim is already tested, to make the experiments less burdensome)?
>
> ***The method uses ~3 timesteps of conditioning (Section 2.1.1)...***
>
> Thank you for the clarification.
>
> ***Baseline methods are built on existing models pretrained from a different data sources that do not include images close to the deployment task...***
>
> Thank you for the clarification again. I appreciate the change in the writing in Section 4.4.
>
> ***There seem to be good already existing, and cited, baselines close to the tasks considered like VIMA (https://arxiv.org/pdf/2210.03094.pdf) and GATO, I'm a bit confused as to why these were not chosen. VIMA especially as it is open-source. There is only a single citation with no discussion as to why it wouldn't make sense to compare in Section 6.1.***
>
> It seems this was missed on the response to the initial review. Would you please provide a response to this here? You have helped my understanding clarifying the comparison to GATO in Table 2, but I still don't understand why VIMA was passed over. It is particularly strange to me that this comparison was not made given that VIMA is also open-source, and predates the method by a year.
>
> ***If my reading is correct that there isn't a directly comparable baseline method with the same data and different architecture...***
>
> Thanks for the clarification.
>
> ***From above, I think this paper missed an opportunity to make a comparison to finetuned language models...***
>
> Thanks for the clarification, and for catching my mistake with the citation. I don't think it is absolutely critical to acceptance, but this seems like a clear way to round out the presentation and justify the large collection of control data required here.
>
> ***The use of" 0-shot" and "k-shot" when referring to finetuning in Section 5.2 is a bit confusing...***
>
> On a second skim of past literature, I agree that this is probably an instance where the terminology has crept a bit, and 0-shot/k-shot was not uncommon to refer to the size of the finetuning dataset in the past. Still I appreciate the change in the wording in the text as I think it helps.
>
> ***Requested changes***
>
> Thanks for the changes so far, they definitely help my understanding and I believe they improve the presentation. There was an additional request for an experiment or some reasoning as to why VIMA was not tested given its close similarities to the proposed method. I would still like a response to that.

---

> > ### Author Response · Authors · 2023-10-17
> > **2nd Followup response to reviewer qZkA**
> >
> > *__Now that this is established, it still seems that there is a missing number for the smaller RobotCat model in order to match the scales of baselines. Again it is fine that this number is not SOTA, but it is important to see in order to understand the contribution. If the argument is that it is too unfair to compare a similarly sized multitask model to single-task models, then wouldn't it be possible to use the RoboCat-lim setting in Section 5.2 for each respective task (or perhaps on the subset of tasks that RoboCat-lim is already tested, to make the experiments less burdensome)__*
> >
> > Thanks for persisting with this point, it will help to make the paper stronger. With earlier experiments in the RoboCat-lim setting, we found that the size does make a difference: we have now repeated these experiments for a fair comparison. In summary, with a smaller 364M model for RoboCat-lim, the performance decreases by 6-7% on the simpler stacking tasks, and 12-21% on the harder pyramid and tower building tasks.
> >
> > We put this analysis in Appendix G.2 and reference it from the main body of the paper (Section 5.4). Note that the baseline comparison in Figure 5a (sim training tasks) is with randomly-selected single tasks from each task family, while the model size comparison uses all tasks.
> >
> >
> > *__[Regarding citation on Pretrained Transformers as Universal Computation Engines] I don't think it is absolutely critical to acceptance, but this seems like a clear way to round out the presentation and justify the large collection of control data required here.__*
> >
> > Thanks for the suggestion, we have added a reference to this work in the paper.
> >
> > *__Other points__*
> >
> > Thank you for your systematic response to all of our points, and please let us know if there are any other issues to address before publication.

---

> > > ### Comment · Reviewer_qZkA · 2023-10-26
> > > **Response to size comparison**
> > >
> > > ***Now that this is established, it still seems that there is a missing number for the smaller RobotCat model in order to match the scales of baselines. Again it is fine that this number is not SOTA, but it is important to see in order to understand the contribution. If the argument is that it is too unfair to compare a similarly sized multitask model to single-task models, then wouldn't it be possible to use the RoboCat-lim setting in Section 5.2 for each respective task (or perhaps on the subset of tasks that RoboCat-lim is already tested, to make the experiments less burdensome)***
> > >
> > > I appreciate the additional results in Figure 26, but it appears I was perhaps not precise enough in specifying what I thought was missing. I would like to see a comparison between the smaller RoboCat model and **the baselines**, not the larger RoboCat model. A size comparison plot that only compares RoboCat to itself does not satisfy that. Perhaps I misunderstood that the initial response to this point was that it was unfair to compare baselines to RoboCat with the same data because it would be way too expensive to train RoboCat models for each task individually, which is why I suggested the RoboCat-lim setting, and comparing against single-task baseline models on those in-distribution tasks. It seems to me, for the sake of transparency, the results of this comparison should either be included in the main text (like Figure 5a), or the conclusions be summarized (section 4.2 or 5.4), not only a reference to the appendix with no further discussion.
> > >
> > > Thanks for your continued engagement, I appreciate the back and forth.

---

> > > > ### Author Response · Authors · 2023-11-01
> > > > **Response re: size comparison**
> > > >
> > > > Thanks for the followup comment and clarification. Your interpretation is correct: we did initially feel it would be an unfair comparison to constrain RoboCat to be the same size as the single task models, given its capacity is used to solve many tasks jointly.
> > > > However, based on your feedback we agree it could be useful to have this comparison to gauge how effective a given model capacity can be in single and multi-task settings.
> > > >
> > > > This actually requires training a different-sized version of the full RoboCat from scratch, which will take some time. We have started a training run and will put the results in Figure 5a when ready, with discussion in the main text.
> > > >
> > > > Thanks again for your suggestion and further clarifications.

---

> > > > > ### Comment · Reviewer_qZkA · 2023-11-06
> > > > >
> > > > > Thanks for following up, and for taking the time to clarify these points, I think it makes the argument much clearer.

---

> > > > > > ### Author Response · Authors · 2023-11-10
> > > > > > **The updated experiments are in the paper**
> > > > > >
> > > > > > Thanks for your patience. We have now completed the necessary experiments and evaluations, and updated both Figure 5a and the associated discussion to compare the performance of the smaller 364M model.
> > > > > >
> > > > > > We hope this addresses your concerns, and appreciate the continued engagement.

---

> ### Comment · Reviewer_qZkA · 2023-11-10
>
> Thanks for the update. I had a look and I feel now that I can start to disambiguate the contributions. I feel now my main concerns are addressed. Of course it would be interesting to understand some more qualitative failures that size helps with, but I believe the paper is already quite long and this can be deferred to the future.

---

### Review · Reviewer_PTw5 · 2023-10-12

**Summary Of Contributions:**

The paper presents RoboCat, a goal-conditioned decision transformer for robotic manipulation that works on visual inputs across different robot arm embodiments and tasks, both in simulation and in the real world. Zero-shot generalization and adaption via learning from demonstration are shown. Furthermore, the model is shown to be capable of generating training data for self-improvement.

**Audience:**

Yes

**Broader Impact Concerns:**

Broader Impact concerns are addressed in sufficient detail.

**Claims And Evidence:**

Yes

**Requested Changes:**

- mention the action space already in the main body of the paper
- if possible, provide an argument or an experiment that proves that multi-embodiment plays a role. I.e., that the improved performance is not just due to having more data. Perhaps one could achieve the same results using a single robot and then just running the learned policy on a different robot.

**Strengths And Weaknesses:**

Strengths
- The paper provides a **thorough empirical evaluation** of the proposed method on different robot arms and tasks
- Specific questions are addressed, proving that cross-task transfer and self-improvement are possible, among others
- The method and experiments are described in great detail, both in the paper and in the appendix
- The writing is clear and easy to follow

Weaknesses
- Moderate methodological novelty: **the method to a large extent is based on Gato**. The difference is in using a different image tokenizer, VQ-GAN, and in predicting image tokens in addition to action tokens.
- Low variability across robots/embodiments: just from reading the abstract and the introduction, it may seem that the paper handles all kinds of robots and tasks. But in fact, the experiments are only done using Sawyer, Franka, and Kuka arms, which all have quite similar kinematics. Furthermore, the control action seems to be the 6 DoF end-effector Cartesian velocity (plus gripper command) — thus the impact of the **"robot embodiment" seems to be basically irrelevant for the learning**, because the tracking of the end-effector velocity is handled by the internal robot controller.

---

> ### Author Response · Authors · 2023-10-17
> **Response to Reviewer PTw5**
>
> Thank you for recognising the strengths of our work, and we appreciate the constructive feedback! We have added a new revision of the paper and highlighted the changes that are relevant to this review, in brown. We hope that this will contribute to an easier and clearer discussion, and we address your comments below.
>
> *__Requested change: mention the action space already in the main body of the paper__*
>
> Thanks for this suggestion and your more detailed comments around this. We have now mentioned in the main body (Section 3.1) that the embodiments use end-effector control, to avoid any confusion about our work.
>
> To clarify further, we employ both 4-DoF and 6-DoF end-effector control (plus the gripper action) for different tasks across the Panda and Sawyer robots in sim and real, and the two robots also have different proprioception observations leading to different input dimensions. While the KuKA also employs end-effector control, gripper control is no longer a 1-DoF action but a 8-DoF dexterous hand. The difference in physical and kinematic characteristics of the different arms also means that they have different state-action distributions, which is part of the challenge of multi-embodiment learning. We have now clarified all of these details in the text.
>
>
> *__If possible, provide an argument or an experiment that proves that multi-embodiment plays a role. I.e., that the improved performance is not just due to having more data. Perhaps one could achieve the same results using a single robot and then just running the learned policy on a different robot.__*
>
> One result that shows this is in our generalisation study, where we aim to carefully study the zero-shot and fine-tuning capabilities of the agent across various axes. For the embodiment axis, we show that zero-shot performance on the unseen KUKA embodiment is zero (given it has an entirely different action and observation space), but with a relatively small amount of KUKA data, we can fine-tune the RoboCat-lim model to achieve 69% success. The result with a held-out object triplet also shows the benefit of multi-embodiment training: despite the Panda and Sawyer having different observation spaces, training with some Sawyer stacking data enables zero-shot generalisation to held-out Sawyer tasks that have only been seen for the Panda. We have clarified discussion around this in Section 5.2.

---

> > ### Comment · Reviewer_PTw5 · 2023-10-18
> > **Thank you, my concerns are resolved**
> >
> > Thank you for the clarifications and for the added discussions in the paper. Especially the added paragraph in Sec. 5.2 about the impact of multi-embodiment training on zero-shot transfer strengthens the argument quite well. With that, all my concerns are addressed.

---

### Author Response · Authors · 2023-10-17
**Response to all reviewers**

Thanks to all of the reviewers for their constructive comments and continued engagement with our work. From our understanding, most concerns were around the clarity of the paper, in terms of the claims made, details about the method, and the experimental design and conclusions drawn. We have attempted to systematically address all of the points raised, and made significant changes throughout the revised submission. For clarity, we have also colour-coded edits based on the comments they are intended to address: blue for reviewer 68NV, red for reviewer qZkA, and brown for reviewer PTw5.

We hope this addresses the concerns raised, and welcome any further discussion.

---

### Decision · Action_Editor_Uo7W · 2023-11-27

**Recommendation:** Accept as is

**Comment:**

The paper proposes RoboCat, a large visual, goal-conditioned transformer model that enables different robotic arms to perform a variety of robotic manipulation tasks in both simulation as well as the real world. RoboCat is capable of zero-shot generalization to new tasks and robots and can be adapted based on a small number of demonstrations. Additionally, the paper shows that the model is capable of generating synthetic data with hindsight relabeling that improves the performance of fine-tuning (self-improvement). The paper investigates RoboCat's effectiveness through a large set of experiments that involve performing various tasks using several different robot arms in both simulation and the real world.

The paper was reviewed by three reviewers from the robot learning community, and there was a healthy amount of discussion between the reviewers and the authors. The topic of the paper---the proposal of a large "generalist"/"foundation" that enables different robot arms to perform different manipulation tasks---is both highly topical and of significant interest to the community. As noted by Reviewer qZkA, RoboCat is the "First large-scale multi-embodiment agent that I'm aware of, really cool and of definite interest to the community given the trend toward large models that others can build on". The reviewers appreciate the thorough empirical evaluation on various robot arms and tasks that demonstrate the capabilities of the model with regards to cross-task transfer and self-improvement. Additionally, the reviewers emphasize that the paper is well written and that the approach is simple and clearly presented, which supports reproducibility.

The reviewers initially identified a few key weaknesses with the paper as originally submitted. Among them, all three reviewers felt that the paper overstated the generalizability of the model to different robots, tasks, and environments as would be expected of a "foundation" model. In particular, the reviewers pointed out that the abstract and introduction emphasized results on a large number of robots and tasks, yet the arms exhibit very similar kinematics and employ an action space that facilitates transfer; as well as that the paper only evaluates on two new downstream tasks. After an extensive discussion with the reviewers regarding these concerns, the authors tempered the paper's claims and rephrased their labeling of RoboCat as a "generalist" model rather than a "foundation" model. The reviewers commented that these changes help to clarify the contributions, which the reviewers find to be well aligned with the experimental results. With these updates to the text, the paper provides a solid contribution to the community.

**Audience:**

The topic of the paper---the proposal of a large "generalist"/"foundation" that enables different robot arms to perform different manipulation tasks---is both highly topical and of significant interest to the robot learning community, a point that several reviewers emphasize.

**Claims And Evidence:**

The paper proposes RoboCat, a large visual, goal-conditioned transformer model that enables different robotic arms to perform a variety of robotic manipulation tasks in both simulation as well as the real world. Extensive experimental evaluation in both simulation and the real world demonstrates that RoboCat is capable of zero-shot generalization to new tasks and robots and can be adapted based on a small number of demonstrations. Additionally, the paper shows that the model is capable of generating synthetic data with hindsight relabeling that improves the performance of fine-tuning (self-improvement). As detailed in the comments below, all three reviewers raised concerns with the claims not being supported by empirical evidence in the original submission, specifically with regards to the model's generalizability to different robots, tasks, and environments. Following a back-and-forth discussion with the reviewers, the authors revised the text, which the reviewers now find is well aligned with and supported by the experimental evidence in the paper.